# Disease-linked regulatory DNA variants and homeostatic transcription factors in epidermis

Douglas F. Porter[1,11] ✉, Robin M. Meyers[1,2,11], Weili Miao [1,11], David L. Reynolds[1,11], Audrey W. Hong[1], Xue Yang[1,3], Suhas Srinivasan [1], Smarajit Mondal [1], Zurab Siprashvili[1], Tania Fabo [2], Ronghao Zhou[2,4], Tri Nguyen[2,4], Luca Ducoli [1], Jordan M. Meyers[1], Duy T. Nguyen[1], Lisa A. Ko [1], Laura N. Kellman [1,3], Ibtihal Elfaki[1], Margaret Guo[1,5], Mårten CG Winge[1], Leandra V. Jackrazi[1], Vanessa Lopez-Pajares [1], Betty B. Liu [2], Yuanhao Qu [3], Imani E. Porter [2,6], Samuel H. Kim [3], Gyuhyeon Kim[1,2], Shiying Tao[1], Jesse M. Engreitz [2,4,7,8,9] & Paul A. Khavari [1,3,10] ✉

Identifying noncoding single nucleotide variants (SNVs) in regulatory DNA linked to polygenic disease risk, the transcription factors (TFs) they bind, and the genes they dysregulate is a goal in polygenic disease research. Here, we use massively parallel reporter analysis of 3451 SNVs linked to risk for polygenic skin diseases with disrupted epidermal homeostasis to identify 355 differentially active SNVs (daSNVs). daSNV target gene analysis, combined with daSNV editing, underscored dysregulated epidermal differentiation as a shared pathomechanism. CRISPR knockout screens of 1772 human TFs revealed 123 TFs essential for epidermal homeostasis, highlighting ZNF217 and CXXC1. Population sampling CUT&RUN of 27 homeostatic TFs identified allele-specific DNA binding (ASB) differences at daSNVs enriched near epidermal homeostasis and monogenic skin disease genes, with notable representation of SP/KLF and AP-1/2 TFs. High TF-occupancy promoters were "buffered" against ASB. This resource implicates dysregulated binding of specific homeostatic TF families in risk for diverse polygenic skin diseases.

Skin tissue is the site of diverse polygenic disorders that comprise a substantial global burden of human disease[1]. The epidermis is structured as a multilayered differentiating epithelium continually regenerated by a population of keratinocyte stem cells in the basal layer (Fig. 1a). Differentiated epithelial cells form the cutaneous barrier to desiccation and to external injury while also producing pro-inflammatory factors for response to barrier disruption, infection, and tissue damage. The gene expression changes accompanying

[1]Program in Epithelial Biology, Stanford University School of Medicine, Stanford, CA, USA. [2]Department of Genetics, Stanford University School of Medicine, Stanford, CA, USA. [3]Program in Cancer Biology, Stanford University School of Medicine, Stanford, CA, USA. [4]Basic Sciences and Engineering Initiative, Betty Irene Moore Children's Heart Center, Lucile Packard Children's Hospital, Stanford, CA, USA. [5]Program in Biomedical Informatics, Stanford University, Stanford, CA, USA. [6]Department of Chemistry and Biochemistry, Hampton University, Hampton, VA, USA. [7]Stanford Cardiovascular Institute, Stanford University School of Medicine, Stanford, CA, USA. [8]The Novo Nordisk Foundation Center for Genomic Mechanisms of Disease, Broad Institute of MIT and Harvard, Cambridge, MA, USA. [9]Gene Regulation Observatory, Broad Institute of MIT and Harvard, Cambridge, MA, USA. [10]Veterans Affairs Palo Alto Healthcare System, Palo Alto, CA, USA. [11]These authors contributed equally: Douglas F. Porter, Robin M. Meyers, Weili Miao, David L. Reynolds. ✉e-mail: dfporter@stanford.edu; khavari@stanford.edu

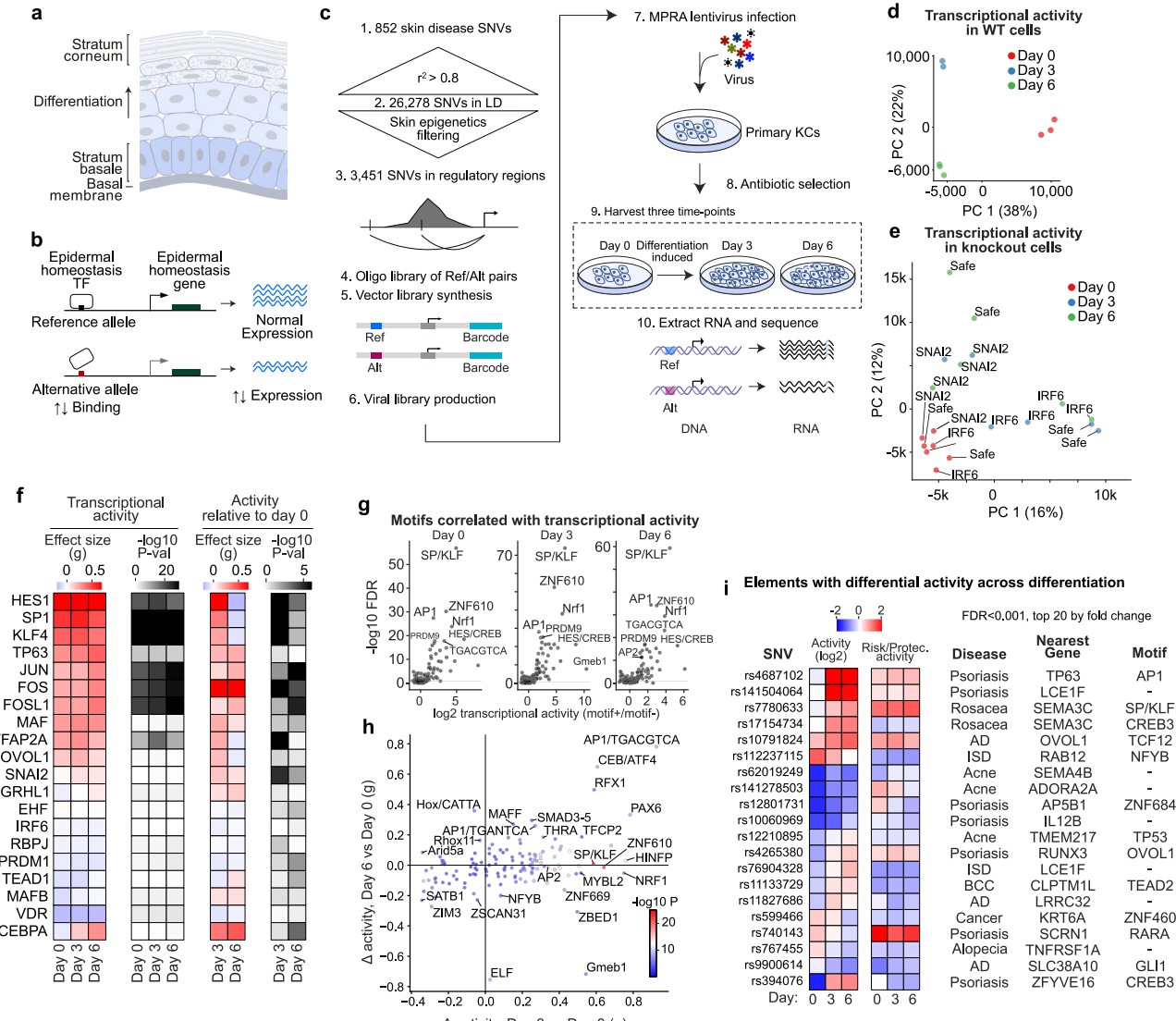

**Fig. 1 | Activity of regulatory variants linked to polygenic skin disease.**
**a** Epidermal schematic. **b** Hypothetical model for one mechanism whereby non-coding variation may affect polygenic skin disease risk. **c** MPRA experimental diagram for assaying variants correlated with skin disease (day 0, undifferentiated keratinocyte populations, and day 3 and 6 of calcium induced differentiation). **d** PCA of MPRA activity (linear scale, outliers clipped) across differentiation shows distinct transcriptional states. **e** PCA of transcriptional activity in keratinocytes treated with CRISPR guides knocking out IRF6 or SNAI2 TFs rotate cells through the transcriptional states of differentiation. "Safe" cells were treated with a safe targeting guide. **f** Activities of known skin TFs correlate with transcriptional activity. Effect size (Hedge's g, or the difference in means divided by the pooled standard deviation, with a correction factor for small samples) measures the difference in element activity for fragments with the given motif *vs* fragments without, and the *P*

value is the two-sided Mann–Whitney test for a difference in activity. When relative to day 0, the day 0 activity is first subtracted from each element. **g** Association of transcriptional activity with JASPAR motif clusters (as fold increase in activity for elements with the motif) suggests SP/KLF motifs are the strongest in promoting transcription, followed by AP-1 and NRF1 motifs. **h** Effect sizes and significance values (hue, two-sided Mann-Whitney) for a change in motif-activity associations across differentiation; note transiently increased activity of SP/KLF, NRF1, and AP-2 motifs at day 3, and sustained activity of CEBP/ATF and AP-1 motifs from early to late differentiation. Gmeb1 is the glucocorticoid receptor binding site, which is transiently activated, then repressed. *P* value, Mann–Whitney. **i** daSNVs with dynamic activity (FDR < 0.001, top 20 by fold change) across differentiation increase activity and reside near differentiation genes.

keratinocyte differentiation in tissue can be replicated in vitro, facilitating identification and study of the regulatory DNA elements and transcription factors (TFs) that impact this process[2–4]. Disruption of the balance between epidermal stem cell proliferation and differentiation is a histopathologic feature observed in prevalent polygenic inflammatory and neoplastic skin diseases[5,6]. Among these disorders are psoriasis vulgaris, atopic dermatitis, and nonmelanoma skin cancers, such as basal cell carcinoma (BCC) and squamous cell carcinoma (SCC). The summed incidence rate of polygenic skin disease, excluding mild acne, is almost 5 out of 100 U.S. adults per year (Supplementary Fig. 1). Further, monogenic disorders such as autosomal recessive

ichthyoses exemplify how loss-of-function variants in key differentiated skin barrier genes[7] cause disease. However, the extent to which dysregulated epithelial homeostasis is linked to inherited regulatory variants that modulate expression of genes important for this process is unclear.

The identification of germline risk loci by genome wide association studies (GWAS) provides a pathway to explore the biologic processes that disease variants may dysregulate in the pathogenesis of polygenic skin disease. A majority of the GWAS-identified risk variants linked to disease in skin and other tissues reside in noncoding DNA[8–10]. These noncoding disease risk-linked variants enrich in open chromatin

with characteristics of regulatory DNA and have therefore been postulated to alter expression of specific target genes in ways that may enable the emergence of disease[11,12]. However, owing to linkage disequilibrium, genomic risk loci can encompass numerous single nucleotide variants (SNVs), an observation which has stimulated interest in high throughput assays that assess variant function to distill those that may impact gene regulation, such as massively parallel reporter analysis[13–15] (MPRA). Single nucleotide resolution compendiums of functional SNVs linked to disease may also aid efforts to better identify potentially dysregulated target genes through techniques such as gene editing, providing additional insight into the pathomechanisms that underly polygenic disease predisposition.

The binding of TFs to their cognate DNA motifs within enhancers and promoters represents a central process in gene regulation[16]. In epidermis, a number of TFs have been identified that are required to maintain the normal balance between epidermal cell growth and differentiation required for tissue homeostasis[17–19]. Some of these non-redundant homeostatic epidermal TFs fall into subgroups with opposing impacts on growth and differentiation. The first subgroup includes TFs, such as SNAI2[20] and TEAD1[21], that sustain undifferentiated progenitor gene expression while repressing differentiation. A second TF subgroup represses progenitor genes while being required for the induction of epidermal differentiation genes, and includes TFs such as IRF6[22] and ZNF750[23]. A global understanding of the TFs essential for epidermal homeostasis, such as might be gained by gene knockout screening to disrupt all TFs, however, is presently lacking. In this regard, datasets that map dynamic TF binding across the genome during the progenitor transition to terminal differentiation and link them to disease-risk SNVs may provide insight into the disruption of normal epidermal gene expression that accompanies polygenic skin disease. Such studies represent incremental early steps in the long-term effort to connect disease risk-linked regulatory DNA variants to the TFs and target genes they dysregulate in the relevant tissue cell types where polygenic disease unfolds.

Identifying disease-linked SNVs that alter transcription in epidermis and other tissues may also help extend the understanding of how DNA variants contribute to pathogenic risk to the level of biochemical mechanism. For example, it may define the TFs whose DNA binding and regulatory activity is altered by individual disease-linked SNVs (Fig. 1b). In this regard, TFs that control tissue function may display differences in allele-specific binding (ASB) at sites of DNA sequence variation, which can result in altered gene expression[24,25]. Insight into the specifics of this paradigm, however, requires addressing a number of current knowledge gaps. These include a paucity of functionally interrogated compendia of disease risk-linked variants in regulatory DNA in disease-relevant cell types and cell states and incomplete identification of homeostatic TFs necessary for health of specific human tissues[18]. These gaps also include an imperfect understanding of the transcriptional networks underlying tissue homeostasis[26] as well as limited linkage of risk variants to the TFs whose DNA binding they may alter at specific genomic sites in relevant contexts[27]. Understanding the functional disease risk-linked DNA variants and the TFs whose activity they affect may provide additional insight into molecular mechanisms of polygenic disease pathogenesis.

Here we generate a compendium of 355 skin disease-linked, differentially active regulatory SNVs (daSNVs), 123 essential homeostatic epidermal TFs, and 298 high-confidence ASB instances in progenitor and differentiating human epidermal keratinocyte populations. The outline of experiments in this work (MPRA, perturb-seq, CUT&RUN) and their connections is diagrammed in Supplementary Fig. 2. daSNVs display dynamic transcription-directing activities across epidermal differentiation that differ by the DNA motif sequences they encompass. Putative daSNV targets are enriched for genes important in epidermal differentiation and inflammation. Essential homeostatic epidermal TFs identified by CRISPR screening encompass previously

known factors as well as newly identified TFs with non-redundant roles in epidermal homeostasis, among which ZNF217 and CXXC1 exerted some of the strongest effects on genomic expression of any TF. Mapping genomic binding of 27 homeostatic TFs identified enrichment for ASB events concentrated at genes important in epidermal differentiation as well as those whose coding sequence mutation causes monogenic human skin disease, proposing biochemical underpinnings of disease-linked DNA sequence variation. TF genomic binding data also supported the existence of a "promoter buffering" paradigm, in which promoter-TF interactions are less sensitive to alterations in DNA sequence than distal regulatory elements. Integrative analyses of these data link specific TFs to disease-related SNVs that dysregulate stem cell differentiation and inflammation genes across diverse common polygenic human skin diseases.

## Results
### Regulatory variants linked to skin disease
To identify variants that may influence epithelial disease risk by altering transcription, a set of 852 lead SNVs linked to risk for common polygenic skin disease was collected from 57 GWAS studies[28] (Supplementary Data 1). Linkage disequilibrium (LD) score correlation suggested basal and SCC share genetic risk variants, whereas other skin diseases have distinct variants from cancer (Supplementary Fig. 3a, b). These inflammatory and neoplastic skin disorders include psoriasis, atopic dermatitis, seborrheic dermatitis, acne rosacea, cutaneous lupus erythematosus, dermatomyositis, alopecia areata, BCC, SCC, as well as cohorts classified broadly as having inflammatory skin disease and multiple keratinocytic cancers. In aggregate, these polygenic skin diseases afflict over 80 million persons in the U.S. alone[29] and each displays evidence of altered epithelial growth, inflammation or differentiation. For example, BCC SNVs are enriched in differentiated keratinocyte ATAC peaks, while atopic dermatitis risk variants are found more frequently in T cells and epithelial cells, including differentiated keratinocytes (Supplementary Fig. 3c, d, full results in Supplementary Data 1), consistent with both keratinocyte-intrinsic and immune cell mechanisms. To include adjacent SNVs with functional impacts on transcriptional regulation, the 852 lead SNVs were expanded to 26,278 SNVs in LD. To focus on variants potentially functional in epidermal keratinocytes, these were then filtered to only include SNVs present in keratinocyte open chromatin regions via ATAC-seq and/or the histone marks H3K27ac/H3K4me1[4], indicative of transcriptional regulatory regions. This produced a final set of 3451 SNVs linked to skin disease risk enriched for features of regulatory DNA in epidermal cells (Fig. 1c, Supplementary Fig. 4a, b). These SNVs were cloned upstream of a minimal promoter driving barcoded reporter expression. A lentiviral library of variants was infected into primary diploid human keratinocytes for MPRA, then barcodes of reporter RNA sequenced (Supplementary Data 2)[30,31]. To mimic the progenitor differentiation process, keratinocytes were harvested either without inducing differentiation ("day 0"), or after 3 or 6 days of calcium-induced differentiation in vitro[2,32–34] to model the transition from the undifferentiated progenitor state through early and late keratinocyte differentiation, respectively.

MPRA libraries were complex and not skewed (Supplementary Fig. 4c, d). All barcodes were observed (Supplementary Fig. 4e, f) and replicates correlated (Supplementary Fig. 4g, h), with the largest difference between Day 0 and Day 3/6 (Supplementary Fig. 4g), consistent with changes in the transcriptional milieu during the differentiation process. There were fewer active enhancers as differentiation progressed, suggesting a general downregulation of transcription at these elements (Supplementary Fig. 4i, $P = 0.002$, two-sided Mann–Whitney of activity values). Transcriptional activity was most positively correlated with DNA sequences from transcription start site (TSS) regions, and negatively correlated with genomic sequences from regions that are quiescent or poorly transcribed in

keratinocytes (Supplementary Fig. 4j), indicating that transcription-directing activity seen in MPRA corresponded with features of the originating genomic locus in situ. Interestingly, many TSS elements activated by day 3 of differentiation were reversed by day 6 (Supplementary Fig. 4k), consistent with the premise that late differentiation represents a more transcriptionally muted state. The resulting MPRA data interrogated the regulatory activity of SNV-containing fragments linked to risk for human polygenic skin disease across a timecourse of keratinocyte differentiation.

Early and late differentiation conditions displayed distinct transcriptional activity on the putative regulatory sequences tested in MPRA, with principal component analysis (PCA) distinguishing samples by differentiation time point (Fig. 1d). This pattern was also reproduced in follow-up MPRA datasets done on knockouts of specific canonical homeostatic TFs. For example, MPRA in keratinocytes across the same time course subjected to knockout of either the IRF6 (pro-differentiation) or SNAI2 (pro-progenitor) canonical TFs resulted in samples moving in the expected directions around the differentiation axis, indicating the samples are separated according to the differentiation phenotype, rather than technical biases (Fig. 1e, Supplementary Fig. 4l). Specifically, the day 3 IRF6 knock-out cells have a more progenitor-like phenotype, and day 6 IRF6 knock-out cells have a day 3-like phenotype, consistent with the impaired differentiation of IRF6 knock-outs. The opposite trend occurred for SNAI2. The transcription-directing activity of the regulatory fragments containing skin disease-linked SNVs, therefore, is modulated by differentiation state, in agreement with previously cataloged[3,4] in situ transcriptional and epigenomic changes observed during this process in epidermal cells.

### Known TF motifs correlate with activity

Motif analysis was performed to identify which factors might drive the observed dynamic activity patterns. It should be noted that the presence of a TF motif in DNA is only suggestive evidence for a TF binding to a given site, and not a reliable indicator of binding[35]. Nevertheless, the motifs of known keratinocyte regulators were evaluated, and those TFs most associated with high activity were GC-rich motifs belonging to TSS-associated proteins, such as the SP/KLF family, followed by AP-1, AP-2, HES1 and CEBPA (Fig. 1f). The activity in progenitors at each enhancer was subtracted from all samples, and motif enrichment was calculated for progenitor-normalized activity (Fig. 1f). GC-rich motifs (e.g., SP1, KLF4, HES1), TFAP2A and OVOL1 peaked in activity at day 3 of differentiation before declining to progenitor levels or below. AP-1 motifs were activated upon differentiation and remained active, in some cases (JUN, FOS, JUNB, FOSL1) becoming more so at Day 6. CEBPA, and GRHL1 motifs were also associated with progressively more reporter activity over differentiation, while the activity of the motif for the stratified epithelial lineage-determining TF, TP63, remained constant. These results support a provisional model in which transcriptional activity across differentiation passes through an initial transition state characterized in part by increased SP/KLF activity.

Analysis was then extended beyond known keratinocyte TF motifs. Clustering motifs into similar families and examining all families in the JASPAR database revealed the C-rich sequence binding SP/KLF family motifs as the most potent in driving transcription overall in MPRA (Fig. 1g). The second most active motif, assigned to the ZNF610 TF, was a C-rich motif that could be grouped with the SP/KLF family. After C-rich SP/KLF-like motifs, the next distinct motif was the AP-1 TGANTCA motif, followed in turn by the NRF1 CGCGNTGCG motif, and the second AP-1 motif cluster of TGACGTCA sequences. Within these clusters, the KLF15 and SP3 versions were the most active overall. The TGACGTCA form of the AP-1 motif and the CEBP/ATF4 motif were strongly and continuously upregulated by differentiation, while the TGANTCA AP-1 motif was less strongly but still continuously upregulated (Fig. 1h). In contrast, the SP/KLF, NRF1 and AP-2 motifs

were more transiently upregulated, and the glucocorticoid response element (Gmeb1 version) motif family was associated with early activation followed by silencing in late differentiation. These data further suggest that AP-1, AP-2, SP/KLF and NRF1 sequences are activated by early differentiation, with specific DNA elements associated with the AP-1 family remaining activated into late differentiation.

### Identification of rs4687102 as a dynamic element during differentiation

The DNA elements with the most time-differential activity (FDR < 0.001 for a timepoint-dependence by MPRAnalyze, then ranked by fold-change) that were also daSNVs were often located near differentiation genes (Fig. 1i). For example, effects on transcriptional activity from rs4687102, linked to psoriasis risk by GWAS, rise sharply during differentiation. rs4687102 is located within a region that loops to the *TP63* gene in primary keratinocytes, as found in a previous H3K27ac-based HiChIP study from our group under similar keratinocyte culturing conditions[36]. TP63 is an essential TF for skin homeostasis, has been implicated in psoriasis and is mis-expressed in this condition[37]. Additional discussion of this allele is in the *Supplementary Discussion* and Supplementary Fig 5.

### Compendium of differentially active SNVs (daSNVs)

To examine which risk alleles might affect transcription, a total of 355 differentially-active SNVs (daSNVs) were identified with allele-specific differential activity on at least one timepoint at FDR < 0.001 (Fig. 2a). GREAT-style analysis[38] revealed an enrichment of genes containing causative mutations for monogenic skin disease in the proximity of daSNVs, 22-fold vs the genome ($P = 10^{-14}$) and 1.9-fold vs other MPRA-tested SNVs ($P = 0.02$). Relative risk/protective allele activity for 229 SNVs had a significant dependence on the differentiation timepoint. daSNVs were enriched in chromVAR chromatin states "Flanking Active TSS" and "Poised TSS" (Supplementary Fig. 6a). The top hits for each timepoint exhibited substantial overlap (Fig. 2b, Supplementary Fig. 6b), with 46% of FDR < 0.001 SNVs significant in multiple timepoints (Fig. 2a, c). Time-dependent daSNVs were identified by log-likelihood ratios comparing the inclusion/exclusion of a timepoint:allele interaction. The top 20 highest-magnitude interaction effects for SNVs that were also linked to a GTEx-defined eGene (genes with expression correlated at FDR 0.05 with the SNV) had a tendency to have their highest difference at the first or last timepoint, and many eGenes were skin or inflammation-related (Fig. 2d). These data provide a compendium of SNVs that display differential transcription-driving activity across an in vitro timecourse of keratinocyte differentiation.

### TF motifs and interactions disrupted in daSNVs

TF motifs were assessed with respect to daSNVs to identify TF families that might be affected by disease-linked genetic variation. Motifs for TFs with known roles in skin homeostasis (AP-1, KLF4, SP1, TFAP2A) were highly enriched in differentially active SNVs (Fig. 2e, f, Supplementary Fig. 6c), implicating TFs that bind these DNA motifs as factors that may translate genetic variation into pathogenic gene regulation at specific loci.

Multiple representative daSNVs were in agreement with this model. The daSNV rs72696969, which affects a motif for the EHF pro-differentiation TF, is connected via H3K27ac HiChIP looping to the promoters of *FLG*, *FLG2* and the antisense *FLG-AS* epidermal differentiation genes, and is an eQTL for *FLG-AS* and *FLG2* (Supplementary Fig. 6d). The risk allele confers lower transcriptional activity, suggesting that rs72696969 might increase predisposition for atopic dermatitis through down-modulating the expression of *FLG* (Supplementary Fig. 6e), a protein essential for epidermal barrier formation that is strongly linked to atopic dermatitis risk, and which is also mutated in the monogenic skin disorder ichthyosis vulgaris[7]. Cas9-AAV-mediated gene editing of the endogenous rs72696969 SNV locus

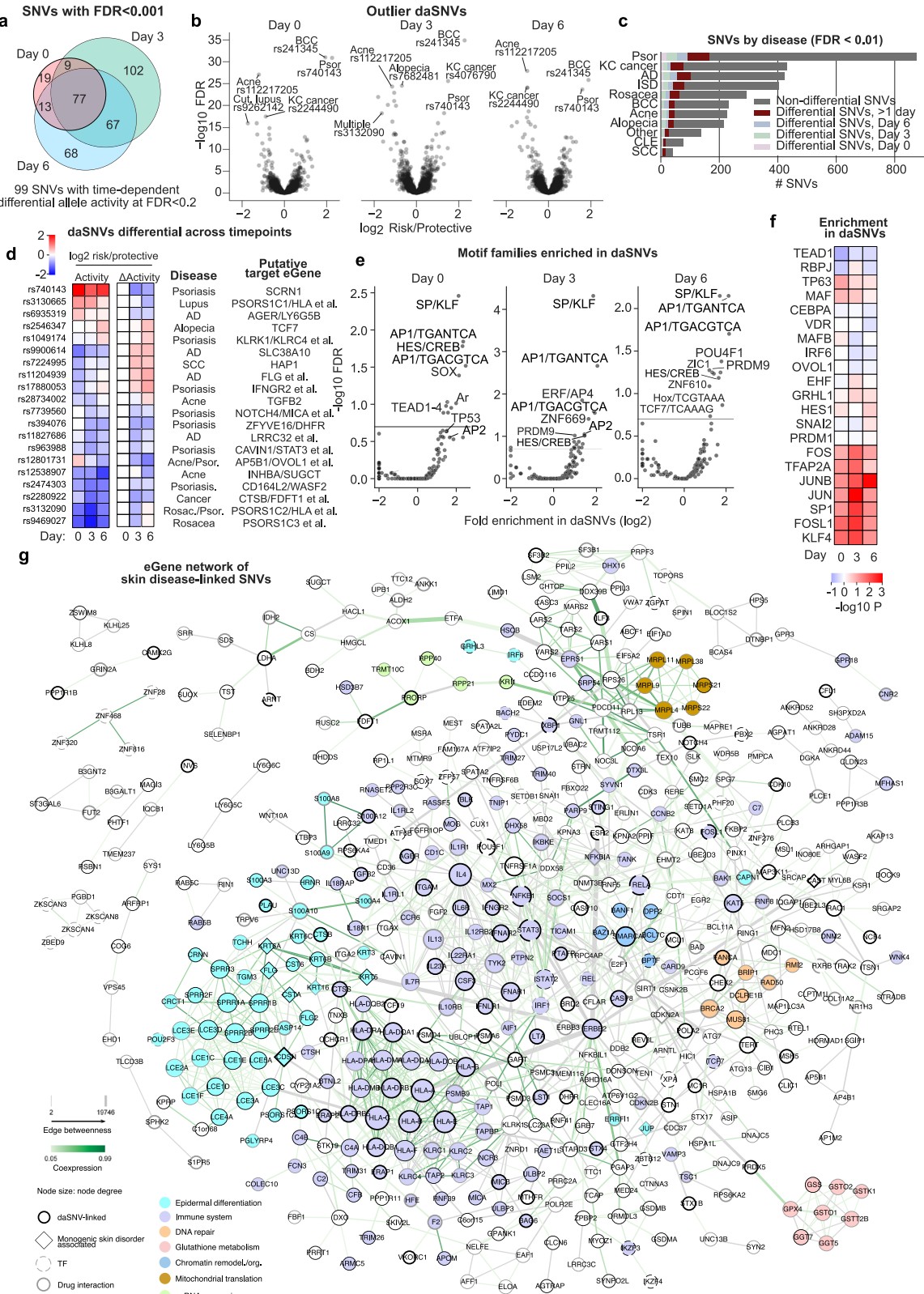

**Fig. 2 | Allele-specific transcriptional activity enriches for AP-1/AP-2/SP/KLF motifs and putative target gene networks. a** 355 differentially active (risk *vs* protective alleles) SNVs (daSNV) across a differentiation time-course. **b** Outlier daSNVs show a consistency between timepoints. **c** Total numbers of SNVs studied in specific diseases and timepoint-disease connections. **d** Time-differential daSNVs (FDR < 0.1), subset here to the top 20 strongest interaction effects for SNVs with eGenes. **e** Enrichment of JASPAR motif clusters in daSNVs vs non-daSNVs; note that SP/KLF/AP-1 motifs are frequently broken in daSNVs across all timepoints, and AP-2 motifs are enriched in daSNVs in early differentiation. **f** Enrichment in daSNVs vs non-daSNVs for the motifs of known skin TFs; SP/KLF/AP-1/AP-2 motifs are the most significantly altered (two-sided Mann–Whitney, negative values represent depletion). **g** Protein–protein interaction network of 529 disease-linked SNV-linked eGenes, with 1252 interactions. Highlight: modules identified by Markov clustering and enriched in a GO term (FDR < 0.05), except keratinization (teal) and immune system (light blue) were annotated manually or by GO term, respectively. Monogenic skin disorder genes are depicted with diamonds, daSNV-linked genes with darker borders.

was performed to generate otherwise genetically identical isogenic primary human keratinocytes that differ only at the risk versus protective daSNV of interest. To control for any gene editing effects at the locus, both risk and protective SNVs were edited (or "re-edited" in the case of the protective allele) into the protective isogenic human genetic background for direct comparison. The disease nucleotide decreased *FLG* expression (Supplementary Fig. 6f), indicating that *FLG* is a rs72696969 target gene and that the atopic dermatitis risk daSNV down-modulates expression of this differentiation gene essential for normal barrier function.

Another daSNV, rs2540334, is an eQTL for the *CASP8* gene and associated with BCC risk by GWAS. rs2540334-containing sequence loops to the *CASP8* promoter and showed altered IRF6 TF binding in vitro (Supplementary Fig. 6g–i). Nearby rs2540334, the variant rs2349075 is also an eQTL for *CASP8*, associated with BCC risk, and the risk allele is associated with decreased transcriptional activity in MPRA (Supplementary Fig. 6g, j). The risk allele breaks an AP-1 motif and results in decreased binding to JUN and JUNB in CUT&RUN of a heterozygous individual, and to JUN in vitro (Supplementary Fig. 6k, l). Edits to the endogenous locus resulted in decreased *CASP8* expression (Supplementary Fig. 6m). These results fit a model in which rs2349075 decreases TF binding to decrease *CASP8* expression, possibly leading to less efficient death of pre-malignant epidermal cells, and thereby increase lifetime skin cancer risk. Altogether, MPRA identified candidate SNVs for further molecular characterization of target gene regulation and TF binding and suggested potential roles for AP-1/AP-2/SP/KLF TFs in translating disease risk-associated variants into altered gene regulation.

## Disease-linked SNV putative target gene network

Disease-linked SNV-linked eGenes (GTEx FDR < 0.05) that could be mapped to STRING were pruned to remove isolated genes and binary interactions, resulting in a network (Fig. 2g) of 529 genes and 1252 interactions, which includes 50 TFs, 285 genes linked to daSNVs, 158 genes with drug interactions and 13 genes associated with monogenic epithelial disorders. Node degree was used to nominate interaction hubs for the STAT3 TF and the ERBB2 receptor tyrosine kinase. Edge betweenness quantified interactions that serve as bottlenecks to various parts of the interaction network[39]: the paths through ERBB2, such as ERBB2-STAT3 and ERBB2-IQGAP1, have large edge betweenness measures and a functional change in one of these genes could have a cascading effect in different parts of the network, which may relate to the critical role of ERBB2 and STAT3 in epidermal homeostasis and carcinogenesis as well as to their impacts in homeostasis-disrupting cancers of other epithelial tissues[40].

Some of the dysregulated gene modules identified by Markov Clustering were functionally enriched with Biological Process GO terms (Fig. 2g). Gene co-expression mirrored patterns in dense modules (e.g., keratinization and cytokine signaling). Central known homeostatic epidermal TFs, including *TP63, KLF4*, and *OVOL1* are eGenes of daSNVs with FDR < 0.01 (Supplementary Data 2) but did not have protein–protein interactions (PPIs) in STRING that place them in Fig. 2g. Genes involved in epidermal differentiation comprised a prominent portion of the network, including those encoding late cornified envelope proteins, small proline-rich proteins (SPRRs), and S100 proteins with known roles in terminal differentiation in epidermis. Skin eGenes are enriched in the initial variant set and related GO terms are not significantly enriched further in the daSNV eGene set (FDR > 0.2, simulation model). A number of these differentiation genes are also affected by pathogenic coding mutations causal for specific monogenic skin disorders. Among these genes and their corresponding Mendelian disorders are *CDSN* (peeling skin syndrome 1), *FLG* (ichthyosis vulgaris), *FLG2* (peeling skin syndrome 2), *CASP14* (autosomal recessive congenital ichthyosis), *KRT6A* and *KRT16* (pachyonychia congenita),

*CAST* (peeling skin with leukonychia), *NFKBIA* (ectodermal dysplasia), *TERT* (dyskeratosis congenita), *JUP* (palmoplantar keratoderma), *CSTA* (peeling skin syndrome 4), and *CTSB* (keratolytic winter erythema). This suggests that inherited risk for polygenic disorders characterized by abnormal epidermal differentiation may arise, in part, from dysregulation of target genes whose coding mutation is causal for monogenic diseases that also disrupt this process.

In addition to differentiation genes, immune system genes comprised the other dominant category of the putative daSNV eGene network (Fig. 2g). The epidermis produces numerous cytokines that act on both keratinocytes and inflammatory cells, a number of which are upregulated and released by the epidermis in response to cutaneous injury[41]. New antibody therapeutics targeting the function of specific cytokines and their receptors have proven effective in a host of complex skin diseases, including psoriasis and atopic dermatitis, and are now in widespread medical use[42,43]. A number of cytokines and receptors impacted by these therapeutics were found in the putative daSNV target gene network, with at least one and - often both - cytokine and receptor produced by keratinocytes themselves. These include *IL23A* (targeted by ustekinumab, guselkumab, risankizumab, tildrakizumab, and mirikizumab in psoriasis), *IL4* and *IL13* (receptor targeted by dupilumab in atopic dermatitis), and *TNFRSF1A* which encodes TNFR1 (anti-TNF agents include infliximab, adalimumab, etanercept, golimumab, and certolizumab in psoriasis). The network also includes cytokine-receptor targets for which therapeutics are being developed, such *IL18R1* and *IL22*. Additional pro-inflammatory genes include interferon network genes *IFNGR2, IFNLR1*, and *IFNAR1* along with TFs, such as *IRF1, STAT2* and *STAT3*. The existence of numerous daSNVs as eQTLs for multiple inflammatory genes is consistent with a potentially pathogenic role for non-coding variants that modulate their expression in epidermal cells and suggests additional therapeutic targets may reside in this gene set.

## CRISPR knockout screen for TFs that modulate epidermal differentiation

CRISPR knockout screens of all known TFs to examine their effects on epidermal progenitor and differentiation gene expression will help identify the spectrum of TFs that modulate epidermal homeostasis, including those TFs active at sites of disease-linked DNA sequence variation. Moreover, the factors associated with highlighted SP/KLF or AP-1/AP-2 DNA motifs might also include understudied TF family members or unrelated motif-binding TFs that have not been previously associated with this process. To address this, CRISPR-flow knockout screening[44] was first performed against a canonical marker of epidermal differentiation expressed in suprabasal keratinocytes, KRT10, based on the premise that those TFs that alter KRT10 expression may impact homeostatic gene expression.

A CRISPR library was generated to target virtually all annotated human TFs[16]: specifically, 1772 genes, along with 372 safe targeting guides (Supplementary Data 3). Primary human keratinocytes from two separate donors were infected with a lentivirus pool containing the TF knock-out library at a low library titer and subjected to calcium-induced differentiation before antibody staining for the KRT10 differentiation marker, FACS sorting then barcode sequencing (Fig. 3a, Supplementary Fig. 7a). Cells were split into high, middle and low KRT10 expression groups (Fig. 3b) with 750,000 unique molecular identifiers (UMIs) per sample (445 per target gene, Supplementary Fig. 7b). UMIs for essential genes were depleted (Supplementary Fig. 7c), consistent with effective Cas9 function in the screen. The top 20 genes most highly depleted were common essential or selective (DepMap score < −0.3), with the exception of NANOG (Supplementary Fig. 7d, Supplementary Data 3). The enrichment of SMAD4[45], PRDM1[46], FOSL1[47], and KLF6[48] is consistent with their known functions in keratinocytes, providing further

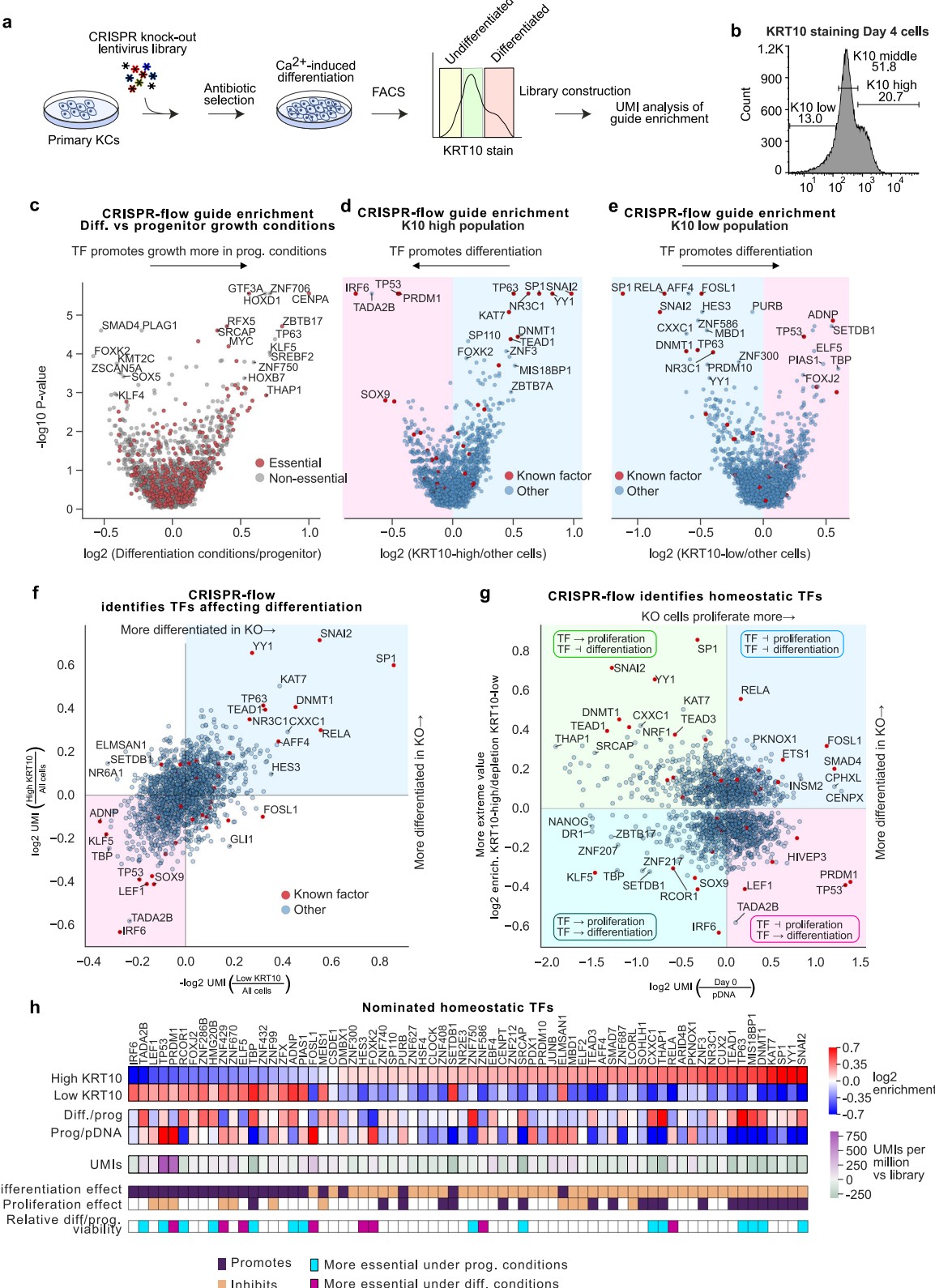

**a** CRISPR knock-out lentivirus library → Primary KCs → Antibiotic selection → Ca²⁺-induced differentiation → FACS → Undifferentiated / Differentiated / KRT10 stain → Library construction → UMI analysis of guide enrichment

**b** KRT10 staining Day 4 cells

**c** CRISPR-flow guide enrichment Diff. vs progenitor growth conditions

**d** CRISPR-flow guide enrichment K10 high population

**e** CRISPR-flow guide enrichment K10 low population

**f** CRISPR-flow identifies TFs affecting differentiation

**g** CRISPR-flow identifies homeostatic TFs

**h** Nominated homeostatic TFs

validation of the screen. Guide counts were also compared between growth and differentiation conditions to identify genes that might affect cells more under one condition (Fig. 3c). As expected, guides targeting genes that are required for cell division were enriched in the differentiation conditions. Also enriched were guides targeting non-essential TF genes, *HOXD1*, *KLF5*, and *RFX5*. KLF5 promotes proliferation downstream of EGFR in basal keratinocytes[49], while

loss of *TP63*, for example, results in cell cycle arrest, explaining their enrichments in differentiated cells[50]. On the other end of the spectrum, the most enriched guide targets in growing cells included those targeting *SMAD4*[45], *FOXK2*[51], *KMT2C*[52], and *KLF4*[53] genes, consistent with known roles in the tissue. Together, these findings are consistent with existing knowledge on keratinocyte biology, supporting the technical validity of the CRISPR-flow screen.

**Fig. 3 | KRT10-based CRISPR-flow TF knockout screen of all TFs identifies known and novel epidermal homeostatic TFs. a** Experimental diagram for a CRISPR-flow screen of all human TFs in primary keratinocytes. **b** Distribution of KRT10 staining intensity across the three sorting groups. **c** Comparison of guide enrichments by MAGeCK for cells grown under progenitor maintenance or differentiation conditions suggests TFs with differential roles across epidermal differentiation. Essential genes (DepMap) are in red. For MAGeCK analysis, the statistical analysis is the robust rank aggregation reported by MAGeCK, which is determined by first a modeling the read counts of a guide by a negative binomial, then ranking the guides by the resulting $P$-values and finally obtaining a gene-specific two-sided $P$ value by permutation test the guide labels. **d** Guides for progenitor maintenance TFs are enriched in KRT10-high sorted cells, while guides for pro-differentiation TFs are depleted. A selection of TFs with known skin homeostasis roles are in red. **e** Guides for differentiation factors are enriched in KRT10-low cells, while progenitor maintenance factors guides are depleted. **f** Correlation between the KRT10-low and high sorted populations reveals outlier effects for known factors IRF6, SNAI2 and SP1. **g** The effect of knock-outs on growth as the enrichment of guides in progenitor cells *vs* plasmid DNA (pDNA), compared with the effect on differentiation, represented by the enrichment (by MAGeCK) of guides in KRT10-high or depletion from KRT10-low cells, whichever absolute value was higher. The interpretation of each quadrant in TF effects is an inset. "Day 0" refers to cells that were not differentiated and instead were grown for four days while the other cells differentiated. pDNA, plasmid DNA. **h** Overview of significant TFs identified by CRISPR-flow. The bottom three rows represent binarizations of effects. Differentiation effects are correspond to whether MAGeCK-determined guide enrichments/depletions in KRT10-high/KRT10-low cells were $P < 0.001$. Binarized proliferation effects denote whether the total UMI counts were in the top or bottom 15%, and relative diff./prog. corresponds to relative depletion between cells seeded for differentiation and cells allowed to keep growing, also indicating the top or bottom 15% of targets.

## Progenitor and differentiation TFs identified by CRISPR-flow

Guides were compared using MageCK[54] between the sorted KRT10 high and low populations, which mark differentiated and undifferentiated cells, respectively (Fig. 3d, e). Guides enriched in the "low" KRT10 population *vs* the "high" KRT10 population are expected to target pro-differentiation factors, as their loss results in an undifferentiated state whereas guides enriched in "high" KRT10 are expected to target genes necessary to sustain the undifferentiated progenitor state. Enrichment in low or high KRT10 populations are different but non-independent hypotheses, so $P$ values for KRT10-high or KRT10-low enrichment were combined into pseudo-P values. Assuming all unexpressed gene targets (which were depleted, $P = 10^{-5}$, Fisher exact, Supplementary Fig. 7e) are false and a range of 100−300 true positives, simulation identified an FDR of 0.24−0.28 for a pseudo-P cutoff of 0.001 (Supplementary Fig. 7f); this yielded 68 TF hits, with a significant enrichment of 15/33 positive control TFs ($P = 7 \times 10^{-14}$, Fisher exact) as a conservative estimate. Plotting enrichment in one population *vs* depletion from the other (Fig. 3f), among the most pro-differentiation TF genes was *IRF6*, consistent with its known dominant role in promoting differentiation[22], while among the most pro-progenitor TF genes was *SNAI2*, consistent with its known essential role in maintaining the progenitor state in keratinocytes[20].

Guides that do not affect growth and division would be expected to be progressively more enriched in the undifferentiated population versus the differentiated population over time, as the undifferentiated population is still capable of growing (Supplementary Fig. 7g). Consistent with this premise, the most significant hits show opposite trends in total UMI counts (Supplementary Fig. 7h). Across all genes, impacts on proliferation and KRT10 differentiation marker expression allowed binning of TFs by their effects on these homeostatic processes (Fig. 3g). TP53 was enriched in the KRT10-low progenitor population (Fig. 3h), likely due to its repressive effect on proliferation. TP63 loss results in cell cycle arrest in keratinocytes[50], explaining its relative depletion from the progenitor pool of cells. On the opposing end of the spectrum, the enrichment of guides for SNAI2, DNMT1[55], TEAD1[21] and YY1[56] is consistent with all these factors being known progenitor maintenance proteins (Fig. 3f, g). SP1 is noteworthy for an extremely large effect on progenitor maintenance relative to its effect on growth (Fig. 3g). This dataset nominates TFs with roles in homeostatic gene expression in epidermal keratinocytes.

## AP-1 and SP/KLF family TFs

Among the most potent TFs regulating the KRT10 differentiation marker, based on MageCK RRA analysis, are known factors, such as RELA, NR3C1, TEAD1, YY1, SNAI2, SP1[47], FOSL1[47], TP63, DNMT1, PRDM1, TP53, IRF6, ADNP, and AFF4[57] (Fig. 3h). The idiosyncratic result of FOSL1 might reflect it being a highly selective gene (DepMap), leading to unusual cell states. Potential novel factors influencing proliferation include CXXC1, ZNF3, SP110, MBD1, and ZNF586, while potential novel factors promoting keratinocyte differentiation include PURB, SETDB1, and TADA2B. Of interest, coding variants in SP110 are associated with atopic dermatitis and SP110 is part of the epidermal anti-viral response[58]. PCA analysis of the 6 significance values corresponding to progenitor *vs* plasmid DNA, differentiated cells *vs* progenitor growth and high *vs* low KRT10 show a main axis of differentiation and a second on comparative growth in progenitor or differentiation conditions (Supplementary Fig. 7i). Together, this screen data reproduced prior results on the roles of known TFs in keratinocyte fate, while also suggesting novel factors.

## Perturb-seq screen for TFs that modulate epidermal homeostasis

The use of a single early differentiation marker, KRT10, may limit the information on TF function provided by a FACS screen. Single cell perturb-sequencing, which can assess expression of a much broader gene set[59], was therefore performed using 1686 guides targeting 402 keratinocyte-expressed TFs (with 78 non-targeting controls), using the same differentiation timecourse as the FACS screen (Fig. 4a, Supplementary Fig. 8a), with undifferentiated progenitor and differentiating cell populations subjected to single cell sequencing. A second perturb-seq screen was conducted afterwards on genes that were hits in CRISPR-flow but not present in the initial perturb-seq screen, bringing the total number of epidermis-expressed TF knockout targets studied by perturb-seq to 502. After initially performing the screen using droplet-based sequencing at Day 3 and 6 timepoints, all 502 TFs were re-screened using split-pool sequencing[60] in a second independent experiment at Day 4 for a total of 274,879 utilized single-guide cell transcriptomes across 6 datasets.

Gene expression maps across these populations matched expectations (Fig. 4b). *KRTDAP, KRT10* and *LCE2A* genes mark general, early and late skin differentiation, respectively. *ITGB1* is a progenitor cell marker. As in the UMAP, basal cells were found at one end of a pseudotime axis, with late differentiated cells at the opposite pole, and the bulk of cells in between (Fig. 4c, d). *KRT10* RNA was not efficiently captured by split-pool sequencing, potentially due to RNA structure, but *GRHL3* was detected as an alternative early differentiation marker. RNA velocity analysis[61,62] at Day 3 was inconclusive but at Day 6 a single axis of cell development was observed that aligned with the SCORPIUS pseudotime trajectory (Supplementary Fig. 8b). Perturbations were successful overall, as cells with a guide targeting a TF had reduced average RNA expression for the target TF, consistent with activated nonsense-mediated decay (Supplementary Fig. 8c). Data from this 502 gene knockout perturb-seq screen enables assessment of TF impacts across a larger set of TF target genes in epidermal homeostasis.

## TF effects on expression of epidermal genes

Differentiation effects were evaluated first based on pseudotime. The strong effects of phenotypic perturbations along a single pseudotime

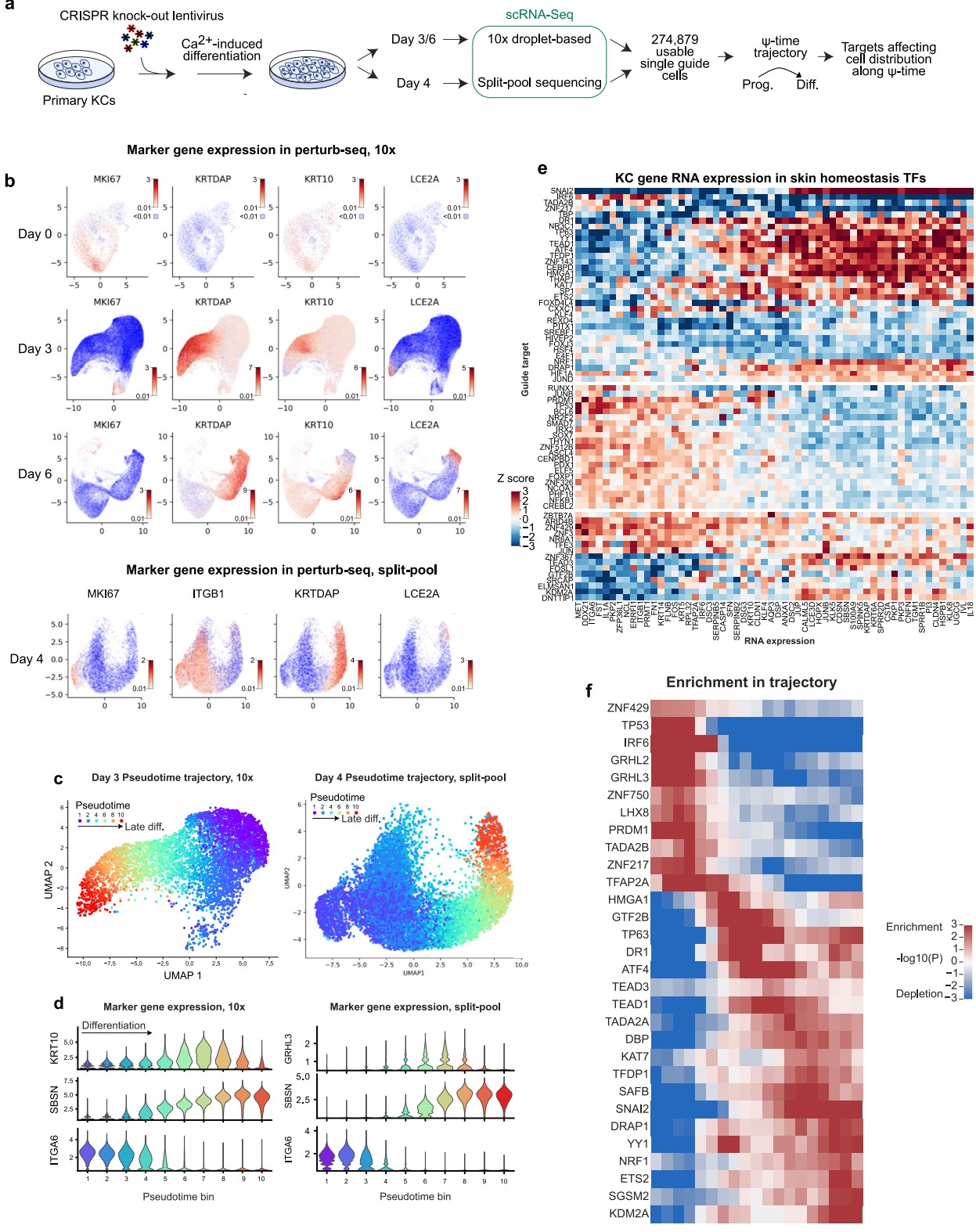

**Fig. 4 | A perturb-seq screen of 502 epidermis-expressed TFs. a** Experimental diagram for the perturb-seq timepoints. **b** Gene expression UMAPs of keratinocyte marker genes for the three timepoints in 10x data (upper panels), and split-pool (lower panels). Split-pool samples (121,301 cells) were randomly subset to 20,000 cells, with further subsetting the "no-expression" cells (blue) to 5000 for clarity. *KRT10* was not efficiently captured by split-pool sequencing. **c** UMAP of Day 3/4 keratinocytes colored by pseudotime, scaled from 1 to 10, in order of increasing differentiation (*left*, 10x data, *right*, split-pool). Cells were randomly subset to 10,000 for clarity. **d** Markers for early differentiation (*KRT10*, *GRHL3*), late

differentiation (*SBSN*) and basal cells (*ITGA6*) across pseudotime bins 1 to 10. **e** The effect of TF knock-outs on RNA expression for abundant RNAs dynamically expressed across differentiation. Colors represent the Z score for expression of the indicated RNA (x-axis) in cells with the given target (y-axis), *vs* other targets. **f** Enrichment of cells with the given guide (row labels) at the indicated position in the pseudotime trajectory (x axis), based on a one-sided Fisher's exact test performed in a sliding window of pseudotime comparing cells with the indicated CRISPR target *vs* all cells. Plotted cells are those with the strongest effect size and FDR < 0.2 for altered pseudotime trajectory, with the addition of TFAP2A.

axis were apparent based on gene expression effects, suggesting a single-axis model to test for differentiation effects (Fig. 4e). FOSL1 was unusual in perturb-seq as well as CRISPR-flow: progenitor RNA expression was decreased, while differentiation RNAs were inconsistently upregulated (Fig. 4e, bottom block). As a selective gene, FOSL1 loss may lead to cells that are neither progenitors nor correctly differentiated. Similar to CRISPR-flow, plotting the average pseudotime for gene knock-outs vs the combined P value for altered pseudotime in individual timepoints, the extremes were defined by the positive control TFs, IRF6 and SNAI2 (Supplementary Fig. 8d, e), further underscoring the potency of these 2 TFs in epidermal keratinocytes. This analysis was extended beyond individual days to determine homeostasis factors by combining samples to increase sensitivity. Plotting the enrichment of KO cells as a function of pseudotime position revealed enrichments in the progenitor population for pro-differentiation factors (IRF6, GRHL3, PRDM1), early differentiation for ATF4 and TP63, and late differentiation for some progenitor maintenance factors (SNAI2, Fig. 4f). Effects on the mean pseudotime of KO cells correlated ($R = 0.37$) between droplet-based sequencing and split-pool (Supplementary Fig. 8f) and were combined into an overall effect on differentiation (Fig. 5a). Perturb-seq overall effect sizes on mean pseudotime values were then correlated with CRISPR-flow effects on cell distribution between high and low KRT10 populations ($R = 0.37$, $P = 10^{-18}$, Fig. 5b), confirming general agreement between methods, with exceptions. Notable among the latter were findings for the ZNF750 TF, where the direction of effect was reversed between KRT10 CRISPR-flow and Perturb-seq. ZNF750 scored as a pro-differentiation factor in perturb-seq but not in CRISPR-flow, however, this is congruent with prior results that ZNF750 drives late differentiation gene induction but in its absence *KRT10* mRNA is stable or increases[23]. The differentiation trajectories for some factors' knock-out cells being enriched for the middle of the pseudotime curves (Fig. 4f) could potentially explain why they might not be identified by CRISPR-flow, while other factors may regulate KRT10 expression without comparably strong effects on overall pseudotime. For example, ATF4 and TEAD1 have enrichments in the middle of the trajectory, suggesting their knockout mediates a loss of the most basal properties without fully engaging the differentiation expression program.

A combined set of 123 putative homeostatic TFs were defined as those with either effects on pseudotime at FDR < 0.2 or significant in CRISPR-flow (Fig. 5c); this list also includes known factor TFAP2A as the only protein with an FDR < 0.01 in the timepoint-enrichment analysis (Fig. 4f) that was not a hit by the other criteria (Supplementary Data 3). This set of TFs encompassed factors previously known to be important in epidermal homeostasis but also nominated new roles for TFs whose impact in this setting has not been previously described. The most enriched TFs based on protein-protein interactions with the 123 homeostatic TFs were MYC, E2F1, TP53, and JUN, followed by additional factors with known roles in skin homeostasis (Supplementary Fig 9a). The most enriched TF networks from the TRRUST database[63] were JUN, YY1, and TP53 (Supplementary Fig. 9b), while the most co-expressed TF was TFAP2A (Supplementary Fig. 9c). A protein-protein interaction network of the hits revealed a large group containing numerous positive controls, linking TP53/63, RUNX1, LEF1, TEAD1/3, SP1, YY1, and AP-1-related factors JUNB, FOSL1, ATF4 and NFE2L2 to putative novel homeostatic TFs, ZNF217, CXXC1 and TADA2B (Fig. 5d). This analysis suggests FOXJ2, CXXC1, and ZNF217 may be novel keratinocyte factors that exert impacts on epidermal homeostasis genes that are among the strongest in the genome. Congruent with MPRA motif analysis (Fig. 5e), AP-1 and SP/KLF family TFs are especially potent, as are other known factors TP63, YY1, RUNX1, TEAD1/3, and SNAI2. These homeostatic TFs represent a pool of candidate regulators that may translate genetic variation to a predisposition for altered epidermal tissue homeostasis.

## TF effects on inflammation

Differentiating keratinocytes arrive at the cutaneous surface armed with an array of pro-inflammatory proteins responsive to tissue injury and infection. The resulting defensive skin barrier against microbial insults requires differentiated keratinocytes as well as surveillance crosstalk with innate and adaptive immunocytes. Aberrant inflammatory signaling is a hallmark of common epithelial diseases characterized by altered epidermal homeostasis. Critical markers for epidermal-immune crosstalk such as neutrophil recruiting chemokines CXCL1/CXCL8[64], TNFα and CCL20 were observed in subpopulations of cells, indicating a range of epidermal inflammatory surveillance states were present (Supplementary Fig. 10a). Many markers were too sporadic in expression to identify TF knock-out effects, so the abundant pro-inflammatory IL1 family cytokines of IL1A, IL1B and IL18 were taken as markers, as they are central to skin homeostasis and disease[65–67]. These IL1 family RNAs were expressed highly dynamically across epidermal progenitor differentiation (Supplementary Fig. 10b, c). RNA velocity analysis identified specific states of differentiation for induction and repression for each cytokine, with the IL1 receptor IL1R1 the last to be induced (Supplementary Fig. 10d). Due to the dynamic expression of IL1 family markers, the effects of TFs on pro-inflammatory cytokine expression were evaluated controlling for pseudotime. There was a slight ($R = -0.18$) negative correlation between pro-differentiation and pro-IL1 family effects (Supplementary Fig. 10e). 22 TFs impacted cytokine production, independent of differentiation (FDR < 0.2, Supplementary Fig. 10f, g), including known factors BCL6[68] and HIF1A[69] (Supplementary Fig. 10g, *right*). ATF4 upregulates IL1A/IL1B/IL18 before late differentiation (Supplementary Fig. 10c, *left*, t-test FDR 0.15), consistent with a report that ATF4 is necessary for the upregulation of IL1A in rat neurons to induce senescence[70], and CUT&RUN in keratinocytes revealed a strong ATF4 peak between IL1A and IL1B that is weakened by differentiation (Supplementary Fig. 10h). CUT&RUN analysis revealed that central keratinocyte inflammation genes are direct targets of TFAP2A and other homeostasis TFs (Supplementary Data S4). These results raise the possibility that modulated cytokine expression might represent a trajectory within differentiation regulated by homeostatic TFs.

## Novel epithelial homeostatic TFs validated by RNA-seq and CUT&RUN

To confirm impacts of potential novel factors identified in the CRISPR-flow and Perturb-seq screens, IRX2, ZNF217, ETV3, NRF1, FOXJ2 and CXXC1 were individually evaluated by siRNA knock-down and RNA-sequencing in progenitor and differentiating keratinocytes (Supplementary Fig. 11a). ETV3 did not satisfy the FDR criteria to be termed a homeostasis gene but was included in this group because it scored significant in prior analysis. Because of the prominence of SP1 in the screens and MPRA motif analysis, SP1 was also knocked-down as a known progenitor TF. SP1, CXXC1, IRX2 and ZNF217 altered expression (FDR < 0.05) of >500 genes, whereas the other TFs affected fewer genes (Fig. 6a, Supplementary Data S5). The screening data categorized SP1 and CXXC1 as essential to maintain undifferentiated progenitor gene expression and prevent differentiation. RNA-seq of SP1 and CXXC1 knock-downs recapitulated this phenotype (Fig. 6b, c), consistent with a new role for CXXC1 in promoting the undifferentiated keratinocyte state and confirming prior findings with SP1[47,71,72]. In contrast, IRX2, FOXJ2, NRF1 and ZNF217 depletion impaired normal differentiation gene induction (Fig. 6d–f), also consistent with the screening results; however, ETV3 did not and was thus not confirmed as a TF affecting the expression of epidermal growth and differentiation genes. IRX2, FOXJ2, NRF1 and ZNF217 were each required for induction of genes involved in the production of the epidermal cornified envelope, underscoring potential regulation of key genes involved in cutaneous barrier formation (Fig. 6f, Supplementary Fig. 11b). A progenitor maintenance role for IRX2 is also tentatively

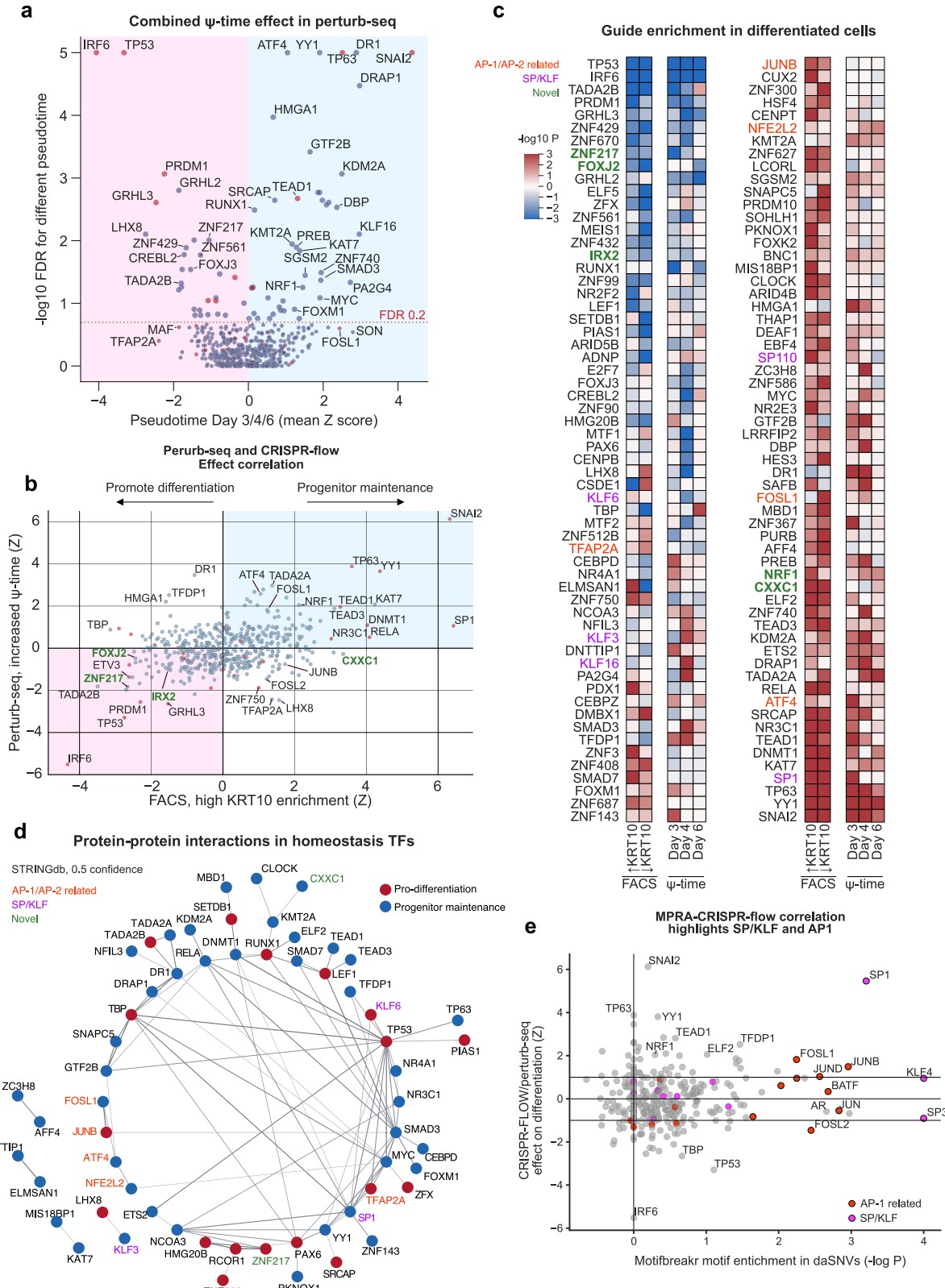

**Fig. 5 | Identification of keratinocyte homeostasis transcription factors.**
**a** Combined differentiation effect in perturb-seq based on altered pseudotime, including both sequencing technologies and all three time-points. Factors may have significant FDR values despite small effects if the Z scores average to near zero but one sample is highly significant. Red dots, known factors. **b** Correlation of differentiation effects between CRISPR-flow (as mean change to the KRT10 High/Low distribution) and perturb-seq (as changes to mean pseudotime), both as Z-scored effect sizes, identifies the most potent TFs as outliers and highlights potential novel TFs (green labels). **c** Guide enrichment in differentiated cells of the strongest TFs (by significance) in CRISPR-flow (calculated as in panel 3c) and perturb-seq (one-sided Mann–Whitney for altered pseudotime vs all other cells with one guide). Depletions from differentiated cells reversed the sign of the −log *P*-value for visualization of the direction of the effect. **d** Physical STRINGdb protein-protein interactions (confidence 0.5) among the homeostasis TFs link known and novel factors, such as ZNF217 to RCOR1. **e** Correlation of broken motif enrichment in MPRA (Motifbreakr motif assignments, Fisher's exact test, two-sided) with averaged CRISPR-flow/perturb-seq effects (Z-scored effect size, taking the larger value between methods) on differentiation highlight AP-1 (red dots) and SP1.

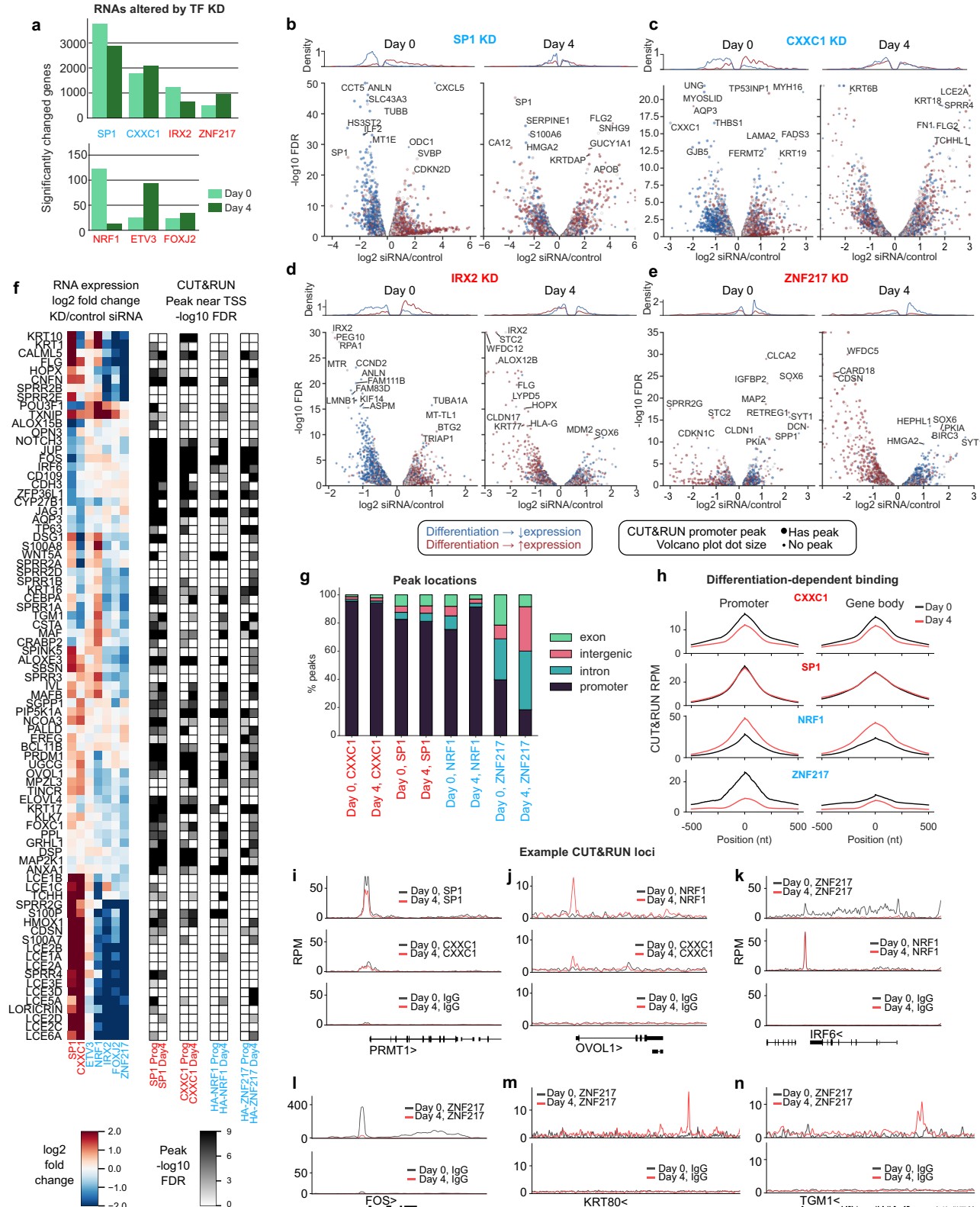

**Fig. 6 | Validation of novel homeostatic epidermal TFs. a** Numbers of significantly changed epidermal genes for novel homeostasis factors and SP1 upon siRNA knock-down (FDR < 0.05, DESeq2). **b–e** RNA expression effects of siRNA knock-downs in progenitor or Day 4 differentiated cells by RNA-seq (DESeq2). RNAs induced/repressed by differentiation are those with 2-fold changed expression across differentiation in WT cells. Only RNAs with |log2| > 0.2 fold changes upon siRNA treatment are plotted. CUT&RUN peaks (MACS2 FDR < 0.1) are assigned to the nearest transcription start site. **f** Effects on RNA levels of differentiation-associated RNAs in Day 4 differentiated cells with the indicated TF siRNA knock-downs (*left panel*), and whether the indicated factors have a CUT&RUN peak near the TSS (*right panel*). **g** The location of peaks in CUT&RUN for the indicated novel TFs and SP1. **h** Mean reads-per-million (RPM) for peaks in promoters or gene bodies. **i–n** Example binding patterns from CUT&RUN at *PRMT1*, *OVOL1*, *IRF6*, *FOS*, *KRT80*, and *TGM1* gene loci.

suggested by Fig. 6d. These data confirm new essential roles for CXXC1 in sustaining progenitor gene expression and for the IRX2, FOXJ2, NRF1 and ZNF217 TFs in epidermal differentiation gene induction.

Knowing the genomic binding sites of TFs with newly identified roles in regulating epidermal growth and differentiation genes may help nominate their target genes and focus studies of how regulatory sequence variants might alter interactions with DNA. CUT&RUN was therefore performed for TFs with newly identified homeostatic roles, yielding good quality data for CXXC1, NRF1, and ZNF217, each of which generated >10,000 peaks, as did the SP1 benchmark TF (Supplementary Fig. 11c). Unbiased de novo motif finding by DREME[73] identified the expected SP1 and NRF1 motifs, a GGCGG motif for CXXC1 consistent with its preference for CGG in promoters[74], and AT-rich sequences for ZNF217, which has no published consensus specificity (Supplementary Fig. 11d). Novel TFs demonstrated distinct binding patterns across differentiation. For example, CXXC1 and SP1 peaks were concentrated in promoters in contrast to ZNF217, which bound more intergenic and intronic regions (Fig. 6g). Moreover, ZNF217 significantly decreased promoter binding over the course of differentiation, while the opposite trend was observed for NRF1 (Fig. 6g, Supplementary Fig. 11e, f). Genome binding dynamics (Fig. 6h) included modest quantitative decreases in TSS association for CXXC1 (Fig. 6i), movement of NRF1 to TSS sequences (Fig. 6j), and multiple patterns for ZNF217. CXXC1 binds strongly to the *YWHAZ* gene promoter and CXXC1 loss upregulated *YWHAZ* in differentiated cells and represses it in progenitors (Supplementary Fig. 11g). ZNF217 was associated with gene bodies more than other factors (Fig. 6g) and displayed a range of binding alterations, including a loss of gene-body association (as for *IRF6* or *YWHAZ*, Fig. 6k, Supplementary Fig. 11g), a loss of promoter binding (Fig. 6l), and a less common gain of promoter binding (seen at *TGM1* and *KRT80*, Fig. 6m, n, which fail to be fully induced during differentiation in the absence of ZNF217). These binding patterns support a model in which ZNF217 has both TSS and gene-body functions, with promoter functions repressed during differentiation. All factors show significant overlaps between TSS peaks and expression changes with knock-downs, with the strengths of association suggesting the TFs are usually activating except for CXXC1 in differentiated cells and ZNF217 in progenitor cells (Supplementary Fig. 11h). Together, these results display newly identified roles for these TFs and distinct patterns of genomic binding during differentiation.

### Global occupancy patterns of 27 homeostatic TFs across epidermal differentiation

MPRA data and TF knockout screens suggested that specific TFs and TF families, among them AP-1/2 and SP/KLF, may have outsized impacts on expression changes linked to skin-disease associated SNVs and on skin homeostasis. However, TF motif analysis alone, and even binding data from other cell types, is an unreliable indicator as to whether a specific DNA sequence is bound in a given tissue cell type[35]. A more detailed understanding requires determining the relevant TF binding patterns in relevant cell contexts, namely keratinocytes undergoing the transition from undifferentiated progenitor-containing populations to terminally differentiated cells. Such data may help determine which SNV sequences are bound by specific TFs, nominate the genes those TFs regulate, and determine if a SNV modifies TF binding to DNA of interest. Together such information may propose mechanistic models of genetic risk and help address whether disease-linked daSNVs alter the interactions of essential epidermal TFs with disease-relevant target DNA.

To map the DNA binding regions of homeostatic TFs during keratinocyte differentiation and to begin to explore potential TF associations with regulatory DNA variants, CUT&RUN was therefore performed for 27 TFs plus the H3K27me3 control histone mark in 298 samples, from 13 diverse individuals across differentiation. Only TFs essential for homeostatic epidermal gene expression were studied, with priority given to those factors with motifs indicated in MPRA as

having potent transcriptional effects. Genomic binding was studied in both undifferentiated primary keratinocyte progenitor populations and in differentiating cells (Fig. 7a, b, Supplementary Fig. 12a, b). This generated a resource dataset of TF binding profiles across cellular differentiation in which many TFs are assayed across unrelated individuals to identify individual-specific binding effects at scale. The 298 samples represented a subset of CUT&RUN datasets that passed quality control metrics from a larger group of >500 attempted samples (Supplementary Data 6) and were characterized by a PCA cluster distinct from the 60 IgG controls generated in parallel, as well as (with a few exceptions) by the enrichment of a cognate TF motif. Only three factors had existing data in human keratinocytes across differentiation (TP63, SNAI2, KLF4, see Supplementary Data 6 for prior datasets), while 15 factors had not been previously assayed in human keratinocytes before by ChIP-seq in any state, including well-established factors such as OVOL1, TFAP2A, YY1, CREB1 and SP1. The resulting datasets, available online at UCSC, provide a resource of genomic binding sites for homeostatic TFs in undifferentiated and differentiating human epidermal keratinocytes.

### Genome binding dynamics of homeostatic TFs

CUT&RUN replicates correlated well (mean $R = 0.77$, Fig. 7b, Supplementary Fig. 12c), with enhancer-associated factors, such as AP-1, distinct from promoter-associated factors, such as the SP/KLF family on the opposite extreme of the correlation matrix (Fig. 7b, PCA in Supplementary Fig. 12d, e). Between these extremes were factors more evenly divided between promoters and enhancers and a section of low-signal samples, such as SNAI2 after differentiation, or OVOL1 in progenitors, matching the periods of inactivity for these TFs. TF binding regions were strongly enriched for their cognate motif with a few exceptions, and numerous examples of multiple motif enrichments occurred (Fig. 7c), such as ATF4/CREB1 at different versions of the AP-1 motif. Investigations of co-binding suggested the possibility that ATF4 and JUNB might interact over differentiation (Supplementary Fig. 13). Most factors bound just upstream of the TSS in both timepoints, or had low levels of TSS association, but CXXC1 exhibited a unique pattern of frequent binding in a one-nucleosome-width window just downstream of the TSS (Fig. 7d). Among TFs studied, CXXC1 was again observed to bind almost exclusively at promoters, followed by NRF1, SP3, and SP1 (Fig. 7e). Factors that mostly avoid TSSs, and found in both intergenic regions and gene bodies, include TEAD1, JUNB, TP63, and JUN. In the middle of this range are proteins that likely function frequently at both enhancers and promoters, including TFAP2A, KLF4, ATF4 and RUNX1. The factors with the largest changes in peak distribution between these categories during differentiation were NRF1, KLF4 and ZNF217 (Fig. 7f). Overall, there was a tendency for the fraction of peaks in promoters to decrease across differentiation, consistent with the transcriptional dampening seen in MPRA data in differentiating cells.

### TF binding to monogenic disease genes and skin disease-linked SNVs

Consistent with their observed impacts on epidermal gene regulation, most homeostatic TFs had hundreds of peaks at or within 50 kbp of genes with keratinocyte homeostasis and differentiation GO terms (Fig. 7g, Supplementary Fig. 14a), at a statistical enrichment relative to expressed genes (Supplementary Fig. 14b, c). A similar TF adjacency pattern was observed for genes whose protein coding mutations cause monogenic human epidermal diseases (Supplementary Fig. 14c, d). Most of these genes are associated with epidermal differentiation, with a large number comprised of cutaneous barrier factors. The distribution of TF binding across the promoters of such genes was unequal, and binding hotspots for specific homeostatic TFs were observed in some gene promoters (e.g., *JUP, SFN, KRT14*). Such promoters tended to be targets of both

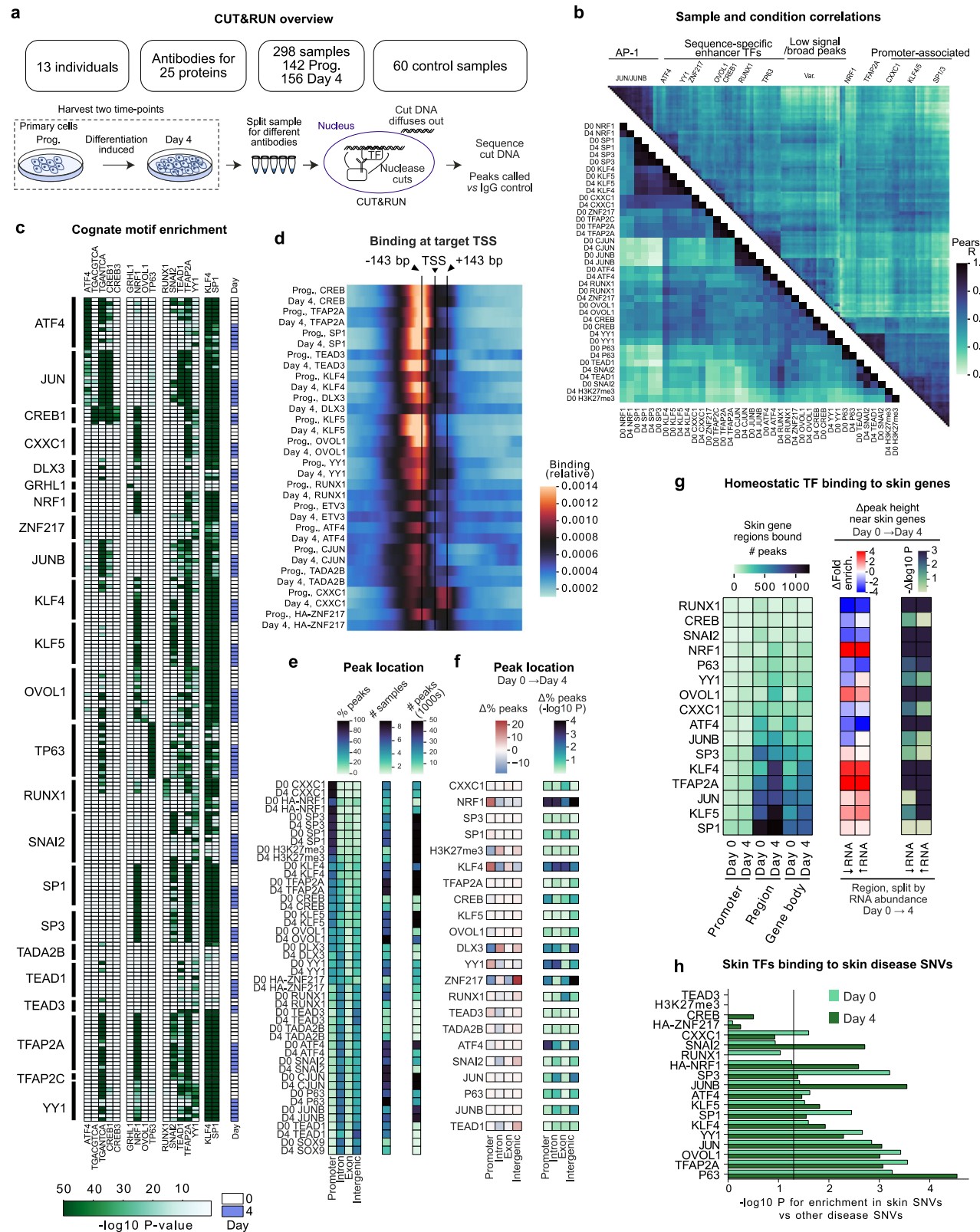

AP-1/2 TFs and multiple SP/KLF proteins (Supplementary Fig. 14a). Some TFs increased association with differentiation genes across the time course (NRF1, OVOL1, and KLF4), while others decreased association, such as SNAI2 (Fig. 7g). Interestingly, homeostatic TFs also bound at skin disease-linked SNVs more than SNVs linked to risk for polygenic diseases in non-cutaneous tissues (Fig. 7h). This enrichment was highest for the essential stratified epithelial TF,

TP63, followed by AP-1, TFAP2A, OVOL1, SP1/3 and YY1. This database of TF binding across differentiation reveals homeostatic TFs display enriched binding near genes involved in differentiation, including those whose protein coding mutation causes monogenic skin diseases. Further, these data also indicate that homeostatic epidermal TFs are more associated with epidermal disease-linked variants than they are with other disease variants, consistent with

**Fig. 7 | CUT&RUN of homeostatic epidermal TFs. a** Overview of CUT&RUN experiments. **b** Correlation between samples (*top triangle*) or conditions (*bottom triangle*) in CUT&RUN data for 25 proteins shows correlation groups of SP/KLF/ AP2A/CXXC1/NRF1, AP-1, and ATF4/RUNX1. A randomly selected 200 of the top 500 peaks for each condition were merged into a union peak set, and the maximum peak height correlated between conditions or samples. **c** TF CUT&RUN data shows enrichment of cognate motifs. For (**b, c**), a subset of datasets is included to facilitate visualization. Motif enrichment is calculated by AME (MEME suite) from JASPAR motifs using FDR < 0.1 peaks. AME calculated motif enrichment relative to random sequence by classifying them as having the motif or not, then applying a one-tailed Fisher's exact test. **d** Averaged binding at target TSS across differentiation shows a subset of proteins (e.g., TEAD3, ATF4, RUNX1, KLF4) change their average TSS association, and CXXC1 exhibits an unusual pattern of binding approximately one nucleosome after the TSS. **e** Peak location, peak number, and the number of

assayed samples for each TF. **f** Changes in peak location across differentiation as % change (*left*) and significance (2-sided t-test, *right*) show NRF1, KLF4 and ZNF217 have the largest changes. **g** Homeostatic TFs binding at epithelial genes, in some cases dynamically across differentiation. "Region", 50 kbp region on either side of the gene bodies. ↑↓RNA regions represent subsetting the combined set of monogenic disease genes and skin genes by those that either increase or decrease in expression across differentiation. ΔFold enrich., the mean change in MACS2 fold enrichment for peaks in the given region. The log10 P value represents a two-sided Mann-Whitney for a change in the fold enrichment across differentiation. **h** Skin disease linked SNVs are more likely to be bound by homeostatic TFs than SNVs linked to other diseases, especially for TP63, TFAP2A, OVOL1, JUN/JUNB, SP1 and KLF4. P values are unadjusted one-sided t-tests for the number of skin disease SNVs in peaks (by replicate) vs other diseases (see Methods).

the possibility that such noncoding variation may confer disease risk, at least in part, by altering TF associations with regulatory DNA.

**Population sampling CUT&RUN identifies allele specific binding**

CUT&RUN samples were evaluated for allele specific binding events (ASBs) in an approach here termed "population sampling CUT&RUN" (Fig. 8a). After removing one set of samples that showed evidence of contamination by a second person, the remaining individuals could be tentatively assigned possible genetic backgrounds that include Asian, African, and European based on the alleles within the CUT&RUN data itself[75]. ASBs were determined by BCF tools variant calling[76], depth filtering, removal of homozygous alleles, filtering problematic genomic regions, binomial test determination of altered binding[77], and the removal of events that were significant in IgG controls, which likely reflected technical artifacts and copy number variations and, finally, mapping simulated protective/risk-linked SNV allele reads to the genome and filtering variants with mapping biases[78] (Fig. 8a). More simply, allele-specific differences were probed by CUT&RUN by identifying heterozygous sites that were not bound in a 1:1 ratio, and for which such imbalances could not be easily explained by technical biases. Because these alleles are drawn randomly from the population, they overlap with the SNVs tested by MPRA only by chance (discussed below). ASBs were only evaluated in individuals heterozygous for the SNVs studied. Multiple categories of ASBs were defined (Fig. 8b). *Putative ASBs* were all those with an uncorrected P value below 0.1 and fold change >20%, *likely ASBs* were *putative* ASBs with disruptions to the cognate TF motif consistent with the directionality of altered binding, and *high confidence ASBs* had an FDR < 0.2 and motif concordance, namely, reduction of TF binding by SNVs that disrupt the TF DNA binding motif. In total, there were 1819 FDR < 0.2 ASB SNVs and 298 high-confidence ASB SNVs. JUN, SP3 and TFAP2A had the greatest number of ASBs (Fig. 8c). Similar to previous studies[27], concordance between motif changes and the direction of altered binding were ~3–8-fold enriched concordant:discordant for sequence specific enhancer-associated TFs (Fig. 8d), even at the unadjusted P < 0.1 cutoff. This indicates that the current data may have similar false positive rates to previous Bayesian approaches[27]. For example, at FDR < 0.2, JUN had 65 concordant ASBs and 1 discordant ASB, indicating both high data reliability and highly motif-dependent binding. Promoter-associated, C-rich sequence binding SP/KLF family proteins, however, showed less enrichment of concordance, especially for SP1. The SP3 motif in JASPAR is longer than the SP1 motif and matches some alternative versions of the SP1 motif; the lower concordance for SP1 than SP3 is likely at least partially a result of comparison with a less accurate motif. Lower SP/KLF concordance more broadly may reflect SP/KLF association in promoters in a fashion that is partially independent of the pure GC sequence with the highest in vitro affinity. Together these observations suggest homeostatic TFs have numerous ASBs, and motivate future efforts for their further characterization.

**TF, SNV location, and allele frequency relationships to ASBs**

Several factors affected ASB frequency and effect. Overall, heterozygous sites in promoters displayed less change in TF binding ($P = 10^{-20}$ Mann–Whitney, or $P < 0.001$ ANOVA, controlling for read depth and protein, Fig. 8e), and were less likely to be ASBs than binding sites in intergenic regions representing putative enhancers ($P = 7 \times 10^{-4}$ Mann–Whitney, or $P < 0.001$ ANOVA, controlling for read depth and protein, Fig. 8f). This trend was notable ($P < 0.01$) for TFAP2A and SP/ KLF, and generally observed across TFs (Supplementary Fig. 15a). This suggests a possible "buffering" effect against DNA sequence variation impacting TF binding within gene promoters. The effect was independent of read depth (Supplementary Fig. 15b). Exons also had significant buffering effects *vs* intergenic sites, but at reduced effect (Supplementary Fig. 15c, $P = 0.02$, t-test). For putative ASBs, there was a very small positive correlation between the frequency of an allele in the population and its binding relative to variants ($R = 0.05$, Supplementary Fig. 15d), as observed in a recent study in brain tissue[79], which was slightly significant ($P = 0.03$, ANOVA). Taken together, these data suggest that promoter regions show evidence "buffering" effects, wherein TF-DNA binding interactions are stabilized against genetic variation.

**Promoter buffering may be related to high TF occupancy**

What might cause the buffering of promoter SNVs? There is increased motif clustering in promoter SNVs for AP-2 and SP/KLF, which could help explain increased resistance to individual mutations (Supplementary Fig. 15e). The information content of modified bases was not different in promoter SNVs, suggesting the effect is not caused by more degenerate bases (Supplementary Fig. 15f). For AP-1 and TP63, cognate motifs were depleted from bound promoter SNV regions *vs* intergenic regions, resulting in decreased mutation frequency, but this was not true for other TFs (Supplementary Fig. 15g–i). The decreased frequency of motif alteration for AP-1/TP63 could help explain their buffering, while the increased motif density could provide the buffering effect for AP-2/SP/KLF factors. Both factors could fit under a general model of increased TF occupancy, as AP-1/TP63 in promoters *sans* motif might be recruited by other TFs. Motivated by this reasoning, the correlation between the number of TFs bound at a promoter and the magnitude of ASB was investigated. Bound SNVs in high-occupied TF target, or "HOT", regions (regions ~10 kbp with many TFs bound[80]) had reduced ASB (Supplementary Fig. 16a) and more TFs bound at a promoter correlated with decreased ASB fold change, especially for KLF4/5 (Supplementary Fig. 16b), suggesting a "peer pressure" model in which higher TF binding density at promoters might be a major factor in resisting mutational effects.

**ASBs and epidermal homeostasis**

Like TF binding peaks, ASBs were enriched near genes involved in epidermal homeostasis (Fig. 8g, Supplementary Fig. 17a, empirical $P < 0.01$ for all ASBs and $P = 0.006$ for the average homeostatic TF

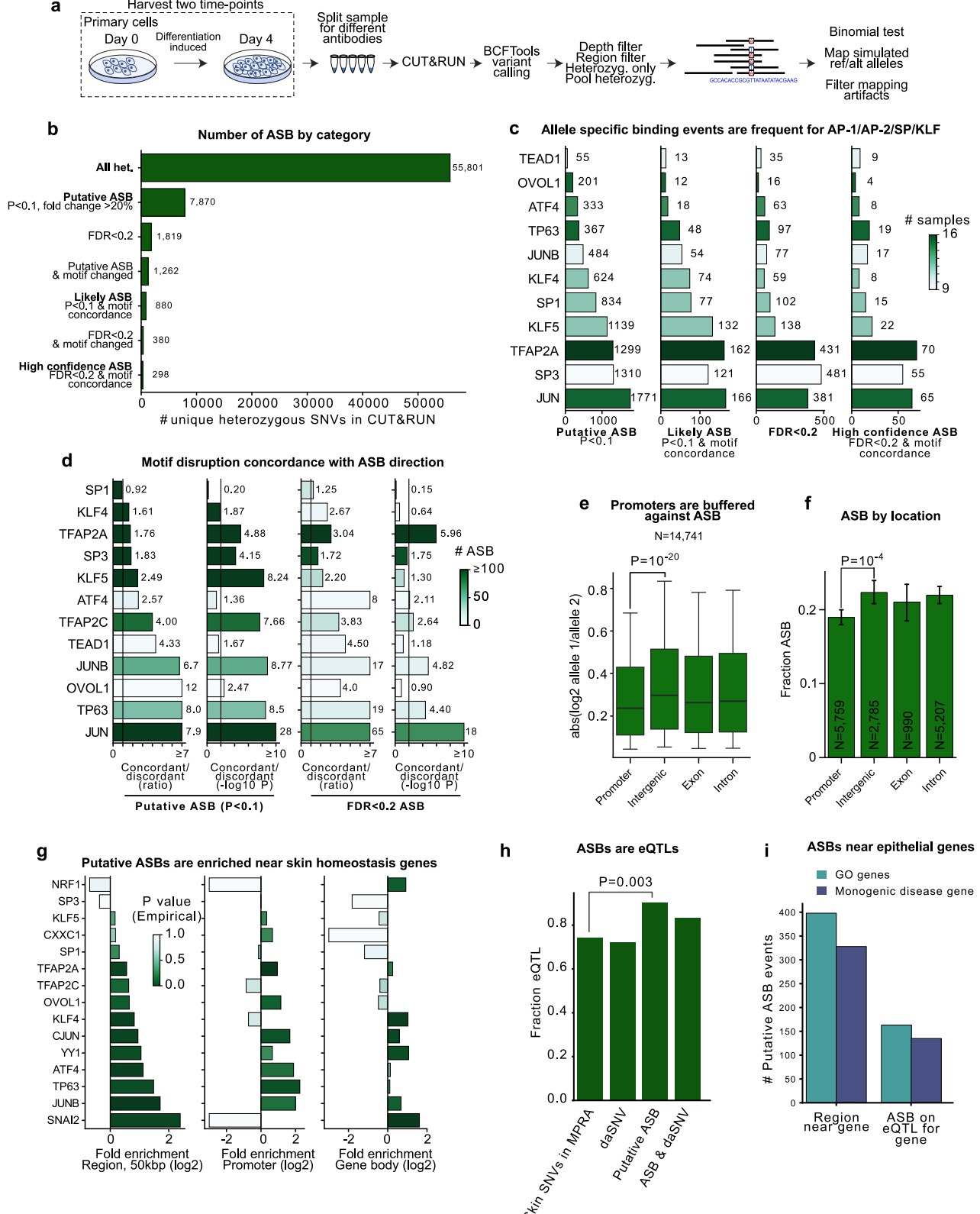

having an ASB enrichment, t-test). ASBs were also enriched near monogenic disease-linked genes (Supplementary Fig. 17a, b, empirical $P < 0.01$ and $P = 0.001$, t-test). SNAI2, TP63, AP-1 and AP-2 TFs were the most enriched at these genes, followed by YY1 and OVOL1 and KLF4. Interestingly, among skin disease-linked SNVs tested in MPRA, ASBs were more likely to be eQTLs than non-ASBs ($P = 0.003$, Fisher's exact, two-sided), and more likely to be eQTLs

than daSNVs were (Fig. 8h), suggesting the population sampling CUT&RUN method may highlight functional disease-linked SNVs. Altogether in this catalog, there were 398 ASB events near homeostasis genes and 135 ASB events were on eQTLs for monogenic epithelial disease genes, suggesting numerous opportunities for noncoding variation to affect disease risk through altered TF binding (Fig. 8i).

**Fig. 8 | Allele specific binding of epidermal homeostatic TFs. a** Experimental diagram for identifying ASB from CUT&RUN data. **b** Number of SNVs in each category of ASB. The number of reads from each allele in a heterozygous site are tested by a two-sided binomial assuming equal probability of binding between alleles. FDR was estimated by Benjamini-Hochberg. **c** Numbers of putative ASBs by indicated ASB category for TFs studied. **d** Concordance of predicted motif strength change and the direction of altered binding; note good agreement for sequence-specific enhancer-binding proteins, but lower agreement for SP/KLF factors that mostly associate with common C-rich sequences in promoters. **e** Absolute log2 fold change in reads at ASBs as a function of genomic location. Whiskers, 10–90% data range. *P* value, two-sided Mann–Whitney. Center line, median. Bounds of box,

25–75% data range. Outliers are not drawn due to the high number of datapoints. "N" is the number of bound heterozygous sites. **f** Bar height is the fraction of heterozygous SNV-protein pairs with at least 20 CUT&RUN reads that are putative (*P* < 0.1) ASBs as a function of genomic location. *P* value, two-sided Mann-Whitney. Whiskers, 95% CI. "N" is the number of bound heterozygous sites. **g** Putative ASBs are enriched near skin disease genes relative to expressed genes (*P* value test for greater enrichment vs random expressed gene selection was determined by taking random subsets of expressed genes and determining what fraction of random selections exceeded the observed number). **h** 90% of ASBs in skin disease-linked SNVs are eQTLs. *P* value, two-sided Fisher's exact. **i** Number of ASB events near epithelial homeostasis genes in the catalog.

## ASBs and transcriptional effects

JUN ASBs correlated with MPRA allelic effects on transcription (*R* = 0.38, *P* = 0.001 t-test), and the SP/KLF family also correlated, albeit less well (*R* = 0.2, *P* = 0.009, t-test, Fig. 9a, Supplementary Fig. 18a). These trends suggest the allelic binding effects observed for these proteins at skin disease-linked variants are relevant to allelic transcriptional effects and vice versa. A correlation was not observed for all proteins (e.g., TFAP2A, *R* = −0.1, *P* = 0.3), suggesting that, while transcriptional and binding effects may be related for a subset of TFs, they are not invariably correlated, especially for proteins other than AP-1/ SP/KLF family members. Outside promoters, daSNVs had a higher altered binding fold change than non-daSNVs (Fig. 9b); although the effect is borderline significant (*P* = 0.05, t-test, two-sided), the trend aligns with MPRA and ASB providing related but distinct read-outs of SNV effects. Relatively few daSNVs were "validated" by an ASB event, but this is mostly explained by few being observed bound in a heterozygous individual (Supplementary Fig. 18b). Interestingly, although we expected risk alleles to have a bias to decreased binding, this trend was not significant in ASBs, nor was there decreased transcriptional activity in MPRA (Fig. 9c); both methods therefore suggest non-coding risk alleles are not dramatically biased to loss-of-binding. Alleles that were both putative ASBs and daSNVs in MPRA were nearly all comprised of ASBs for AP-1/2 or SP/KLF TF proteins, suggesting these TF families may have a role in "reading" non-coding disease linked variants into altered transcription (Fig. 9d).

## ASB vignettes

To extend these observations, ASBs were filtered to those linked to skin disease and further filtered to those associated with monogenic skin diseases. This results in numerous examples of potentially functional ASBs (Supplementary Data 7). For example, the rs4704864 disease-risk daSNV is an eQTL for the *NIPAL4* gene whose coding mutation disrupts normal epidermal differentiation and leads to a generalized skin disease, namely autosomal recessive ichthyosis[81]. This daSNV disrupts a perfect AP-1 motif within an intron of the *NIPAL4* gene that loops to its promoter; the disease risk variant causes reduced transcriptional activity in MPRA (Fig. 9e, f). Gene editing was performed at this site to produce isogenic populations of human keratinocytes that differ only at the SNV of interest. The disease daSNV decreased *NIPAL4* the expression, but not expression of the adjacent *THG1L* gene, indicating that this daSNV modulates *NIPAL4* levels; rs4704864 sequence is bound strongly by JUN, with the risk allele less bound (*P* = 0.09, two-sided binomial, Fig. 9g, h, Supplementary Fig. 18c). A plausible model posits that disruption of JUN binding by the rs4704864 daSNV risk variant results in reduced transcription of *NIPAL4*, which in turn decreases the integrity of the cutaneous barrier, potentially predisposing to cutaneous inflammation.

Another example of a daSNV that alters binding of the JUN TF is the psoriasis SNV rs10217259, which is looped to the *KLF4* gene promoter (Fig. 9i–k). In this case, the alternate allele is more bound by JUN and results in more transcriptional activity on MRPA. Potentially, this daSNV may increase JUN binding and over-express *KLF4* in a fashion that dysregulates epidermal differentiation, as physiologic *KLF4*

expression is essential TF for this process[23,53]. An additional example, which highlights a difference between MPRA and population sampling CUT&RUN, is the rs1264326 daSNV, which has been linked by GWAS to risk for inflammatory acne rosacea. rs1264326, which is located just upstream of the promoter for the pro-inflammatory *DDR1* gene[82,83], is an eQTL for *DDR1* as well as the nearby *PSORS1C1* inflammation-linked gene (Fig. 9l). The SNV is bound by TFAP2A and SP3, but is an ASB only for TFAP2A, and MPRA suggests potentially decreased variance with the risk allele, without a clear change in average expression (Fig. 9m, n). However, the risk allele substantially enhances TFAP2A DNA binding, consistent with a model in which the rs1264326 daSNV enhances TFAP2A promoter binding to alter expression of *DDR1* and *PSORS1C1* pro-inflammatory genes and drive risk for acne rosacea. Identified daSNVs that alter TF binding and transcription-directing activity provide potential paradigms for how regulatory DNA variants may contribute to disease risk and provide a resource for future studies of molecular mechanisms in polygenic skin disease.

## Discussion

Here we examine the interplay between disease-linked DNA variation and TFs in somatic progenitor differentiation in epidermis. The activity of regulatory DNA variants linked to prevalent human disorders of epidermal tissue homeostasis was assessed, demonstrating that disease variants manifest their functional impacts depending on progenitor differentiation. The TFs essential for epidermal progenitor maintenance and differentiation were identified, uncovering a number of potent new TF roles in the process. Finally, integration of DNA variants with essential homeostatic TFs was pursued through genome-wide TF binding data in a population of individuals to uncover ASBs for key TFs linked to target genes of interest. The present data provide support for a provisional model that may apply as a potential pathogenic mechanism for a portion of noncoding disease-linked SNVs (Fig. 9o, p). These data compendia comprise a resource for further studies of the pathogenesis of polygenic skin diseases and are included a web resource (https://arvid.stanford.edu), comprising a provisional atlas for regulatory TFs and sequence variants in disorders that impact epidermal homeostasis. Taken together, these findings provide a resource for understanding how DNA variants interact with specific TFs across progenitor differentiation to modulate genes important in epidermal progenitor cell maintenance and differentiation.

Previously, we surveyed the grammar of TF motifs in keratinocytes by MPRA but only evaluated 11 GWAS SNVs[4]. We here identify 355 differentially active SNVs (daSNVs) in regulatory DNA linked by GWAS to risk for prevalent inflammatory and neoplastic human skin diseases were identified. Putative daSNV target genes include those involved in epidermal differentiation and inflammation, highlighting the potential roles that dysregulation of these biological processes may play in genetic predisposition to the common polygenic skin diseases studied here. Knockout of 1772 TFs in KRT10 CRISPR-flow followed by single-cell perturb-seq of 502 of these TFs nominated 123 TFs with non-redundant impacts on epidermal gene regulation, including new roles for ZNF217, CXXC1, FOXJ2, IRX2 and NRF1, and additionally 22 TFs implicated in

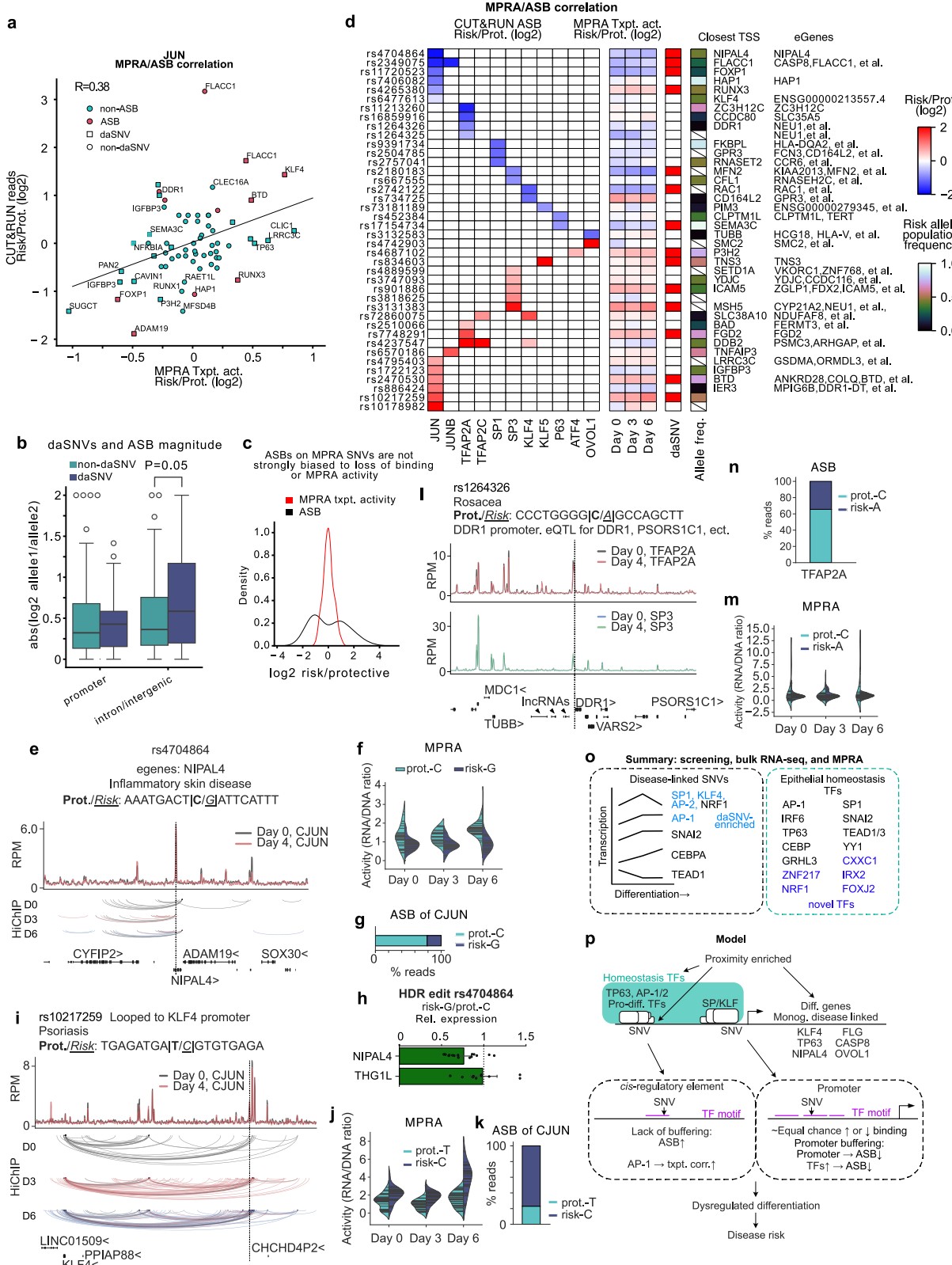

inflammatory signaling. CUT&RUN performed on 27 homeostatic TFs in undifferentiated and differentiating keratinocytes from 12 individuals identified dynamic TF genomic binding enriched around epidermal differentiation genes, including those whose coding mutation causes monogenic skin disorders. SP/KLF and AP-1/2 family factors were prominently represented in daSNV motif impacts, essential homeostatic TF functions, and differentiation

gene binding enrichments, suggesting these TFs may play a role in translating inherited noncoding genetic variation into gene dysregulation.

Epidermal differentiation is broadly disrupted in the disorders studied here, including psoriasis, atopic dermatitis, BCC, SCC, and even dermatomyositis and cutaneous lupus erythematosus[84–88]. It has been unclear, however, whether altered epidermal homeostasis is a

**Fig. 9 | Allele specific binding events and transcriptional activity. a** For skin disease associated SNVs, there is a positive correlation in JUN CUT&RUN reads for risk vs protective alleles with transcriptional activity from the same alleles. **b** ASB fold changes may be greater on daSNVs. ASBs were categorized by whether they were daSNVs in MPRA, and by genomic location. The *P* value is from a two-sided t-test. Outlier values were clipped to 2 for visualization. Whiskers, max value within 1.5 interquartile ranges. Center line, median. **c** ASB and transcriptional changes are not significantly decreased on risk alleles. **d** Overlap of putative ASB with skin disease associated MPRA shows potential sites of ASB-driven changes in transcriptional activity at homeostasis genes. Allele freq., frequency of risk allele (crossed white box if the RSID was not found). daSNV, FDR < 0.01. **e** rs4704864 is an eQTL for the monogenic skin disease gene *NIPAL4* and breaks a JUN motif. **f** The risk allele results in reduced transcriptional activity in MPRA (FDR 0.0009), **g** JUN binding in CUT&RUN (FDR 0.0009), **h** *NIPAL4* RNA expression by qPCR after genomic edits. Cas9-AAV-based homology directed repair gene editing (HDR). *THG1L* is a nearby control gene. Whiskers, 95% CI. **i–k** The risk allele of rs10217259

results in increased transcriptional activity (FDR 0.004) and increased JUN binding (FDR 0.18). rs10217259 is looped to the KLF4 promoter and is associated with psoriasis. **l–n** rs1264326 is associated with rosacea and located adjacent to the *DDR1* gene promoter; the risk allele exhibited more transcriptional variability and TFAP2A binding (FDR 0.02), although the risk allele was not significantly increased or decreased in MPRA activity. **o** Summary of screening, bulk RNA-seq, and MPRA results. **p** Provisional model of one potential mechanism of transcriptional control and disease-linked noncoding variant function in epidermal cells with the following features. Homeostatic TFs bind (and exhibit ASBs) near differentiation and monogenic disease-associated genes, including *KLF4*, *TP63* and *NIPAL4*. The AP-1/2 and SP/KLF families may help convert SNVs to altered binding and transcription. The promoter region is buffered against binding changes, especially when many TFs are bound. Risk alleles are not greatly biased to loss-of-binding or activity. AP-1 ASBs predict transcriptional changes. Noncoding SNVs act through altered homeostatic TF binding to alter differentiation and disease risk.

contributor to, or secondary consequence of, core pathogenic processes that drive the emergence of these inflammatory and neoplastic skin diseases. The present work provides support for a rationale to explore potential primary contributions of differentiation gene dysregulation to the emergence of common polygenic skin diseases.

Breaches in the cutaneous barrier trigger a cascade of inflammatory responses to fight incoming pathogens, a process facilitated by the production of a host of pro-inflammatory cytokines in the epidermis[89,90]. Consistent with a role for dysregulation of this process in inflammatory and neoplastic skin disorders, daSNV targets also encompassed genes encoding proteins with known pro-inflammatory activities in skin, including IL23A, IL4, IL13, TNFR1, IL18R1, and IL22 as well as interferon network proteins IFNGR2, IFNLR1, IFNAR1, IRF1, STAT2, and STAT3. IL23A, IL4, IL13, and TNFR1 are targets of medically efficacious drugs in widespread use in psoriasis and atopic dermatitis, consistent with the observed enhanced likelihood of success for polygenic disease therapeutics directed at GWAS-linked targets. Prominent among other putative daSNV target genes involved in immunity and inflammation were those encoding major histocompatibility (MHC) genes. While both class I and II MHC proteins can be expressed by keratinocytes in epidermis[91,92], HLA-C class I genes are of particular interest as they reside within the strongest risk locus for psoriasis[93,94]. The specific mechanism whereby HLA-C alleles contribute to psoriatic disease development, however, is presently not definitively established. Understanding whether dysregulated MHC gene expression may play a pathogenic role in psoriasis and the other studied polygenic skin diseases is a topic of future interest as is the development of new anti-inflammatory approaches to prevent and treat these disorders.

TF expression and activity are dynamic over cellular differentiation, including in epidermal keratinocytes[95]. Consistent with this, transcriptional activity at disease-linked sequences was dynamic, displaying a transient peak in SP/KLF/AP-2/NRF1 motif-driven activity in early differentiation and a sustained increase in AP-1 and CEBPA activity as differentiation progressed, accompanied by a decrease in TEAD1/3 activity (summary schematic in Fig. 9o). Homeostatic epidermal TF knockouts followed by bulk RNA-seq confirmed a prominent role for SP1, AP-1, and TP63 TFs in epidermal gene regulation along with new roles for CXXC1, ZNF217, IRX2, FOXJ2, and NRF1. Two of these novel TFs, ZNF217 and NRF1, exhibit some of the most dynamic binding across differentiation. CXXC1 is known to be recruited by the glucocorticoid receptor as part of a complex to methylate H3K4[96], and the glucocorticoid receptor is the most frequent treatment target for psoriasis and atopic dermatitis[97,98], supporting further study of the interplay between epidermis-intrinsic CXXC1 activity and surrounding inflammatory cells. The protein interaction network of homeostatic TFs identified potential, previously understudied TF-TF interactions that may affect epidermal homeostasis. The convergence on SP/KLF/AP-2/NRF1 from MPRA and TF knockout screens especially

implicate these TF families in both transcriptional control and in potentially converting human DNA variation to altered disease risk.

Population sampling CUT&RUN uncovered a "buffering" effect at promoters against DNA variants with respect to altered TF binding. In this regard, SP/KLF family TFs indicated increased promoter buffering associated with an increased number of additional TFs at the same promoter, suggesting a "peer pressure" model of TF binding to promoter DNA. Such a promoter buffering effect may represent a conserved mechanism to avoid potentially catastrophic promoter failure for physiologically important genes. Consistent with the need to preserve promoter function, it is estimated that enhancers, observed here to lack such comparable buffering, are evolving at a 5-fold faster rate than promoters in studied tissue types[99]. In this regard, it has recently been shown that specific promoter regions are hotspots for mutation in humans, but that human-mouse conserved promoters have a reduction in gene expression QTLs relative to flanking sequence[100]. This buffering can be compared with the previous observation that ASBs in "HOT" regions show reduced motif concordance, suggesting non-motif-driven binding that might be more resistant to variation[79]. Promoter buffering might occur through a variety of mechanisms, including collaborative TF binding interactions that stabilize SP/KLF TF association with promoters, however, further studies will be required to address this and other potential models. It was also observed that JUN ASBs were correlated with transcriptional output in MPRA and motif disruption, indicating potent transcriptional influence of JUN and its cognate motif at skin disease-linked SNVs.

Disease-linked daSNVs may enhance disease predisposition by altering the expression of critical gene regulators themselves. For example, centrally important homeostatic epidermal TF genes, such as *OVOL1, KLF4*, and *TP63* have daSNVs in their own regulatory DNA that are also ASBs for other homeostatic TFs. Because of the capacity for altered TF expression to dysregulate the expression of thousands of direct and indirect target genes, this finding suggests that noncoding variants can alter TF binding to, and expression of, other important TFs to modulate disease risk. The rs12801731 and rs4687102 daSNVs/ASBs, associated with *KLF4* and *TP63* genes, respectively, are especially notable in this regard as *KLF4* knockout impairs terminal differentiation in mice and *TP63* gene deletion abolishes formation of stratified epithelium altogether[53,101]. These observations together fit a model in which non-coding variation affects disease risk through dysregulation of multiple layers of transcriptional control.

Polygenic disorders are believed to arise through an interplay between multiple disease-relevant cell types. For example, a recent study of human epidermal SCC[102] identified no fewer than 20 distinct cell types in skin tumor tissue, a number of whom, such as stromal fibroblasts, are known to be critical to disease progression[103]. Only one of those types, namely epithelial keratinocytes that make up the bulk of the epidermis, was the focus of the present work in skin,

underscoring the need to expand studies to other cell types important in the pathogenesis of polygenic skin disease. In this regard, noncoding SNVs affecting gene regulatory elements active in other cell types, including fibroblasts, immune, and inflammatory cells of central importance in diseases such as epidermal cancers, psoriasis, and atopic dermatitis, are of particular interest. Future work to identify and functionally characterize all the regulatory risk variants and the relevant target genes they dysregulate in the diverse cell types that may contribute to emergence of polygenic disease will shed additional light on disease pathogenesis and on new potential preventive and therapeutic targets.

## Methods

### Ethics approval and consent to participate
The protocol was reviewed and approved by the Stanford Human Subjects Institutional Review Board (IRB), with approval number #35324. The written informed consent was obtained from all human subjects or their parents in the case of newborns.

### Disease incidence
Numbers are from the 2021 GDB database[104] (USA, all ages, both sexes), except those without numbers in the GBD database: dermatomyositis[105] (Alberta, CA adults), lupus[106], rosacea[107] (U.K.) and severe acne (estimated as 15% of the GBD incidence of acne vulgaris[108]).

### Cell-type heritability and genetic risk correlation analysis with LDSC
Stratified LDSC analysis, as previously described by Finucane et al.[109], was performed to obtain cell type-specific heritability estimates for epithelial and immune cell types in skin disease using GWAS summary statistics. Complete harmonized statistics for 9 skin disease GWAS were downloaded from the NHGRI-EBI GWAS Catalog (https://www.ebi.ac.uk/gwas/home). LDSC software was downloaded from Github (https://github.com/bulik/ldsc, v.1.0.1) and baseline model LD scores, weights, and allele frequencies (1000 Genomes European Phase 3) were downloaded from the Broad Institute Google Cloud storage platform: https://console.cloud.google.com/storage/browser/broad-alkesgroup-public-requester-pays/LDSCORE.

ATAC peaks for primary human epithelial and GM12878 cells from Donohue et al.[36], were downloaded from GEO (GSE188398). Cell type-specific annotations were created using the LDSC **make_annot.py** script. Background annotations were separately created from the union set of epithelial cell peaks and immune cell peaks. These annotations were added to the baseline model for LDSC heritability analysis to control for all open peaks and improve cell type-specificity of effects detected.

Summary statistics were prepared for LDSC analyses using the LDSC **munge_sumstats.py** script with all default settings except chunksize=500000. To calculate cell type-specific heritability for a trait, the LDSC script **ldsc.py** was run using the --h2-cts flag, with the background annotations added to the baseline model (--ref-ld-chr) and testing for heritability enrichment in the cell type-specific annotations (--ref-ld-chr-cts). To calculate genetic correlation between traits, ldsc.py was run using the --rg flag for every pairwise combination of summary statistics.

### Cell culture
HEK293T (Takara Bio, #632180) cells were maintained in Dulbecco's modified Eagle's medium (DMEM) media (Gibco, #11995-065) supplemented with 10% fetal bovine serum (FBS) and 1% penicillin–streptomycin at 37 °C with 5% $CO_2$. Primary human neonatal foreskin keratinocytes were isolated and cultured in 50% Keratinocyte-SFM (Life Technologies 17005-142) and 50% Medium 154 (Life Technologies M-154-500) with the addition of Pen/Strep (Life Technologies 15140-122), the composition termed 50:50 media. Because of the

source of primary cells, all keratinocytes are of male sex. All human cells were collected and analyzed by protocols in accordance with the NIH genomic data sharing policy and approved by the Stanford Human Subjects Institutional Review Board. Progenitor keratinocytes were maintained at subconfluence. Keratinocytes were induced to differentiate by seeding cells at confluence in 50:50 media with the addition of 1.2 mM calcium (added roughly 12 h after seeding at confluence) and cultured for 3–6 days.

### Genetic variants linked to skin disease
Lead SNVs from the GWAS catalog[28] (v.1.0) were collected by filtering for the diseases: inflammatory skin disease, psoriasis, atopic dermatitis, esophageal SCC, BCC, acne (severe), esophageal cancer (squamous cell), dermatomyositis, cutaneous lupus erythematosus, rosacea, cutaneous SCC, recalcitrant atopic dermatitis, seborrheic dermatitis, dermatomyositis or juvenile dermatomyositis, alopecia areata, cutaneous psoriasis, psoriasis vulgaris, rosacea symptom severity, non-melanoma skin cancer, keratinocyte cancer (MTAG) and "multiple keratinocyte cancers". Four additional SNVs (rs763035[110], rs224108[111], rs11150780[111] and rs118179173[112]) were included from literature. All SNVs in LD with the lead SNVs ($r^2 > 0.8$) were identified with either LDlink or HaploReg[113] within the populations "EUR", "AFR", "AMR", "EAS", "SAS" and "ALL" separately. This expanded the 852 lead SNVs to 26,278 SNVs. SNVs were filtered for putative regulatory activity if they overlapped a union set of peaks from ATAC-seq (ENCSR968JDE), H3K27ac ChIP-seq (ENCSR171FYM), or H3K4me1 ChIP-seq (ENCSR250WAV) experiments performed in foreskin keratinocytes. This epigenetic filtering reduced the number to 3539 SNVs. SNVs were filtered if they contain an indel greater than 3 bp. The final library was built by including 162 bp of genomic sequence centered on the SNV. 250 random controls were generated by randomly sampling the genomic fragments, scrambling their sequence, and randomly substituting the central nucleotide to create a random SNV. Lastly, if multiple SNVs were within 81 bps, sequences representing the most common haplotypes according to the 1000 Genomes project were included in the library but were not analyzed for this study. Each genomic and random fragment was barcoded with 10 random 16 bp barcodes. The final library contained 82,950 oligos (including controls) and included 3457 SNVs.

### Gene transfer and knockout
For virus production, HEK293T cells in 10 cm plates were transfected with 9 μg of each retroviral expression construct using Lipofectamine 3000 (Invitrogen, L3000015). After 48 h, viral supernatants were collected and concentrated using Lenti-X concentrator (TaKaRa, 631231). The optimized viral titer was then added to primary human keratinocytes along with polybrene (5 μg/mL). The next day, fresh media was added, and the infected cells were selected using puromycin (1 μg/mL) or blasticidin (5 μg/mL).

### Gene knockdown
siRNAs were applied to knockdown target genes. 2.5 μL of 100 μM each siRNA was transfected to 10 cm plates containing 1.3 M resuspended human primary keratinocytes, using Lipofectamine RNAiMax (Invitrogen, 13778150). After 24 h, fresh 50:50 media was added to the cells and cells were subsequentially seeded for differentiation after 48 h transfection. The negative control was the ON-TARGET Smart pool negative control (Horizon, D-001810-10-20). siRNA pools were purchased from Horizon: FOXJ2 (L-009350-00-0010), ETV3 (L-010509-00-0010), SP1 (L-026959-00-0010), IRX2 (L-032005-01-0005), CXXC1 (L-008545-01-0005) and ZNF217 (L-004987-01-0005).

### Massively Parallel Reporter Assays (MPRA)
Oligo libraries were synthesized by Agilent, PCR amplified and ligated into pGreenFire-mCMV (EF1a-puro, System Biosciences) following published protocols[4]. Test sequences are in Supplementary Data 1. For

each MPRA biological replicate, 10 million keratinocytes were transduced in 15 cm plates in 50:50 with the addition of 5 µg/mL polybrene. If a knock-down was performed, lentivirus expressing a lenticrispr v.2 construct with Cas9 and a guide against the knock-down factor was infected at the same time as the guide library: IRF6 (CAT AAG TAG ATC TCA AAC GG), SNAI2 (CCT TGT GTT TGC AAG ATC TG) or NRF1 (AAT TGG GCC ACG TTA CAG GG). Cells were selected in 0.8 µg/mL puromycin for 24–48 h after transduction. Once selected, cells were seeded for progenitor and differentiation conditions. Sequencing library construction followed previous methods[4]: total RNA was isolated using Qiagen's RNeasy Plus kit (Qiagen, #74136) and mRNA subsequently purified using Dynabeads mRNA DIRECT purification kit (ThermoFisher, 61011). Reverse transcription was performed with 500 ng mRNA, 100 nM primer and SuperScript IV (ThermoFisher, 18090050) according to the manufacturer's protocol. Reactions were incubated with 1 µL Themolabile Exonuclease I (NEB, M0568L) at 10 min at 37°, followed by inactivation at 85° for 5 min and cDNA purification with AMPure XP beads (Beckman Coulter, A63880) at a 1:1.1 sample-to-bead ratio. PCR amplification was performed in 50 µL reactions containing 5 µL cDNA, SYBR green (ThermoFisher, S7563), and PrimeStar Max DNA Polymerase (Takara, R045B). Reactions stopped in the early exponential phase, pooled, concentrated, gel purified and sequenced. Sequencing was performed on a NovaSeq 6000, paired-end 150 bp reads, with an average of 14 million reads per sample.

## CRISPR-flow cell culture

Two biological replicates of primary human keratinocytes from separate donors were infected with lentivirus containing the TF knock-out library. Guide sequences are included in Supplementary Data 3. Specifically, 16 million cells in 8 15 cm plates, in 20 mL 50:50 media, were infected with 100 µL polybrene, Cas9 and guide library virus. The next day, cells were lifted and replated at 2 million cells per 15 cm plate with Blasticidin and Puromycin, while also plating 100,000 cells per well in a 6-well plate to determine survival by the Cell-Titer Blue assay in the presence or absence of drug. Cell-Titer Blue measured survival at ~18–20%, giving ~400 cells per guide. Nine days after infection, 40 million cells were harvested and seeded for differentiation across 18 wells at 1.6 million cells per well in 6-well plates. When seeded for differentiation, cells were visually inspected for the expected morphology changes. At the same time, five 10 cm plates were seeded at 100,000 cells per plate for the "progenitor" population. The next day, $Ca^{2+}$ was added to 1.2 mM in the media of the differentiating cells and half of the media was replaced each day until harvesting after four days. At harvest, single-stain controls of 1 million cells were aliquoted. Total cell counts were 70 million differentiated cells per replicate and 3.5 million undifferentiated cells per replicate.

## CRISPR-flow staining and sorting

Cells were resuspended in Cell Staining Buffer (Biolegend, #420201) at 100 µL per million cells, combined with 1.5 volumes Fix/Perm buffer (Biolegend, #426803), incubated for 20 min at room temperature, washed once with Perm/Wash buffer (10 mL per sample, Biolegend #426803), then with Cell Staining Buffer and stored at 4° until stained. For staining, cells were washed with Perm/Wash buffer, then stained with 1 µL anti-KRT10, AF647-conjugated antibody (Novus Bio, NBP2-47825AF647) per 1 million cells for 20 min in the dark at room temperature (100-fold dilution). Cells were then washed with Perm/Wash buffer, then Cell Staining Buffer. After washing, cells were resuspended in 5 mL Cell Staining Buffer and passed through the filter lid of a FACS tube. 400 µL of the filtered, differentiated cells was aliquoted as the "unsorted" population. Cells were sorted on a FACSymphony S6 Sorter into high, medium and low KRT10 populations. Sorted cells were washed with PBS with 0.1% BSA once, transferred to 1.6 mL Eppendorf tubes and frozen.

## CRISPR-flow sequencing library construction

To extract DNA, cells were resuspended in 200–400 µl of lysis buffer (50 mM Tris-HCl, pH 8.1, 10 mM EDTA, 1% SDS), and heated at 65 °C for 10 min with shaking to reverse formaldehyde cross-linking. Samples were cooled and 4 µL 100 mg/mL RNAse A (Qiagen, #19101) was added. Samples were mixed and incubated at 37° for 30 min with shaking. After RNAse digestion, 20 µL proteinase K (20 mg/mL, ThermoFisher AM2546) was added and samples were incubated for 2 h at 37°, then 95° for 20 min (no shaking). DNA was purified using Ampure/SPRI beads. Specifically, 1.8X sample volume of beads was added, mixed by pipetting and incubated for 4 min before loading on a magnet, incubating for 5 min and discarding the supernatant, then washing twice with 80% ethanol, air-drying the beads for 5 min and eluting in 100 µL 10 mM Tris pH 7.5. A typical yield was ~4–5 µg DNA per million cells. PrimeStar (Takara, #R045B) PCR reactions in 100 µL reaction volumes were performed with 500 ng genomic DNA and 20 pmol each primer (GGW130/GGW209, sequences below) per reaction. The PCR program was 98° for 1 mi, then 5 cycles of 98° 10 s, 56° 10 s, 72° 15 s, and a final 72° for one minute at the end. A Zymo PCR cleanup kit (#D4013 or #D4029 for large volumes) was used to combine the PCR reactions, eluting in 100 µL of the kit's elution buffer. A SPRI clean-up was then performed to remove primers. Specifically, 1.1X bead volume SPRIselect beads were added to the reactions, pipet mixed 15 times, incubated at room temperature for 5 min, placed on a magnet, supernatant removed, washed twice with 200 µL 80% ethanol, air dried for 2 min, removed from the magnet, pipet mixed 15 times with 46 µL 10 mM Tris pH 7.5, incubated for 2 min at room temperature, placed again on the magnet, and 45 µL transferred to a new tube. This purified product was then input to a second PrimeSTAR PCR reaction in 100 µL volumes to attach the full sequencing primers and barcodes. 20 pmol of each primer (TruSeq adapters with dual i5/i7 indexing) were combined with 50 µL PrimeSTAR, 43 µL water and 3 µL 33X SYBR dye. The PCR program was the same as the first, except without the final 72° elongation for one minute, and samples were stopped after 20 cycles based on the amplification trace. PCR product was then run on a 2% agarose gel and gel purified before sequencing on a NovaSeq 6000, paired-end 150 bp reads, with an average of 15 million reads per sample.

The primer sequences are:

GGW130

ACACGACGCTCTTCCGATCTNNNNNNNNNNTGTGGAAAGGAC-GAAACACC

GGW209

GTGACTGGAGTTCAGACGTGTGCTCTTCCGATCGTAA-TACGGTTATCCACGCGG

Example barcoding i7 GGW133-D701 (Index = ATTACTCG): Adds 32 bp, primes in GGW209, adds R2 index (i7).

CAAGCAGAAGACGGCATACGA-GATCGAGTAATGTGACTGGAGTTCAGACGTG

Example barcoding i5 GGW132-D501 (Index = TATAGCCT): Adds 50 bp, primes in GGW130, adds R1 index (i5).

AATGATACGGCGACCACCGAGATCTACACTATAGCCTA-CACTCTTTCCCTACACGACGCTCTTCCGATCT

## Perturb-seq

CRISPR guides (Supplementary Data 3) were cloned into a lenti-CRISPRv2 vector according to published protocols[114,115]. Guide sequences are included in Supplementary Data 3. Primary keratinocytes were transduced with Cas9 lentivirus (aiming for MOI ≥ 9) and gRNA library lentivirus (aiming for MOI ≤ 1.5). Virus-containing media was replaced with regular 50:50 media 24 h post-transduction. 48 h post-transduction, drug selection was performed with blasticidin (0.5 µg/ml) and puromycin (1 µg/ml) for a minimum of 3 days, after which cell viability was assessed by CellTiter-Blue (Promega, G8080). The following cell culture was performed the same as CRISPR-flow, except cells were harvested after 0, 3, or 6 days of differentiation. Cell

harvesting followed the 10X Genomics protocol for Single Cell Suspensions from Cultured Cell Lines for Single Cell RNA Sequencing. 40,000 cells per well were added to 10X Genomics Chromium chips. Libraries were prepared using the Chromium Single Cell 3' gene expression and feature barcode library kits (RevA and RevB, 10X Genomics) according to manufacturer's instructions. Sequencing was performed on a Novaseq 6000, paired-end 150 bp reads. According to the estimates from Cellranger (10X Genomics), before cell filtering, the depth for day 3 (larger library) was 4.3 billion reads for gene expression (25,000 reads per cell, 175,000 cells before filtering, ~100 cells/guide). For the smaller library, there were 2.9 billion reads for gene expression (66,000 reads per cell, approximately 44,000 cells, ~100 cells/guide). There were ~8000 reads per cell for the guide library.

## Split-pool

To express guides for split-pool, the CROP-seq vector was obtained from Addgene (CROP-seq-opti #106280) and modified by inserting the 10x "capture sequence 1" in the guide backbone. Perturb-seq was carried out as for 10x sequencing, except harvesting at day 4 of differentiation. Split-pool sequencing was performed according to the SPLiT-seq protocol[60].

## RNA-seq

Total RNA was extracted using the RNeasy Plus kit (QIAGEN, 74136) and RNA-seq libraries were prepared with QuantSeq 3' mRNA-Seq V2 Library Prep Kit FWD with UDI for Illumina (Lexogen, 191.96) with UMI (Lexogen, 018.96) following the manufacturer's protocol. RNA-seq was performed on a NovaSeq X Plus, paired-end, 150 bp length, with ~30 M reads for each sample. Differentiation was confirmed by marker gene expression (e.g., ITGB1, MKI67, LCE2A) in addition to visual inspection of cellular morphology.

## CUT&RUN

CUT&RUN was performed on 0.4-1 million primary keratinocytes/ experiment using reagents from the EpiCypher kit (EpiCypher, 14-1048), either using antibody to the endogenous TF (the majority) or infected with virus expressing HA-tagged TFs (NRF1 and ZNF217). Keratinocytes were washed with PBS, harvested with trypsin, quenched with DMEM, and resuspended in cold PBS + 0.1% BSA to be maintained afterwards at 4 °C. Cells were washed twice with PBS + 0.1% BSA, and quantified with a Countess. Nuclei were extracted by resuspending in 100 μL nuclei extraction buffer per 1–2.5 M cells (20 mM HEPES pH 7.5, 10 mM NaCl, 0.5 mM spermidine, 0.1% BSA, 0.1% NP-40, and 1 tablet Roche mini protease inhibitor per 10–15 mL) and incubating for 10 min on ice. 5–10 volumes of wash buffer (20 mM HEPES pH 7.5, 150 mM NaCl, 0.5 mM spermidine, 0.1% BSA, 0.05%Triton X-100, and Roche protease inhibitor) were added, cells centrifuged at 600 RCF, and supernatant removed. After an additional wash, cells were quantified using a Countess. 400K-1M nuclei were bound to 10 μL activated Concanavalin A beads (EpiCypher, 21-1401) per sample for 10 min incubation at 4 °C. After removing supernatant, beads were incubated with rocking at 4 °C for 2 hr with 1 μL antibody diluted in 50 μL Antibody Buffer (wash buffer with 2 mM EDTA). After staining, cells were washed twice with wash buffer, and incubated with 1.5–2.5 μL pAG-MNase (EpiCypher, 15-1016) per sample for 1 h at 4 °C with rocking, then washed once with 150 μL wash buffer, then 150 μL low salt buffer (3.5 mM HEPES pH 7.5, 0.5 mM spermidine, 0.1% BSA, 0.05%Triton X-100, and Roche protease inhibitor) and resuspend in 50 μL low salt buffer. After 5 min on ice, 2 μL of 250 mM $CaCl_2$ was added to the beads and beads were subsequentially incubated on ice for 18–25 min. The reactions were quenched with an equal volume 2X STOP buffer (300 mM NaCl, 20 mM EDTA, 40 mM EGTA, 0.025% SDS, 0.5% Triton X100, 0.05 mg/mL RNAse A, 0.05 mg/mL glycogen) for 30 min at 37 °C according to the manufacturer's protocol. Beads were placed on a magnet, and supernatant containing released fragments

was removed. DNA was extracted by a Zymo DNA Clean and Concentrator (Zymo Research, D4014), eluting in 27 μL water. Library was prepared using the NEB Ultra II DNA library prep kit (NEB, E7103S) following dx.doi.org/10.17504/protocols.io.bagaibse[116], with the following modifications: the adapter was diluted 20–25 fold, PCR was performed with SYBR dye to visualize amplification, and after PCR, 20 μL of water, then 30 μL of beads (0.6X instead of 0.8X) beads were added for the initial selection to remove large small fragments, followed by the second size selection using the addition of 20 μL beads (1X instead of 1.2X) for library capture. Sequencing was performed on a NovaSeq X Plus, paired-end, 150 bp length, with 15–20 M reads for each sample.

## Recombinant protein purification

Recombinant target proteins were produced in HEK293T. 20 μg of pLEX FHH-tagged target protein plasmid was transfected per 15 cm plate of ~80% confluent HEK293T cells using Lipofectamine 3000 (Invitrogen). After 48 h, cells were harvested, lysed on ice for 30 min in lysis buffer (50 mM Tris-HCl pH 7.5, 300 mM NaCl, 1 mM EDTA, 1% Triton X-100, 1X Protease Inhibitor Cocktail [Sigma, P8340]), then sonicated for 3 cycles of 10 s at an amplitude of 10%, with 10 s pauses. Lysate was centrifuged at 16,000 g for 10 min before quantification, and subsequently added to a saturating volume of anti-FLAG M2 affinity gel (Millipore Sigma, A2220), using a volume determined by a small scale, pilot purification for each protein. Purification was overnight at 4 °C and the next morning beads were washed three times with wash buffer (50 mM Tris-HCl pH 7.5, 3 mM EDTA, 0.5% NP-40, 500 mM NaCl, 10% Glycerol, 0.1 mM DTT). Prior to elution, M2 beads were primed with two washes of Elution Buffer (1X PBS), then protein was eluted with 0.5 mg/mL 3X FLAG peptide in Elution Buffer. Elution was performed once at 1.5X bead volume, 1 h incubation. Eluates were concentrated with 3k molecular weight cutoff columns (Millipore, UFC500396). Finally, protein concentration was with a BSA standard curve on a Bis-Tris gel and staining using InstantBlue Coomassie Protein Stain (Abcam, ab119211).

## Microscale thermophoresis

Recombinant IRF6 protein was labeled using the Monolith His-Tag Labeling Kit RED-tris-NTA (Nanotemper Technologies, MO-L018) according to the manufacturer's protocol at a 1:2 dye:protein ratio, in a reaction with a final concentration of 100 nM protein. The labeled protein concentration was determined using a BSA standard curve. Labeling was evaluated by a capillary scan at 60% LED power. Protein was diluted to 600 units using Nanotemper-provided 1x PBST, resulting in a target protein concentration of approximately 50 nM. The labeled protein was then mixed with the ligand and incubated for 5 min at room temperature before being loaded into Monolith NT.115 Capillaries (NanoTemper Technologies, MO-K022). MST was measured using a Monolith NT.115 instrument (NanoTemper Technologies) at the ambient temperature of 25 °C, with instrument parameters set at 60% excitation power and Medium MST power. Data from at least three independently-pipetted measurements were evaluated using the fraction bound (MO.Affinity Analysis software, NanoTemper Technologies). The sequences for DNA are: protective, 5'-CAA TCA CCT ACT TAA TTC TAA-3'; risk, 5'-CAA TCA CCT ATT TAA TTC TAA-3'.

## Vector construction for homology-directed repair

The donor ssAAV vector used as a template for homology-directed repair (HDR) was constructed by cloning genomic DNA fragments flanking the 5' and 3' end of the target SNPs (1862 bp for rs72696969, 1918 bp for rs2349075 and 1395 bp for rs4704864) into the AAV transfer plasmid between AAV ITR sequences. For each editing experiment, two consecutive constructs were generated differing by possessing the reference or alternate alleles. During genomic amplification, rs72696969 was engineered with reference allele G *vs* alternate

allele A, reference allele A vs alternate allele G for rs2349075, and reference allele C vs alternate allele G for rs4704864.

For genomic DNA amplification, following primers were used:
FLGF: 5' ATCAACGCGTGCTAGCCACGGATCTGCTTTTGACCT
FLGR: 5' GCTTGATATCGAATTCCCATGCTAACAAGCACGCAT
CASP8F: 5' ATCAACGCGTGCTAGCTTACAACCCAGTGGCTTTCATT
CASP8R: 5' GCTTGATATCGAATTCGCCTGTTCATGGTGCTCAGA
NIPAL4F: 5' ATCAACGCGTGCTAGCCCTGAGACCTTCATGATGTGT
NIPAL4R: 5' GCTTGATATCGAATTAACCCTTTAGGCTGGTACTATTC

Primers contained homology arms to the AAV transfer vector allowing In-Fusion assembly into NheI/EoRI digested AAV donor plasmid. After confirmation of the insert sequence integrity, constructs were used for AAV virus production using the AAV Helper-free system in 293 T cells. AAV donor vectors containing reference or alternate mutations were contransfected with pHelper and AAV-DJ plasmids and ssAAV virus crude extracts prepared 72hrs post transfection in accordance with prior work[117]. AAV-DJ serotyped donor ssAAV virus was produced at genomic titer of $2-3 \times 10^{13}$ TU/mL.

### CRISPR and AAV mediated homology-directed repair

The guide sequences targeting SNPs for CRISPR/Cas9 genome editing were predicted using the CHOPCHOP web tool[118] and were ordered from IDT as sgRNA. The following gRNA sequences were used in this study:

For rs72696969: 5' GACCTTATTACACAATAGAG
For rs2349075: 5' AGTCTCAATATGAGTCATTG
For rs4704864: 5' TACAGCAGGCATGTGATAAA

For CRISPR/Cas9 mediated genome editing, 73 pmol of the sgRNA was complexed with 61 pmol Recombinant Alt-RspCas9 protein (IDT) in 10 μL of Human Keratinocyte Nucleofector™ Kit solution (Lonza, VPD-1002) for 10 min and immediately used for nucleofection of $8 \times 10^5$ primary keratinocytes with the Amaxa nucleofection apparatus (Lonza) using program T-018. After recovery, cells were mixed with AAV virus containing either reference or alternate allele of the donor template at MOI $2.5 \times 10^5$ for rs72696969 and MOI $3 \times 10^6$ for rs2349075 and rs4704864 edits, split into 2 wells of a 6-well plate and propagated for 72 h. After cultures reached 60–80% confluence, cells were transferred into 10 cm plates, during which time genomic DNA was isolated and evaluated for the editing efficiency using PCR amplification and sequencing of the bulk cell population, as well as by cloning the amplified fragments into pBluescript vectors and evaluating editing efficiency by individual colony sequencing. The FLG rs72696969 SNP was edited in a homozygous background with editing efficiencies of K236K AAG/AAA – 76%, K236R AAG/AGG – 71%. The NIPAL4 SNP rs4704864 was edited in a heterozygous background with reference C/C allele gain – 24.1%, alternate G/G allele gain – 12.4%. Finally the CASP8 SNP rs2349075 was edited in a heterozygous background with reference A/A allele gain – 13.9%, alternate G/G allele gain – 17.7%. Allelic effects on RNA expression were determine by qPCR between the two edited populations (reference and alternate).

### Data analysis

**MPRA analysis.** UMIs and barcodes were extracted from sequencing reads using UMI-tools[119] v. 1.1.5. Bowtie[120] v. 1.3.1 was used to map barcodes to a reference index of the barcode library allowing for up to one mismatch. The number of UMIs per guide was calculated from the mapped read files using the UMI-tools count program with the "directional" method. MPRAnalyze[121] v. 1.9.1 was used to determine differentially active genomic fragments. SNVs were only analyzed if they had at least 5 barcodes detected in the plasmid DNA library for both the alt and ref allele. Differential activity for the individual timepoints was calculated using "analyzeComparative()" with DNA design "-barcode", RNA full design "-allele" and reduced design "-1". For the analysis of timepoint-allele interactions, the RNA full design was

"-barcode + timepoint + timepoint:allele" and reduced design "-barcode + timepoint".

**ChromHMM enrichment.** ChromHMM annotations[122] from the 15-state model in keratinocytes (E057) were obtained from Roadmap[123]. For determining enhancer activity without considering risk/protective allele differences, each state was separately analyzed, with a two-sided t-test used to test enhancer activity (median adjusted alpha values from MPRAnalyze) in the state vs enhancers not in the state (no multiple hypothesis correction). The test for interaction terms was the same except using the timepoint interaction term from MPRAnalyze. For testing daSNV activity, daSNVs were defined as those identified by MPRAnalyze with an allelic difference $P < 0.01$, then a two-sided Fisher's exact test was performed to determine whether daSNVs were enriched in each state (no multiple hypothesis correction).

**rGREAT analysis of daSNV enrichment in functional genes.** rGREAT v. 2.6.0 was installed in R v. 4.4.3. A "gmt" formatted file for monogenic skin disease genes was created manually and loaded with rGREAT::read_gmt, with coordinates drawn from org.Hs.eg.db. MRPA SNV locations were defined as the single-nucleotide position of the SNV and enrichment calculated with the rGREAT::great function using the parameters mode = "oneClosest" and extension = "10000"; that is, the window of gene association was narrowed to just the closest within 10 kbp. The enrichment of daSNVs relative to all tested SNVs was calculated by adding by passing all tested SNVs with the background parameter.

**Split-pool analysis.** Split-pool analysis was performed with custom snakemake pipelines, using kallisto[124] in "SPLiT-seq" mode to quantify RNA expression. Scanpy[125] objects output by kallisto were converted to Seurat objects and then processed like 10x data.

**Perturb-seq analysis.** Raw 10x data was processed using *Cellranger* (v. 3 for the larger screens, v.5 for the smaller screens) to produce h5 files, which were converted to Seurat objects. Count data was normalized with SCTransform[126]. Cells were filtered to only those with a single sgRNA, less than 5% mitochondrial reads and at least 200 RNAs. Dimensionality reductions were performed with the RunPCA and RunUMAP Seurat functions, except for the pseudotime analysis. Pseudotime analysis was performed by passing the SCT normalized expression data to the reduce_dimensionality function of SCORPIUS, using 10 dimensions and Pearson distances, then trajectories were inferred with SCORPIUS[127]. For the Day 3 larger screen, the data was subset to the 2000 most variable features (using the Seurat function FindVariableFeatures) before dimensionality reduction to reduce memory usage. To test for effects on pseudotime scores, single-sided Mann–Whitney U tests were performed on cells with guides for a target gene vs all other cells with a single guide. For effect sizes, Hedge's g values were computed with equal variances; alternatively, effect sizes were defined by taking the average pseudotime value for cells with guides targeting a given gene, subtracting the mean pseudotime for the experiment, and converting these values to Z scores. P values were combined across Perturb-seq and FACS datasets by Stouffer's method, weighting the P values by the total number of cells divided by the number of guides in the experiment. Hedge's g values were combined by taking the weighted average of the experiments' g values, with weights defined as the average cells per guide. One-sided p values were combined separately, and a final two-sided p value was computed by taking the lower of the one-sided p values and multiplying by two. FACS studies were given equal weight as the Day 3 larger screen in the P value combination. To combine effect sizes between FACS and perturb-seq, the MageCK RRA estimated log fold changes in FACS data were converted to Z scores, the Pertub-seq g values were also Z

normalized, and the overall effect magnitude was the average Z score, with weights the same as for *p* value combination.

To analyze enrichment of cells at specific points along the pseudotime axis, a gaussian kernel density estimate was fit to the pseudotimes of all cells and to cells with a guides targeting the gene of interest using the scipy.stats.gaussian_kde function, which was then used to estimate the cell count density across the pseduotime axis evaluated at 40 evenly-spaced positions. A window of length 10 was rolled along the pseudotime space and one-sided Mann–Whitney U tests were applied to the density estimates from cells with the gene of interest *vs* all cells. Only the enrichment of cells with the guides of interest were evaluated. The p values from these tests at each position were combined across the perturb-seq experiments using Stouffer's method and weighting by the number of cells with guides targeting the gene of interest. The negative $\log_{10}$ value of these scores were plotted, clipping the maximum value at 4.

**IL1 family effects.** The gene expression for IL1A, IL1B and IL18 was defined by the SCT (Single-Cell Transform) normalized values in Seurat and LOWESS smoothing was used to generate an "expected" gene expression value for a given pseudotime value. The "expected" gene expression value was subtracted from the actual observed value for each cell to generate the residual. Whether a gene has a significant effect on gene expression was calculated by performing a t-test between the residuals for KO cells vs non-KO cells. This number was finally Benjamini-Hochberg corrected to an FDR to adjust for testing multiple TFs.

**CRISPR-flow analysis.** Raw data was processed with a custom snakemake pipeline that counts the number of UMIs per guide in each cell pool. The number of UMIs per guide were input to MageCK RRA analysis, pairing biological samples together ("paired" option) and using the safe targeting guides as controls. Counts in the "high KRT10" group were compared against all of the other divisions of that cell population (unsorted, low and medium KRT10 groups), and the reverse for the "low KRT10" group. The combined statistic for depletion from KRT10-low and enrichment in KRT10-high was taken as the pseudo-P-value from combining by Fisher's method the depletion from KRT10-low and enrichment in KRT10-high. When the p-values for the enrichment in high KRT10 *vs* depletion in low KRT10 (and vice versa) are combined, they are only pseudo-p-values (here termed ψP), since each sorting group is part of the background set for the other and the *P* values are therefore not independent (in addition to the fact that that enrichment and depletion are different hypotheses in this case). The minimum of the ψ*P* values for increased or decreased KRT10 was used as the overall statistic for an effect on KRT10 expression. To determine the FDR of these ψP, it was assumed that (1) all un-expressed genes are true negatives, (2) there is a value true_positives as the true number of TFs with an effect in the genome. A simulation of picking genes from the genome according to a range of false positive rates was run and the false positive rate that resulted in the observed fraction of un-expressed genes was taken as the estimated FDR for a given ψP value.

**RNA-seq analysis.** QuantSeq reads with UMIs were processed with a custom Snakemake pipeline according to the manufacturer's recommendations. First, UMIs were moved to read names using umitools[128] v1.1.2 (umi_tools extract –bc-pattern=NNNNNN), then adapters, poly(A) and low quality bases were removed with BBMap v.39.01 (bbduk.sh ref=data/polyA.fa.gz,data/truseq.fa.gz $k$ = 13 ktrim=r useshortkmers=t mink=5 qtrim=t trimq=10 minlength=20). Reads were mapped with STAR[129] (v.2.7.4a) to hg38, Gencode v39 annotations (--limitOutSJcollapsed 2000000 --outFilterMultimapNmax 20 --alignSJoverhangMin 8 --alignSJDBoverhangMin 1 --outFilterMismatchNmax 999 --outFilterMismatchNoverLmax 0.1 --alignIntronMin 20 --alignIntronMax 1000000 --alignMatesGapMax 1000000), followed by filtering with samtools v.1.18 (samtools view -b -F 4 -F 256), deduplication using the UMI (umi_tools dedup), then featureCounts (from Rsubread[130] v.2.10.5) quantification of reads per gene using bam inputs (featureCounts -s 1 -t exon -g gene_id). DESeq2[131] (v.1.38.3) was performed regressing out batch effects, and lfcShrink was type = "normal".

**CUT&RUN read processing and mapping.** CUT&RUN analysis was performed with custom Snakemake pipelines. Adapters were removed using cutadapt[132] v.4.5, aligned to hg38 with bowtie2[120] v.2.4.2 (--end-to-end --very-sensitive --no-mixed --no-discordant -I 10 -X 700), and duplicates removed (samtools fixmate -m {input} - | samtools sort -T {params.tmp} - | samtools markdup - - | samtools view -b -f 0×3 -F 0×400 −). Replicates were combined with samtools merge. Peaks were called with MACS2[133] (v.2.2.9.1) vs IgG controls (macs2 callpeak -t {input.expt} -c {input.ctrl} -g hs -f BAMPE -q 0.1). Peaks were filtered using the ENCODE exclusion list (ENCFF356LFX). For motif analysis, random sequences matching the peak sequence lengths were output at the same length distribution and number (minimum 20,000 sequences). Motifs were identified from FDR < 0.01 MACS2 peaks either de novo with DREME[73] v.5.5.4 (-m 5 -t 14400) or using the JASPAR2022_CORE_vertebrates database[134] and AME[135] v.5.5.3 (--seed 14).

**Allele specific binding analysis.** Allele specific binding analysis was performed with custom Snakemake pipelines. The CUT&RUN snakemake bam files are deduplicated based on retaining only one read with a given start and end coordinate mapping, keeping the read with the highest sequencing quality score (samtools markdup), thereby removing PCR duplicates while being agnostic to the mapping quality. Variants were called with MAPQ > 10 read filtering using BCFtools[76] v.1.17 (bcftools mpileup -q 10 -Ou -f {input.fa} {input.bam} 2 >/dev/null | bcftools call -vmO z -o {output.vcf}). Known variants were obtained from dbSNP[136] build 151 (00-All.vcf.gz). Variants were filtered against regions of frequent structural variation (NCBI's nstd186). For further analysis, only variants with bcftools "0/1" genotypes were retained. Read counts from multiple heterozygous samples with the same anti-TF antibody were summed before hypothesis testing for allele specific binding. The probability for allele-specific binding was calculated with a two-sided binomial test with null hypothesis $P = 0.5$ using the read counts from the DP4 VCF field. Mitochondrial variants and regions significant at a binomial $P < 0.01$ in any of the 60 IgG samples or their combination were removed. To control for mapping biases, 100 reads of length 70 (a short length chosen to increase the detection of mapping errors, which is roughly the bottom 10% of actual insert lengths) were randomly distributed in the 140 bases centered around the SNV of either reference or alternate sequence. The simulated reads were mapped against the genome and processed through the same pipeline, except without MAPQ filtering. ASBs with 20% differences in mapped read numbers between reference and alternate alleles were removed as mapping artifacts. Motifs were identified using FIMO[137] v.5.1.1 and the JASPAR2022_CORE_vertebrates database[134] (--thresh 1e-3). The maximum -log10 $P$ value of the cognate FIMO match was used as the measure of a motif match, with decreases in the significance of the match denoting a predicted decrease in binding. The JUN::JUNB JASPAR2022 motif was used for both JUN and JUNB, and the TFAP2A motif was used for TFAP2A and TFAP2C.

**CUT&RUN SNV enrichment analysis.** The enrichment of skin disease SNVs relative to other diseases was specifically for the diseases stroke, chronic obstructive pulmonary disease, coronary disease and type II diabetes. All SNVs for these diseases were epigenetically filtered in the same manner as the skin disease SNVs, then combined for a total of 4582 variants. This number was downsampled to the 3443 skin disease variant number. Peaks (MACS2 FDR < 0.1) overlapping with SNVs were counted and the number of overlapping variants for other diseases was

compared with the number of overlapping skin disease SNVs; enrichment was calculated by a one-sided t-test for increased binding to skin disease SNVs.

**CUT&RUN co-binding analysis.** MACS2 FDR < 0.1 peaks and random-sequence controls length-matched to the peaks were analyzed by centrimo (MEME suite v. 5.1.1) using –neg option to control Day 0 to Day 4 and vice versa; parameters were otherwise defaults. Motifs were taken from the JASPAR2022 CORE vertebrates, non-redundant, dataset. The same MACS2 FDR < 0.1 peak sets were used for peak overlaps, which were computed with pybedtools[138] v. 0.9.1. The top 2000 peaks by FDR were downsampled to the number of peaks in the smallest dataset (916, Day 4 RUNX1).

## Statistics and reproducibility
At least two biological duplicates were used. No statistical method was used to predetermine sample size. Samples that failed quality control (e.g., lack of distinctiveness from IgG controls in CUT&RUN) were excluded. Experiments were not randomized, and experimenters were not blind to group assignment. Omics data was analyzed with published packages and corrected for multiple hypotheses.

## Reporting summary
Further information on research design is available in the Nature Portfolio Reporting Summary linked to this article.

## Data availability
The CRISPR-flow guide library and MPRA sequencing data generated in this study has been deposited in the GEO database under accession code GSE255326. Under the same accession are additional processed data files for CUT&RUN and RNA-seq, including bigwig files of CUT&RUN data. Raw primary keratinocyte sequencing data is available under restricted access at accession code phs003977 [https://www.ncbi.nlm.nih.gov/projects/gap/cgi-bin/study.cgi?study_id=phs003977.v1.p1] to protect patient privacy; access can be obtained by request from dbGaP for general research use. MPRA data is available at the ARVID website [https://arvid-data.shinyapps.io/skin/] and reads-per-million normalized CUT&RUN data, along with peaks, can be accessed in the UCSC genome browser at this link [https://genome.ucsc.edu/s/Max/cnr]. GWAS data for LD correlation analysis was obtained from the GWAS catalog [https://www.ebi.ac.uk/gwas/home], at accessions GCST90429798, GCST9002716, GCST90041916, GCST90013410, GCST90137411, GCST90137412, GCST90041917, GCST90044515, and GCST90014456. Allele frequences for LD correlations was from the publicly available "1000 Genomes European Phase 3" dataset [https://console.cloud.google.com/storage/browser/broad-alkesgroup-public-requester-pays/LDSCORE]. ATAC peaks and HiChIP-seq were downloaded from GEO, accession codes GSE188398 and GSE188401 under the SuperSeries GSE188405. SNV filtering for MPRA utilized published data from ENCODE at accessions ENCSR968JDE, ENCSR171FYM and ENCSR250WAV. Source data are provided with this paper. All other data supporting this study are in the article or its Supplementary Information.

## Code availability
Data processing used existing code/software packages. Pipelines for CUT&RUN analysis are available at https://github.com/khavarilab/cutandrun-analysis.

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

## Acknowledgements

This work was supported by the USVA Office of Research and Development (to P.A.K.), a USVA Merit Review grant (BX001409 to P.A.K.), NIAMS/NIH (AR076965 & AR045192 to P.A.K., AR082351 to D.F.P., AR084077 to W.M.), NHGRI/NIH (HG010856 to P.A.K.) and generous support from K. Fields and G. Rayant (to P.A.K). R.Z. was supported by an Impetus grant, which was supported by the Hevolution Foundation and Robert Rosenkranz. T.N. was supported by the American Heart Association Postdoctoral Fellowship AHA000POSTCH0217435. This work was supported by the UCSC Genome Browser Group for Data Integration and Data Dissemination (U24 HG002371). We thank F.C. Porter (Caltech) for advice on statistical analysis.

## Author contributions

D.F.P., D.L.R., R.M.M., and P.A.K. conceived of the project. D.F.P., W.M., D.L.R., A.W.H., X.Y., S.M., Z.S., L.D., J.M.M., D.T.N., L.A.K., and I.E. performed experiments. D.F.P., R.M.M., T.F., Z.S., S.S., M.G., L.N.K., I.E.P., G.K., M.C.G.W., L.V.J., V.L.P., and P.A.K. analyzed data. S.T. cultured primary cells. R.Z., T.N., and J.M.E. developed the split-pool Perturb-seq protocol. B.B.L., S.H.K., and Y.Q. assisted with performing and analyzing split-pool. D.F.P. and P.A.K. wrote the paper with input from the other authors.

## Competing interests

J.M.E. has received materials from 10x Genomics unrelated to this study and has received speaking honoraria from GSK plc and Roche Genentech. The remaining authors declare no competing interests.
