## [Transparent Peer Review file · Nature Communications]

Disease-Linked Regulatory DNA Variants and Homeostatic Transcription Factors in Epidermis

Corresponding Author: Dr Paul Khavari

Version 0:

Reviewer comments:

Reviewer #1

(Remarks to the Author)

The study by Porter and colleagues, titled "Disease-linked regulatory DNA variants and homeostatic transcription factors in epidermis," investigates regulatory variants and networks that govern the homeostasis and differentiation of keratinocytes. The authors have identified a panel of candidate regulatory variants linked to several skin diseases. They have assembled a large number of experimental datasets generated by this study into a publicly accessible web resource representing an atlas of regulatory networks, TFs, and variants underlying epidermal homeostasis. The experimental data spans various technologies, including MPRA, CRISPR-flow, Perturb-seq, siRNA RNA-seq, and CUT&RUN profiling keratinocytes at different development stages. With these data, the authors specifically focus on individual variants and TFs, carefully dissecting the etiology of several skin diseases and causative regulatory variants and allele-specific binding of corresponding TFs. This study provides a valuable resource for studying epidermal differentiation and skin diseases.

Major Concerns:

1. While the presented study clearly demonstrates the causative effects of multiple TFs, the lack of reported TF-TF interactions is rather glaring. Those could probably be simply inferred as either overlapping ChIP peaks or proximal motif pairs.
2. Throughout the manuscript, there appear to be multiple instances of inferring a bound TF simply via the presence of a particular motif. Computationally predicted TF binding motifs are notoriously prone to incorrect TF calls. Please go through the text to emphasize the highly speculative nature of such calls. These motif-TF associations contrast drastically with the solid nature of the experimental data on TF binding presented in other parts of the manuscript.
3. The section "Identification of rs4687102..." appears rather speculative and might benefit from being moved to Supplemental Materials. While the "ATF1/2/4 motif" appears to be disrupted, the possibility of a disruption of a motif of another TF hasn't been rejected. The experiments result in an insignificant change in the binding of ATF4. While it might be possible that ATF4 binding is affected, there is a lack of solid evidence to confirm this, and the alternative hypotheses haven't been examined and rejected.
4. It would be useful to comprehensively compare daSNVs with the other MPRA SNVs. For example, as compared to the other MPRA SNVs, what is the enrichment of differential activities across differentiation stages in daSNVs, and what is the enrichment of daSNVs in the neighborhood of homeostatic skin genes?
5. Two types of assays have been used to identify homeostatic TFs, repeatedly highlighting the crucial role of AP-1 and SP/KLF for epidermal homeostasis. However, there is a lack of explanation that an AP-1 TF, FOSL1, gives contradictory observations. FOSL1 knockout cells are enriched in both differentiated and undifferentiated populations (Fig. 3f). Also, I could not find any AP-1 TFs marked in the Perturb-seq results reported in Fig. 4e. What are their functions based on these results?
6. JUN/SP/KLF ASBs correlate well with MPRA results. The "buffering effect" of ASBs is also reported more frequently in promoter regions than in intergenic regions, which is interesting. However, further investigation is likely needed to confirm this finding. For example, how often is the "buffering effect" caused by multiple, closely positioned binding sites? How

frequently is it caused by low sequence specificity of TF binding (low information gain vs. high information gain of TF motifs)? What is the enrichment of the "buffering effect" in daSNPs vs. non-daSNPs?

Minor Concerns:

1. More than three hundred daSNVs were reported, which are suspected to be disease-causal. However, it is unclear if all the SNVs in Fig. 1j are daSNVs (it appears some do not show significant activity differences between risk vs. protective alleles). I'd suggest excluding likely non-causal daSNVs from this plot.
2. Fig. 2g is not very informative. For example, it is hard to locate the epidermal differentiation genes discussed in the text (page 11). Should they be represented in the nodes with the same color? Also, it might be better to represent genes associated with monogenic skin disorders using the same color. The same applies to the pro-inflammatory genes.
3. The "risk/protective activity" scale in Fig. 2d is different from that in Fig. 1j.
4. Please detail H3K27ac HiChIP experiments. Were they generated as part of this study or downloaded from a public repository?
5. Known TFs fall into two major functional groups, sustaining the progenitor state and facilitating differentiation. It might be helpful to represent TFs in different groups using different colors in Figs. 3, 4, and 5.
6. Line 390: FOSL1 is listed twice.
7. Should "Figure S4M" be "Extended data Fig. 4m"? There appear to be some other similar figure mis-citations.
8. In Figs. 7f, 7j, and 7m, the p-values between alleles should be provided either graphically or as part of the figure legend. In Fig. 7m, the activity differences between alleles seem insignificant across differentiation stages.
9. Similarly, in Figs. 7g, 7k, and 7n, the p-values of ASBs should be provided.
10. Line 188: "the transcriptional activity of rs4687102...". The target gene of the affected regulatory element is likely transcriptionally active, not the sequence variant itself.
11. Please report the fraction of CUT&RUN peaks with the primary motif of CXXC1, NRF1, and ZNF217 (lines 519-547).

Reviewer #2

(Remarks to the Author)

Reviewer #3

(Remarks to the Author)

In this manuscript, Porter and colleagues performed large-scale functional screenings to understand the roles of transcription factors involved in epidermis homeostasis on allelic function of common genetic variants associated with polygenic skin diseases. They performed 1) MPRA of 3,451 GWAS variants along the differentiation process of cultured keratinocytes, 2) CRISPR knockout screening of 1,772 transcription factors by measuring their effect on FACS-based keratinocyte differentiation level as well as single-cell based transcriptome changes along the keratinocyte differentiation trajectory, 3) allelic transcription factor (27 known TFs implicated in epiderma homeostasis) binding assessment using CUT&RUN approach on keratinocyte cultures from multiple individuals. They conclude that transcription factors that are important in keratinocyte differentiation captured in their in vitro model play an important role in multiple polygenic skin diseases. They provide validation experiments on several aspects of their findings. The amount of work put into this manuscript is commendable, and the resources made available online will be valuable for the research community. I recommend the authors to address the following points mainly for clarity and transparency.

Major points:

1. The three main parts of the paper each has a proper rationale on their own, but their convergence and logical connections could be improved. This will be helpful for readers to appreciate the main conclusions vs. additional insights from this dense paper. For example, but not limited to:
 - a. It seems the TF selection for CUT&RUN was guided by the results from the CRISPR screening and MPRA motif results. A flow chart of TF prioritization process connecting the MPRA and the knockout screening parts would be helpful.
 - b. Related to the above point, overlap of findings on the known & new homeostatic TFs from three main parts would be helpful. Figure 7O-P attempted to do this, but summary figures of the overlap between MPRA dSNV and CUT&RUN ASB in terms of the TFs will be helpful (perhaps conclusions gleaned from Figure S7K?).
 - c. How do indications of timepoint-specific TF bindings compare between MPRA (many dSNVs are timepoint-specific) and CUT&RUN approaches?
 - d. Do the CRISPR screening (and validation of a subset) and ASB by CUT&RUN help prioritize daSNVs with higher

confidence and/or help validate their target gene connections? What are the proportions of daSNVs validated by various complementary approaches? Are some of the ASBs also eQTLs?

e. If it helps to move some sections of the manuscript to supplementary text for brevity, TF effects on inflammation part might be one of them, given a weak support of the main argument.

2. Relevance of the choice of foreskin keratinocytes as the experimental system seems to be established in the previous studies, but it would be helpful to add more connections and validation if possible.

a. I wonder if any validation of differentiation status (e.g., inspection of morphology changes and/or marker gene expression) was done to standardize the timepoints between different screening approaches.

b. GWAS diseases were selected based on “disrupted epidermal homeostasis” as a key biological process. Is there literature support for a subset of included diseases representing main categories (e.g., cancer, autoimmune etc) based on genetic correlation studies, for example?

c. Similarly, have a subset of the diseases shown enrichment of keratinocyte-specific expression or regulatory signatures by LDSC, for example?

3. Some parts of the methods could benefit from more details.

a. MPRA part significantly lack necessary details required to interpret the results (e.g., construct, barcoding strategy, test sequence size etc). If the methods closely followed a previous study, please cite and state differences.

b. Sequencing chemistry and depth for MPRA and CRISPR-Flow/Perturb-seq could be informative.

c. Please provide primer sequences and gRNA sequences that are mentioned in the methods.

d. In the Perturb-seq analysis, how were the doublets filtered? It seems 40,000 cells were loaded in a 10X reaction, which could increase the doublet count significantly. Since the pseudotime and RNA velocity inference analyses could be affected by doublets, it is important to make sure to remove them stringently.

Minor points:

1. Line 142: “samples moving in the expected directions”. This might make more sense if controls (WT or non-targeting control) are plotted in Fig 1f together. There is no explanation of ST26 and NRF1 knock out.

2. Line 187: it is not clear how these presented SNPs were selected. Target genes of the variants based on eQTL or HiChIP might be more relevant than the nearest genes.

3. Line 209: Are 73 of 229 daSNVs linked to eGenes or promoter contact or both?

4. Line 231: Please describe the HiChIP dataset(s), including the cell type.

5. Line 406: it seems FACS screen was done at day 4 of differentiation while the perturb seq was done at day 3 and 6.

6. Line 436: knock-down cells must be knock-out cells.

Reviewer #4

(Remarks to the Author)

Authors present a resource in the form of massively parallel reporter gene analysis, CRISPR knockout screens and cut and run experiments to determine disease-associated SNVs and transcription factors important in skin differentiation through the use of in vitro assays in keratinocytes. They go on to identify a set of allele-specific differences important in maintenance of skin homeostasis which has relevance to epithelial biology. Whilst the data represent a large and potentially important catalogue for future use by the community I do have some issues with the presentation in the current draft of the manuscript. Specifically:

1. The abstract switches from MPRA and daSNV identification /editing to CRIPSR screens of TFs followed by CUT&RUN assays for a subset of these. CRIPSR screens identify potentially novel genes in the context of keratinocyte differentiation but how this relates to MPRA is not stated and how the allele specific differences were probed in the CUT&RUN approach and how these data also relate to MPRA experiments is unclear as written. Conclusions beyond a resource of genes and TFs important in skin differentiation are lacking and since the authors previously published a very similar resource using similar approaches it would be helpful to focus in on the salient new findings. This is compacted by the abstract from reference 4 which reads “Variants linked to human polygenic skin disease were enriched in these time-dependent combinatorial motif rules.” The abstract of the current submission would be greatly improved with a better take home message of new findings beyond the final sentence which really just reinforces what the scientific community already know – dysregulated differentiation of a tissue leads to disease.

2. Some of the figures are overly complicated and not well explained by the legends. A simple example being Figure 1d where the legend is “Number of variants by disease” essentially showing the 852 GWAS lead skin disease hits from reference 29, 26,732 from the same dataset that were in LD but not lead hits followed by 3,451 of the 26,732 that were associated with chromatin in cultured keratinocytes in reference 4, all stratified by the disease where the SNVs were identified from. In this example disease enrichment barely changes and has no real impact on the conclusions made. Overall, why data are being presented and what data are should be clearer since it is very difficult to distill the approach from the current presentation and much of the data could be punted to supplemental or even removed to make it easier to read.

3. There is a great deal of focus on disease genes since this is the starting point for the SNV screening but I feel this is overstated throughout since all the conclusions are drawn from in vitro experiments with keratinocytes. I appreciate that differentiation is important and that many genes perturbed in skin disease are involved in keratinocyte differentiation, but this is not a new concept and feels overstated throughout.

4. Did the MPRA experiments include replicate sequences from reference 29? I appreciate that the new submission includes new data from CRIPSR and CUT&RUN but there is no real acknowledgement of the fact that MPRA has previously been

performed in an identical way (albeit using 10 million keratinocytes as opposed to 12 million used in 2021) and no comparisons are commented upon. How the libraries differed would be helpful, maybe they are completely different sequences but they are comparing wild-type and mutated pairs which could be represented by allelic differences and would likely include similar sequences since much of the data are based on the same starting point.

Version 1:

Reviewer comments:

Reviewer #1

(Remarks to the Author)

In the revised manuscript, the authors have included additional Perturb-seq split-pool results, which further support the major findings and validate the quality of the accompanying data. The strong “buffering effect” of ASBs in promoters has been analyzed and well-explained. Most concerns have been properly addressed.

Below are some minor concerns:

1. Comparisons between daSNVs and other MPRA daSNVs (i.e., non daSNVs tested in MPRA) or all 26,278 SNVs identified as skin-disease-associated variants. How much are they associated with biological pathways and enriched in the proximity of monogenic genes linked to skin diseases? It will be a GREAT-like analysis (<http://great.stanford.edu/public/html/index.php>).

These analyses will help determine whether daSNVs are involved in core pathways underlying skin cell differentiation and/or skin disease development.

Furthermore, including ASB SNVs in these analyses could compare functional preferences between daSNVs and ASB SNVs.

2. Disparity between CRISPR and Perturb-seq for FOSL1. The response letter discusses this issue in detail, but the manuscript only briefly mentions it with the statement: “The idiosyncratic result of FOSL1 might reflect it being a highly selective gene” (page 15). It would be helpful to incorporate a more detailed summary of this discussion into the main text.

3. The statement “Perturbations were successful overall (Extended Data Fig. 8c)” requires further clarification. Is this conclusion based on the general observation that KO cells exhibit low gene expression levels (as shown in Fig. 8c)? Does TF knockout always lead to gene downregulation? A clearer explanation would be helpful.

4. The statement “ATF4 upregulates IL1A/IL1B/IL18 before late differentiation (Extended Data Fig. 10g)”(page 18) would be more compelling if supported by p-values or statistical significance measures.

5. Based on Fig. 6d, does IRX2 function for maintaining undifferentiated cells? Page 19, the statement is “In contrast, IRX2, FOXJ2, NRF1 ... depletions impaired normal differentiation gene induction (Fig. 6d-f)”.

6. The references to fig. 5h and fig. 5i might be typos (page 20).

7. “Both methods suggest that non-coding risk alleles may be less biased toward loss of binding than coding variants are biased toward loss of expression or folding”. Here, it lacks sufficient supporting the statement regarding coding variants.

Reviewer #2

(Remarks to the Author)

Reviewer #3

(Remarks to the Author)

The authors sufficiently addressed all the points raised by this referee.

Version 2:

Reviewer comments:

Reviewer #1

(Remarks to the Author)

The authors have thoroughly addressed all the concerns.

Reviewer #2

(Remarks to the Author)

REVIEWER COMMENTS

Reviewer #1 (Remarks to the Author):

The study by Porter and colleagues, titled "Disease-linked regulatory DNA variants and homeostatic transcription factors in epidermis," investigates regulatory variants and networks that govern the homeostasis and differentiation of keratinocytes. The authors have identified a panel of candidate regulatory variants linked to several skin diseases. They have assembled a large number of experimental datasets generated by this study into a publicly accessible web resource representing an atlas of regulatory networks, TFs, and variants underlying epidermal homeostasis. The experimental data spans various technologies, including MPRA, CRISPR-flow, Perturb-seq, siRNA RNA-seq, and CUT&RUN profiling keratinocytes at different development stages. With these data, the authors specifically focus on individual variants and TFs, carefully dissecting the etiology of several skin diseases and causative regulatory variants and allele-specific binding of corresponding TFs. This study provides a valuable resource for studying epidermal differentiation and skin diseases.

We are grateful to the Reviewer for their encouraging comments about the work as well as its value as a resource along with the very helpful suggestions to improve the manuscript.

The paper has been extensively revised based on feedback from the Reviewers feedback. Briefly, newly added data includes the following:

1. The **inclusion of an independent replication perturb-seq experiment**, this time using split-pool single cell sequencing and doubling the number of analyzed cells, substantially increasing statistical power.
2. ~25 new figure panels, many altered figures and overall reorganization, with **the supplement being increased from 7 to 18 figures, and main text figures being increased from 7 to 9.**
3. Addition of authors from Engreitz, Cong and Greenleaf labs at Stanford.
4. Application of more stringent quality metrics for CUT&RUN datasets, leading to removal of AHR, KHDRBS1 and HOPX from the paper.
5. Streamlining and revising Fig. 1. **New Fig.1f and 1i.**
6. Revisions to **Fig 2d and 2g.**
7. **New Fig 4** including split-pool.
8. **New Fig 5** including split-pool.
9. Revised Fig 8 (parts formerly in Fig 7), with **new panels 8h and 8i.**
10. Revised Fig 9 (parts formerly in Fig 9), with **new panels 9a-d**, and revised 9p.
11. **New Fig S1:** Sankey diagram of epithelial disease categories and incidence to help contextualize the medical relevance of the study.
12. **New Fig S2:** Flow chart of connections between experiments.
13. **New Fig S3:** Genetic correlations for disease risk between diseases.
14. **Fig S4** revised from previous Fig S1, now including part of Fig 1.
15. **New Fig S8-9** with splitpool perturb-seq.
16. **New Fig S13** on new analysis.

- 17. **New Fig S15e-i** with new analyses.
- 18. **New Fig S16** with new analyses.
- 19. **New Fig S18b.**
- 20. Revised abstract, introduction and discussion.
- 21. New Results paragraph on skin disease genetics.
- 22. New Results discussion of split-pool perturb-seq.
- 23. New Results sentence on TF-TF co-binding over differentiation.
- 24. New Results section on promoter buffering.
- 25. New Results sections on ASBs, daSNVs, eQTLs and the connection between MPRA and ASB.

Major Concerns:

1. While the presented study clearly demonstrates the causative effects of multiple TFs, the lack of reported TF-TF interactions is rather glaring. Those could probably be simply inferred as either overlapping CHIP peaks or proximal motif pairs.

We thank the Reviewer for the suggestion. While extensive analysis of potential interactions between TFs in the present CUT&RUN data is being performed in a separate manuscript in preparation that uses machine learning approaches in an independent PhD student thesis effort by a graduate student in our group, the current manuscript has also been revised in response to the Reviewer suggestion to begin to address this. In addition to **Fig 7b**, which illustrates correlations in binding between TFs, **Fig 7c** on cross-motif enrichment, and PCA plots in **Fig S12d** and **Fig S12e** that suggest the proximity of TFs based on similarity of binding, we now expand this analysis in the **new Fig S13a** (shown below). This revealed a number of co-binding associations (i.e., JUNB-JUN and SP1/3KLF4/5):

Additionally, differential Centrimo analysis of the enrichment of TF motifs in CnR peaks has now been performed to illustrate interactions as well as changes in interactions over differentiation (new Fig S13b, c), reproduced below:

Change in enrichment over differentiation

These analyses assess TF co-binding in undifferentiated and differentiating epidermal cells and suggest, for example, that an interaction between ATF4 and JUNB appears over differentiation. These observations are now noted in the revised text.

2. Throughout the manuscript, there appear to be multiple instances of inferring a bound TF simply via the presence of a particular motif. Computationally predicted TF binding

motifs are notoriously prone to incorrect TF calls. Please go through the text to emphasize the highly speculative nature of such calls. These motif-TF associations contrast drastically with the solid nature of the experimental data on TF binding presented in other parts of the manuscript.

We appreciate and agree with the Reviewer's point that the presence of a DNA motif does not automatically infer the presence of a bound TF and have accordingly revised the text to make the very tentative nature of these associations clearer, including the sentence *"It should be noted that the presence of a TF motif in DNA is only suggestive evidence for a TF binding to a given site, and not a reliable indicator of binding."*

3. The section "Identification of rs4687102..." appears rather speculative and might benefit from being moved to Supplemental Materials. While the "ATF1/2/4 motif" appears to be disrupted, the possibility of a disruption of a motif of another TF hasn't been rejected. The experiments result in an insignificant change in the binding of ATF4. While it might be possible that ATF4 binding is affected, there is a lack of solid evidence to confirm this, and the alternative hypotheses haven't been examined and rejected.

We thank the Reviewer for the improvement. We have added alternative hypothesis discussion to the manuscript based on ChIP-seq studies and have now moved the figure to the supplement as recommended.

4. It would be useful to comprehensively compare daSNVs with the other MPRA SNVs. For example, as compared to the other MPRA SNVs, what is the enrichment of differential activities across differentiation stages in daSNVs, and what is the enrichment of daSNVs in the neighborhood of homeostatic skin genes?

We thank the Reviewer for this insightful suggestion and have revised the manuscript to include these comparisons. We note representative daSNVs differential across timepoints are presented in Fig 2d, and address the two questions below:

1. As compared to the other MPRA SNVs, what is the enrichment of differential activities across differentiation stages in daSNVs?

Using "differential activity" to refer to the change in overall activity (ref and alt together) at a given timepoint from the overall activity at Day 0, we interpret the question as whether daSNVs have more differential activity than non-daSNVs. MPRA daSNVs show a slight enrichment of differential activity relative to non-daSNVs at Day 3 ($P=0.009$, Mann-Whitney U, 2-sided), but not at Day 6 ($P=0.36$). The effect is slight, and could be caused by daSNVs having more transcriptional activity in general, which leads to more detectable changes (that is, zero activity at day 0 may result in zero ref/alt differences and zero changes in activity across timepoints) (**Reviewer Fig. R1-1** below):

2. As compared to the other MPRA SNVs, what is the enrichment of daSNVs in the neighborhood of homeostatic skin genes?

Homeostatic genes have more SNVs clustered nearby them in the total test SNV pool, so picking SNVs at random and taking their nearby genes (or eGenes) as a gene list can easily produce enrichments of homeostatic genes, even if the subset of SNVs is random. To control for this effect, we performed a simulation test in which SNV subsets of the same size as the daSNV set are picked at random repeatedly and enrichment of the true daSNV set is then calculated relative to this random distribution. The results of this are noted below (**Reviewer Fig. R1-2 and R1-3**), with P values representing the fraction of random subsets with at least the same degree of enrichment or depletion:

Although some categories have $P < 0.05$, no value is close to significant when FDR adjusted. It should be noted that these categories are enriched relative to the genome, as the GWAS SNVs are enriched near homeostatic genes:

These points are now clarified in the revised manuscript text and we thank the Reviewer for raising them.

5. Two types of assays have been used to identify homeostatic TFs, repeatedly highlighting the crucial role of AP-1 and SP/KLF for epidermal homeostasis. However, there is a lack of explanation that an AP-1 TF, FOSL1, gives contradictory observations. FOSL1 knockout cells are enriched in both differentiated and undifferentiated populations (Fig. 3f). Also, I could not find any AP-1 TFs marked in the Perturb-seq results reported in Fig. 4e. What are their functions based on these results?

We thank the Reviewer for highlighting this. We start out by noting that FOSL1 is a strongly selective gene (see Depmap), and that survivorship-bias may influence results for such factors in perturb-seq screens. If a “true” phenotype is death or senescence, then the population we sample guides from may be unusual in their response to FOSL1 loss. FOSL1 is unusual in the CRISPR flow results, as seen in its relative isolation in a PCA plot of guides (Fig S7i). FOSL1 guides are enriched in keratinocytes allowed to continue growing (Day 0) relative to the input guide library (Fig 3g). This indicates intact FOSL1 inhibits proliferation. In addition, FOSL1 knock-out cells are depleted from both KRT10 high and low populations (Fig 3f) but are more depleted from KRT10 low than KRT10 high, leading to a classification (Fig 3g) as anti-differentiation and anti-proliferation. Being anti-proliferation should generally bias towards appearing pro-differentiation, so one property is not likely to be creating the other. In the perturb-seq data, FOSL1’s effect on gene expression is visible in Fig 4e (6th row from bottom). It is strongly depleted in progenitor gene expression, but only mildly upregulating differentiation gene expression (for example, a mild up-tick in KRT10). This is consistent with the depletion of FOSL1 guides from both progenitor and differentiated cell populations, with a greater depletion from progenitor populations – the model in which intact FOSL1 promotes proliferation and differentiation. To obtain more conclusive perturb-seq results, we have performed a validation perturb-seq experiment using the same guide libraries (this time in a CROP-seq vector) using a homebrew split-pool sequencing protocol and lower sequencing depth. Plotting the enrichment of AP-1/AP-2 KO cells along the differentiation trajectory (below) suggests there is a tendency for some AP-1 factors (TFAP2A, FOSB, MAF, MAFG, FOSL2, JUNB) to show a lack of differentiation upon KO, while some others (ATF4, FOSL1) show the opposite (**Reviewer Fig. R1-4** below),

In the new sequencing data, *FOSL1* KO cells are more enriched in the differentiated population, which would suggest a contrary result to intact *FOSL1* promoting differentiation. However, this enrichment could be influenced by the pro-proliferation effect of *FOSL1* (as in the Fig S7g diagram), and the cells differentiated in the absence of *FOSL1* may not have a correct differentiation pattern (as suggested by Fig 4e in the 10x data). We therefore find it most likely that damage to cellular health with *FOSL1* loss leads to unusual cell states that do not fit nicely onto the model we are using, and survivorship-bias from the cells detected with *FOSL1* guides may create inconsistency. These points are now clarified in the revised text.

6. JUN/SP/KLF ASBs correlate well with MPRA results. The “buffering effect” of ASBs is also reported more frequently in promoter regions than in intergenic regions, which is interesting. However, further investigation is likely needed to confirm this finding. For example, how often is the “buffering effect” caused by multiple, closely positioned binding sites? How frequently is it caused by low sequence specificity of TF binding (low information gain vs. high information gain of TF motifs)? What is the enrichment of the “buffering effect” in daSNPs vs. non-daSNPs?

This excellent question led to analyses that strengthened conclusions in the paper, and we thank the Reviewer for stimulating a closer look at this topic. In addition to the considerations below, we have also suggested a “peer pressure” model for promoter buffering, which is found in the results and the **new Fig S16**. SNVs may disrupt TF binding by altering known TF DNA binding motifs, by disrupting a nearby motif for another cooperatively binding TF, or by altering binding in some manner not captured by comparing sequences to the known motif PWMs, including by creating sites for DNA associated proteins that may compete for binding. How the potential sources of affinity may relate to the final output of binding is a complex topic, with the biological reality deriving from potentially complex biophysical assemblies that would not necessarily match predictions from simple models based on PWM matches. We attempt here to

address this in a limited way based on the suggestions from the Reviewer; we believe that an expanded biochemical analyses coupled with a statistically sophisticated model would be a topic of interest for an additional paper. Specific revisions in response to each question are noted below:

1. “For example, how often is the “buffering effect” caused by multiple, closely positioned binding sites?”

To address this question, we quantified motif clustering for peaks in different locations, using a simple distance cutoff of 50 nucleotides. It is indeed the case that there were larger clusters of motifs in the peaks at SNVs in promoters, suggesting that this effect likely contributes to increased resistance to mutation (the new Fig S15e below).

2. How frequently is it caused by low sequence specificity of TF binding (low information gain vs. high information gain of TF motifs)?

We interpret this suggestion as relating to the idea that SNVs in the promoter might affect lower information content positions than SNVs elsewhere. If background frequency of each base is 0.25, we calculate the IC at that position as the sum of $P(\text{base in PWM}) * \log_2(P(\text{base in PWM}) / 0.25)$. We could then test if the IC of SNVs in peaks at promoters is different from SNVs elsewhere, and whether they are correlated with the impact on binding. The IC of mutated bases in motifs in promoters were not generally different from those mutated elsewhere (the new Fig S15f below).

Related to this, the density of motifs and fraction of SNVs modified in promoter regions has also been quantified (the new Fig S15i below).

3. "What is the enrichment of the "buffering effect" in daSNPs vs. non-daSNPs?"

The number of daSNVs with ASB data is not high enough for strong statistics: only 369 interactions occur on SNVs included in MPRA, 86 interactions of which are on 47 daSNVs. The daSNVs in this group have ~1.3-fold higher fold changes ($P=0.1$, Mann-Whitney U, and $P=0.05$ by two-sided t-test), consistent with elements functional in MPRA enriching for altered TF binding. Testing the interaction between SNV location and daSNV status on fold change in ASB by ANOVA gave $P=0.16$, so a significant interaction was not identified. However, there may indeed be an effect: the ASB magnitude as a function of daSNV status and location is presented in the new Fig 9b below.

These points are now clarified in the revised manuscript text and we are grateful to the Reviewer for raising them.

Minor Concerns:

1. More than three hundred daSNVs were reported, which are suspected to be disease-causal. However, it is unclear if all the SNVs in Fig. 1j are daSNVs (it appears some do not show significant activity differences between risk vs. protective alleles). I'd suggest excluding likely non-causal daSNVs from this plot.

We apologize for the prior lack of clarity and thank the Reviewer for this comment. We have now excluded non-daSNVs from Fig 1j. Fig 1j has been regenerated using the following approach: 1. Time-differential effects were calculated separately for risk and protective alleles by MPRAnalyze, comparing \sim timepoint with \sim 1 models. Elements with $FDR < 0.001$ were retained. 2. Elements were ranked by the absolute log fold change for timepoint effects averaging risk and protective alleles. The maximum timepoint effect between D3 and D6 was used for ranking. 3. Elements that were not daSNVs at any of the three timepoints were excluded. 4. The top 20 remaining elements were plotted in the new Fig 1j. These points are now also clarified in the revised manuscript text.

2. Fig. 2g is not very informative. For example, it is hard to locate the epidermal differentiation genes discussed in the text (page 11). Should they be represented in the nodes with the same color? Also, it might be better to represent genes associated with monogenic skin disorders using the same color. The same applies to the pro-inflammatory genes.

We apologize for the lack of clarity and have examined each node in Fig 2g to recolor inflammation related genes. Epidermal differentiation genes are colored cyan and are now referred to as "epidermal differentiation genes" in the figure legend. Inflammation genes (as defined by inclusion under the GO term "immune system process") are now indicated by color. Monogenic skin disorder genes were drawn in a diamond shape because they overlap with functional categories (such as epidermal differentiation genes and inflammation) that are indicated by color; using a shape enables both

properties to be indicated at once. These changes have been incorporated into the revised Fig 2g.

3. The “risk/protective activity” scale in Fig. 2d is different from that in Fig. 1j.

We thank the Reviewer for this and have now made the scale bars in the figures previously termed Fig 1j and 2d the same. We have also regenerated Fig 2d according to a more specific test of time:allele interaction effects. MPRAnalyze was used to compare the models of ~allele + timepoint with ~allele + timepoint + allele:timepoint, calculating log likelihood ratios. Elements with FDR<0.1 in this comparison were retained, non-daSNVs were removed, and then elements were ranked by the maximum absolute timepoint:allele interaction effect. The top 20 SNVs with eGenes are now also plotted in the new Fig 2d.

4. Please detail H3K27ac HiChIP experiments. Were they generated as part of this study or downloaded from a public repository?

We apologize for the prior lack of clarity regarding the source of the H3K27ac HiChIP data. This is now more clearly cited in the text: *Donohue, L. K. H. et al. A cis-regulatory lexicon of DNA motif combinations mediating cell-type-specific gene regulation. Cell genomics 2, (2022)*. This work was performed previously by our laboratory using similar culturing conditions. We have added a clarifying note on this in the text.

5. Known TFs fall into two major functional groups, sustaining the progenitor state and facilitating differentiation. It might be helpful to represent TFs in different groups using different colors in Figs. 3, 4, and 5.

We appreciate this suggestion and have implemented it. Pro-differentiation genes are now in pink and progenitor genes are in blue.

6. Line 390: FOSL1 is listed twice.

This typo has now been corrected.

7. Should “Figure S4M” be “Extended data Fig. 4m”? There appear to be some other similar figure mis-citations.

Thank you, this has now been corrected.

8. In Figs. 7f, 7j, and 7m, the p-values between alleles should be provided either graphically or as part of the figure legend. In Fig. 7m, the activity differences between alleles seem insignificant across differentiation stages.

P-values between alleles have now been added to the figures. Previous Fig 7m (now Fig 9m) is now referred to in the figure legend as showing “the risk allele was not

significantly increased or decreased in MPRA activity”. We interpret the MPRA in Fig 9m as suggesting (visually and weakly) a change in variance, and the vignette is mentioned as a contrasting case between MPRA and CUT&RUN.

9. Similarly, in Figs. 7g, 7k, and 7n, the p-values of ASBs should be provided.

P values have now been added as recommended.

10. Line 188: “the transcriptional activity of rs4687102...”. The target gene of the affected regulatory element is likely transcriptionally active, not the sequence variant itself.

Thank you, the language has been corrected.

11. Please report the fraction of CUT&RUN peaks with the primary motif of CXXC1, NRF1, and ZNF217 (lines 519-547).

This has now been added to the **revised Fig S11d**.

Reviewer #2 (Remarks to the Author):

I co-reviewed this manuscript with one of the Reviewers who provided the listed reports. This is part of the Nature Communications initiative to facilitate training in peer review and to provide appropriate recognition for Early Career Researchers who co-review manuscripts.

We thank the Reviewer for their efforts in coordinating this very helpful review of the work.

Reviewer #3 (Remarks to the Author):

In this manuscript, Porter and colleagues performed large-scale functional screenings to understand the roles of transcription factors involved in epidermis homeostasis on allelic function of common genetic variants associated with polygenic skin diseases. They performed 1) MPRA of 3,451 GWAS variants along the differentiation process of cultured keratinocytes, 2) CRISPR knockout screening of 1,772 transcription factors by measuring their effect on FACS-based keratinocyte differentiation level as well as single-cell based transcriptome changes along the keratinocyte differentiation trajectory, 3) allelic transcription factor (27 known TFs implicated in epiderma homeostasis) binding assessment using CUT&RUN approach on keratinocyte cultures from multiple individuals. They conclude that transcription factors that are important in keratinocyte differentiation captured in their in vitro model play an important role in multiple polygenic skin diseases. They provide validation experiments on several aspects of their findings. The amount of work put into this manuscript is commendable, and the resources made available online will be valuable for the research community. I recommend the authors to address the following points mainly for clarity and transparency.

We thank the Reviewer for their recognition of the labor behind this manuscript and the value it can provide to the community.

The paper has been extensively revised based on feedback from the Reviewers feedback. Briefly, newly added data includes the following:

1. The **inclusion of an independent replication perturb-seq experiment**, this time using split-pool single cell sequencing and doubling the number of analyzed cells, substantially increasing statistical power.
2. ~25 new figure panels, many altered figures and overall reorganization, with **the supplement being increased from 7 to 18 figures, and main text figures being increased from 7 to 9.**
3. Addition of authors from Engreitz, Cong and Greenleaf labs at Stanford.
4. Application of more stringent quality metrics for CUT&RUN datasets, leading to removal of AHR, KHDRBS1 and HOPX from the paper.
5. Streamlining and revising Figure 1. New Figure 1f and 1i.
6. Revisions to **Fig 2d** and **2g**.
7. **New Fig 4** including split-pool.
8. **New Fig 5** including split-pool.
9. Revised Fig 8 (parts formerly in Fig 7), with **new panels 8h and 8i.**
10. Revised Fig 9 (parts formerly in Fig 9), with **new panels 9a-d**, and revised 9p.
11. **New Fig S1**: Sankey diagram of epithelial disease categories and incidence to help contextualize the medical relevance of the study.
12. **New Fig S2**: Flow chart of connections between experiments.
13. **New Fig S3**: Genetic correlations for disease risk between diseases.
14. **New Fig S4** was previously part of Fig 1.
15. **New Fig S8-9** with splitpool perturb-seq.
16. **New Fig S13** on new analysis.
17. **New Fig S15e-i** with new analyses.

- 18. **New Fig S16** with new analyses.
- 19. **New Fig S18b.**
- 20. Revised abstract, introduction and discussion.
- 21. New Results paragraph on skin disease genetics.
- 22. New Results discussion of split-pool perturb-seq.
- 23. New Results sentence on TF-TF co-binding over differentiation.
- 24. New Results section on promoter buffering.
- 25. New Results sections on ASBs, daSNVs, eQTLs and the connection between MPRA and ASB.

Major points:

1. The three main parts of the paper each has a proper rationale on their own, but their convergence and logical connections could be improved. This will be helpful for readers to appreciate the main conclusions vs. additional insights from this dense paper. For example, but not limited to:

a. It seems the TF selection for CUT&RUN was guided by the results from the CRISPR screening and MPRA motif results. A flow chart of TF prioritization process connecting the MPRA and the knockout screening parts would be helpful.

We thank the Reviewer for this valuable suggestion, which we believe significantly enhances the clarity of the paper. A flow-chart has accordingly been added in the **new Fig S2**, below. We did pick TFs based on the union set of TF motifs from MPRA, CRISPR hits, and known literature TFs as well as the availability of suitable antibody reagents. As a result, some logical targets were not included in the final dataset because most antibodies failed to generate usable data.

b. Related to the above point, overlap of findings on the known & new homeostatic TFs from three main parts would be helpful. Figure 7O-P attempted to do this, but summary figures of the overlap between MPRA dSNV and CUT&RUN ASB in terms of the TFs will be helpful (perhaps conclusions gleaned from Figure S7K?).

We agree with this suggestion and the comparison between MPRA hits and ASBs has now been expanded upon in the revised Fig 8, 9 and S18. Comparison figures now include:

1. The **new Fig 8h**: compare fraction of eQTLs.
2. The **new Fig 9a**: JUN in CUT&RUN vs MPRA.
4. The **new Fig 9b**: ASB magnitude vs daSNV.
5. The **new Fig 9c**: Risk vs protective directional effects in each.
6. The **new Fig 9d**: Alleles in MPRA that were ASBs.
7. The **new Fig 9e-m**: example SNVs.
3. The **new Fig S18a**: SP/KLF in CUT&RUN vs MPRA.
8. The **new Fig S18b**: upset plot of daSNVs in ASBs.

The provisional model has also been updated with conclusions gleaned. We believe these responses to Reviewer comments have substantially improved the work and are grateful for this suggestion.

c. How do indications of timepoint-specific TF bindings compare between MPRA (many dSNVs are timepoint-specific) and CUT&RUN approaches?

Evaluating average peak number changes between CUT&RUN timepoints (as a proxy for overall “activity”) compared to motif activity correlations produces the following chart (**Reviewer Fig 3R-1** below).

SNAI2 shows increased motif activity and decreased CUT&RUN binding because it is a repressor that is repressed by differentiation. GRHL1, OVOL1 and NRF1 are activators that are activated. Of interest, for unclear reasons, JUN and JUNB maintain their binding while their motif activity increases. The reason for this unknown to us. Focusing on SNVs and taking JUN as an example, there is a small increased ASB fold change for differentiated cells vs progenitor cells that is significant ($P=0.01$, Mann-Whitney U, two-sided). Simply correlating binding JUN D4 vs D0 and MPRA D3 vs D0 does not produce a correlation (**Reviewer Fig 3R-2** below):

The unexpected discordance between motif-activity correlations and binding for AP-1/AP-2 factors represents a potential topic of interest for a future body of work.

d. Do the CRISPR screening (and validation of a subset) and ASB by CUT&RUN help prioritize daSNVs with higher confidence and/or help validate their target gene connections? What are the proportions of daSNVs validated by various complementary approaches? Are some of the ASBs also eQTLs?

We thank the Reviewer for the suggestions and have addressed these as follows:

1. Do the CRISPR screening (and validation of a subset) and ASB by CUT&RUN help prioritize daSNVs with higher confidence and/or help validate their target gene connections?

We believe this may be the case on the basis that altered binding is among the straightforward mechanisms for a daSNV to be causal. The **new Fig 9d** highlights SNVs that were both daSNVs and had ASBs, and we note their closest genes are frequently very relevant to skin disease. A table on these SNVs is below for the Reviewer.

SNV	Nearest TSS	eGene(s)	Description	Role in skin?
rs4704864	NIPAL4	NIPAL4	Monogenic skin disease	Y
rs2742122	RAC1	RAC1	Required for epidermal development	Y
rs4265380	RUNX3		KC prolifer./diff. regulator	Y
rs11720523	FOXP1		Regulatory T cell development	Y
rs2349075	FLACC1	CASP8	CASP8 is a central apoptosis factor	Y

rs2180183	MFN2	TARDBP, MIIP, ect.		Likely
rs17154734	SEMA3C		Inflammation, neural crest development	Likely
rs834603	TNS3	TNS3	Cell adhesion, including in skin	Y
rs4687102	P3H2		Collagen modification, including in skin	Y
rs901886	ICAM5	TYK2, ICAM5	TYK2 is a drug target in psoriasis	Y
rs3131383	MSH5	C4A, C4B, ect.	Immune genes	Y
rs7748291	FGD2	FGD2, MTCH1	Cdc42 GEF	Likely
rs2470530	BTD		BTD is mutated to cause biotinidase deficiency	Y
rs10217259		ENSG00000213557	Ribosome L31P43 (pseudogene), with restricted expression	Unclear

2. What are the proportions of daSNVs validated by various complementary approaches?

By various complementary approaches, we believe the Reviewer might be including (1) the CUT&RUN data showing binding or ASBs, (2) the daSNV being an eQTL, and (3) the small number of SNVs validated by genome editing. We have addressed this by plotting the fraction of daSNVs falling in each category (with the exception of the genome editing subset, due to its small size) in the **new Fig S18B**, below.

3. Are some of the ASBs also eQTLs?

This is a great question and yes, 60% of ASB events were with a SNV considered an eQTL. **Reviewer Fig 3R-3** below displays the number of eGenes per ASB event. This result is consistent with most SNVs not being eQTLs, but many TF-bound SNVs having several eGenes, with some variants near HLA loci having 10-29 eGenes.

Interestingly, subsetting to SNVs linked to skin disease in our MPRA library, ASB SNVs were more likely to be eQTLs than SNVs that were not ASBs (90% vs 74%), consistent with ASBs identifying functional elements. These data are now included in the **new Fig 8h**.

e. If it helps to move some sections of the manuscript to supplementary text for brevity, TF effects on inflammation part might be one of them, given a weak support of the main argument.

The figures on inflammation have all been placed in supplement as suggested. Inflammation is so central to the skin diseases studied in this work, however, that mention of it has been retained in the main text.

2. Relevance of the choice of foreskin keratinocytes as the experimental system seems to be established in the previous studies, but it would be helpful to add more connections and validation if possible.

a. I wonder if any validation of differentiation status (e.g., inspection of morphology changes and/or marker gene expression) was done to standardize the timepoints between different screening approaches.

This is an excellent idea. We are able to distinguish by eye some basic features – cells seeded for differentiation can be distinguished visually from cells growing in the progenitor state, and sick cells adopt dendric morphology, etc. – but we could not distinguish by eye between cells at day 2 vs day 3 of differentiation, or other such subtle distinctions. We could generally observe that differentiation is occurring and that the cells are somewhere between initiation of the process and day 6 based on morphology, but we did not also assay marker gene expression. However, seeding cells for differentiation and assaying marker expression is done routinely in the lab and the lab’s protocol is reliable up to differences in magnitude of about a day of differentiation (that is, given similar seeding densities, different biological samples are generally similar in differentiation trajectory to within about a day or so). In addition to cellular morphology, marker gene expression was also used to validate progenitor, early and late differentiation cell states. These points are now clarified in the revised methods.

b. GWAS diseases were selected based on “disrupted epidermal homeostasis” as a key biological process. Is there literature support for a subset of included diseases representing main categories (e.g., cancer, autoimmune etc) based on genetic correlation studies, for example?

This is a valuable point. Some correlations could be found; for example, from the reference <https://www.nature.com/articles/s41467-023-38389-6> we could place SLE and rosacea together. However, we could not find a published comparison that includes the various diseases in this work and correlates their genetic risk. To better categorize the diseases, we have added the following:

1. A Sankey diagram (**the new Fig S1**) that includes the incidence numbers of the various diseases, categorizes them by molecular mechanism/clinical feature, and adds some brief description.
2. LD correlation analysis (**Fig S3, Reviewer Table below**). We began with identifying GWAS studies with summary statistics available:

Disease	Acc	Cases	Controls	Pop	Pop	Status
Acne	GCST90092000	20,165	595,231	20,165 European ancestry cases, 595,231 European ancestry controls	EUR	missing
Acne	GCST90080359	1,390	382,295	1,390 European ancestry cases, 382,295 European ancestry controls	EUR	CDS only
Alopecia	GCST006661	NA	NA	52,874 British ancestry males	EUR	No MAF
Atopic dermatitis	GCST90429798	1,706	17,190	1,706 Dutch ancestry cases, 17,190 Dutch ancestry controls	Dutch	Usable
Atopic dermatitis	GCST90080337	2,972	375,981	2,972 European ancestry cases, 375,981 European ancestry controls	EUR	CDS only
Atopic dermatitis	GCST90027161	22,474	774,187	22,474 European ancestry cases, 774,187 European ancestry controls	Finish	Usable

Basal cell carcinoma	GCST90081626	4,723	34,891	4,723 European ancestry cases, 34,891 European ancestry controls	EUR	Genes only, no snps
Basal cell carcinoma	GCST90041916	4,257	452,019	4,257 European ancestry cases, 452,019 European ancestry controls	EUR	Usable
Basal cell carcinoma	GCST90013410	17,416	375,455	17,416 European ancestry cases, 375,455 European ancestry controls	EUR	Usable
Basal cell carcinoma	GCST90137411	20,791	286,893	20,791 European ancestry cases, 286,893 European ancestry controls	EUR	Usable
Squamous cell carcinoma	GCST90137412	7,402	286,892	7,402 European ancestry cases, 286,892 European ancestry controls	EUR	Usable
Squamous cell carcinoma	GCST90041917	557	455,719	557 European ancestry cases, 455,719 European ancestry controls	EUR	Usable
Squamous cell carcinoma	GCST90077641	638	38,976	638 European ancestry cases, 38,976 European ancestry controls	EUR	CDS only
Psoriasis	GCST90013885	NA	NA	407,746 British ancestry individuals	EUR	No MAF
Psoriasis	GCST90019016	15,967	28,194	15,967 European ancestry cases, 28,194 European ancestry controls	EUR	No MAF
Psoriasis	GCST005527	10,588	22,806	10,588 European ancestry cases, 22,806 European ancestry controls	EUR	No MAF
Psoriasis	GCST90044515	338	456,010	338 European ancestry cases, 456,010 European ancestry controls	EUR	Usable
Psoriasis	GCST90080349	5,629	377,623	5,629 European ancestry cases, 377,623 European ancestry controls	EUR	CDS only
Psoriasis	GCST90014456	5,459	324,074	5,459 European ancestry cases, 324,074 European ancestry controls	EUR	Usable
Rosacea	GCST90044511	144	456,204	144 European ancestry cases, 456,204 European ancestry controls	EUR	Neg heritability
Rosacea	GCST90080360	4,002	378,283	4,002 European ancestry cases, 378,283 European ancestry controls	EUR	CDS only
Seborrheic dermatitis	GCST90080338	2,156	379,083	2,156 European ancestry cases, 379,083 European ancestry controls	EUR	CDS only

Unfortunately, only a subset of these studies could be fully analyzed, due to low size or lacking MAF data. Those that could be fully analyzed were correlated in the following diagram in the **new Fig S3a**:

As expected, given their shared neoplastic origin in epidermis, SCC and BCC correlate in their disease risk, while the other two diseases are associated with other distinct variants.

3. A dot plot of the overlap in risk alleles based on the risk alleles has been generated from GWAS data (the new **Fig S3b**). This revealed stronger overlaps for the SCC and BCC cancers, between inflammatory skin disease and psoriasis, and between rosacea and SLE. We note these three stronger overlaps are consistent with the literature and molecular pathology of the diseases, supporting the validity of this simple overlap calculation.

c. Similarly, have a subset of the diseases shown enrichment of keratinocyte-specific expression or regulatory signatures by LDSC, for example?

We thank the Reviewer for this astute suggestion and have now accordingly performed and added LDSC information in the **new Fig S3c-d**. LDSC analysis of SNV enrichment in cell type specific ATAC peaks revealed an enrichment for basal cell carcinoma SNVs in differentiated keratinocyte ATAC peaks. These GWAS studies represent some of the larger datasets available, and the smaller BCC study GCST90041916 did not show a significant enrichment. The fact that squamous cell carcinoma failed to show enrichment in differentiated KCs might simply reflect limitations in the input data, as the two studies are smaller than the BCC studies. The two atopic dermatitis GWAS studies that could be analyzed in this manner disagreed, with the Finnish sample enriching for GM12878 immune cells, and the general European one enriching for keratinocytes along with other epithelial cells; however, both studies agreed in the enrichment of T cell ATAC peaks (**new Fig S3d**), underscoring complexities in pathogenesis of these polygenic skin diseases as well as the epidermal and non-epidermal cell types that contribute to them.

3. Some parts of the methods could benefit from more details.

The methods section has now been expanded for clarity, with addition of ~3 additional pages of experimental detail.

a. MPRA part significantly lack necessary details required to interpret the results (e.g., construct, barcoding strategy, test sequence size etc). If the methods closely followed a previous study, please cite and state differences.

The methods section “Genetic variants linked to skin disease” and “MPRA” have been expanded to include the requested information.

b. Sequencing chemistry and depth for MPRA and CRISPR-Flow/Perturb-seq could be informative.

The methods section “MPRA” has been expanded to include the MPRA sequencing chemistry and depth. The “CRISPR-flow sequencing library construction” and “Perturb-seq” sections now include their chemistry and depth.

c. Please provide primer sequences and gRNA sequences that are mentioned in the methods.

Primer sequences and gRNA sequences are now included in the Methods and in Table S3.

d. In the Perturb-seq analysis, how were the doublets filtered? It seems 40,000 cells

were loaded in a 10X reaction, which could increase the doublet count significantly. Since the pseudotime and RNA velocity inference analyses could be affected by doublets, it is important to make sure to remove them stringently.

We appreciate the Reviewer's expertise on this. Doublets were removed by selecting for cells with specifically one guide RNA, which we believe will be more successful than usual doublet-removal methods. When it is incorrect, is likely to be incorrect due to a doublet with a cell that lacks a detected guide RNA. Although this failure is likely to occur, it does not appear to create artificial populations because low and high UMI count cells do not separate in the UMAP in the 10x data. It would of course alter the positions of cells that are doublets, but so long as this occurs randomly, it should only lower the sensitivity of the Mann-Whitney and effect size-based analysis of pseudotime values. Attempts at more stringent filtering of the new split-pool data resulted in known factors matching more poorly to established biology, suggesting that 10x-style doublet filtering reduced accuracy. In-progress work for a follow-up manuscript may improve these limitations.

Minor points:

1. Line 142: "samples moving in the expected directions". This might make more sense if controls (WT or non-targeting control) are plotted in Fig 1f together. There is no explanation of ST26 and NRF1 knock out.

We apologize for the lack of clarity. ST26 cells were the safe-target control cells and have been relabeled as such. NRF1 was removed for clarity. The figure and text have been updated to make this less confusing.

2. Line 187: it is not clear how these presented SNPs were selected. Target genes of the variants based on eQTL or HiChIP might be more relevant than the nearest genes. We apologize for the prior lack of clarity and thank the Reviewer for this comment. We have now excluded non-daSNVs from Fig 1j (now labeled Fig 1i). This figure has been regenerated using the following approach: 1. Time-differential effects were calculated separately for risk and protective alleles by MPRAnalyze, comparing ~timepoint with ~1 models. Elements with $FDR < 0.001$ were retained. 2. Elements were ranked by the absolute log fold change for timepoint effects averaging risk and protective alleles. The maximum timepoint effect between D3 and D6 was used for ranking. 3. Elements that were not daSNVs at any of the three timepoints were excluded. 4. The top 20 remaining elements were plotted in the **new Fig 1i**. The nearest gene is included because of limited space for additional columns in the figure, and that not all of these elements are QTLs; in addition, some are QTLs for many genes and looping data is context dependent and subject to experimental limitations, increasing the complexity of using looping for annotation. We believe that the nearest gene was a simple indicator of function present for all of these elements, which demonstrates that time-differential daSNV loci are clustered around genes of functional importance to skin homeostasis. For Figure 2d we have followed the Reviewer's suggestion and subset to SNVs with eGenes and indicated the eGene target rather than the nearest gene. These points are now also clarified in the revised manuscript text.

3. Line 209: Are 73 of 229 daSNVs linked to eGenes or promoter contact or both?
73 are linked by both promoter contact and as eQTLs. We have now clarified this in the text.

4. Line 231: Please describe the HiChIP dataset(s), including the cell type.
The source of the H3K27ac HiChIP data is cited in the text: *Donohue, L. K. H. et al. A cis-regulatory lexicon of DNA motif combinations mediating cell-type-specific gene regulation. Cell genomics 2, (2022)*. This work was performed previously by our laboratory using similar culturing conditions. We have added a note on this in the text.

5. Line 406: it seems FACS screen was done at day 4 of differentiation while the perturb seq was done at day 3 and 6.

We have now performed an additional perturb-seq at Day 4 of differentiation, approximately doubling the number of sequenced cells. Day 4 was chosen as an average between Day 3 and Day 6. KRT10 is an early differentiation marker that reduces again in late differentiation, so the CRISPR-flow screen was performed late enough to contain sufficient KRT10-positive cells (>10-15%, or at least Day 3), but not so late that confounding effects from late differentiation cells might become problematic.

6. Line 436: knock-down cells must be knock-out cells.
Corrected.

Reviewer #4 (Remarks to the Author):

Authors present a resource in the form of massively parallel reporter gene analysis, CRISPR knockout screens and cut and run experiments to determine disease-associated SNVs and transcription factors important in skin differentiation through the use of in vitro assays in keratinocytes. They go on to identify a set of allele-specific differences important in maintenance of skin homeostasis which has relevance to epithelial biology. Whilst the data represent a large and potentially important catalogue for future use by the community I do have some issues with the presentation in the current draft of the manuscript.

We are pleased that the Reviewer noted the value in the data and has focused very helpful criticisms on improving the work's presentation.

The paper has been extensively revised based on feedback from the Reviewers feedback. Briefly, newly added data includes the following:

1. The **inclusion of an independent replication perturb-seq experiment**, this time using split-pool single cell sequencing and doubling the number of analyzed cells, substantially increasing statistical power.
2. ~25 new figure panels, many altered figures and overall reorganization, with **the supplement being increased from 7 to 18 figures, and main text figures being increased from 7 to 9.**
3. Addition of authors from Engreitz, Cong and Greenleaf labs at Stanford.
4. Application of more stringent quality metrics for CUT&RUN datasets, leading to removal of AHR, KHDRBS1 and HOPX from the paper.
5. Streamlining and revising Figure 1. New Figure 1f and 1i.
6. Revisions to **Fig 2d** and **2g**.
7. **New Fig 4** including split-pool.
8. **New Fig 5** including split-pool.
9. Revised Fig 8 (parts formerly in Fig 7), with **new panels 8h and 8i.**
10. Revised Fig 9 (parts formerly in Fig 9), with **new panels 9a-d**, and revised 9p.
11. **New Fig S1:** Sankey diagram of epithelial disease categories and incidence to help contextualize the medical relevance of the study.
12. **New Fig S2:** Flow chart of connections between experiments.
13. **New Fig S3:** Genetic correlations for disease risk between diseases.
14. **New Fig S4** was previously part of Fig 1.
15. **New Fig S8-9** with splitpool perturb-seq.
16. **New Fig S13** on new analysis.
17. **New Fig S15e-i** with new analyses.
18. **New Fig S16** with new analyses.
19. **New Fig S18b.**
20. Revised abstract, introduction and discussion.
21. New Results paragraph on skin disease genetics.
22. New Results discussion of split-pool perturb-seq.
23. New Results sentence on TF-TF co-binding over differentiation.
24. New Results section on promoter buffering.

25. New Results sections on ASBs, daSNVs, eQTLs and the connection between MPRA and ASB.

1. The abstract switches from MPRA and daSNV identification /editing to CRISPR screens of TFs followed by CUT&RUN assays for a subset of these. CRISPR screens identify potentially novel genes in the context of keratinocyte differentiation but how this relates to MPRA is not stated and how the allele specific differences were probed in the CUT&RUN approach and how these data also relate to MPRA experiments is unclear as written. Conclusions beyond a resource of genes and TFs important in skin differentiation are lacking and since the authors previously published a very similar resource using similar approaches it would be helpful to focus in on the salient new findings. This is compacted by the abstract from reference 4 which reads “Variants linked to human polygenic skin disease were enriched in these time-dependent combinatorial motif rules.” The abstract of the current submission would be greatly improved with a better take home message of new findings beyond the final sentence which really just reinforces what the scientific community already know – dysregulated differentiation of a tissue leads to disease.

The Reviewer raises excellent points that have helped improve the revised work. Rearranging the text of the comment to respond point-by-point:

“The abstract switches from MPRA and daSNV identification /editing to CRISPR screens of TFs followed by CUT&RUN assays for a subset of these. CRISPR screens identify potentially novel genes in the context of keratinocyte differentiation but

how this relates to MPRA is not stated

The connection between MPRA and the CRISPR screens is now clarified with a flow chart in the **new Fig. S2** and the section “ASBs and transcriptional effects”.

how the allele specific differences were probed in the CUT&RUN approach

The revised text now describes how allele specific differences were probed in the section “Population sampling CUT&RUN identifies allele specific binding”.

how these data also relate to MPRA experiments is unclear as written.”

Comparisons between MPRA and allele specific differences in CUT&RUN are explored in the new Fig 9, and the section “ASBs and transcriptional effects”. MPRA motif analysis highlighted some families of TFs, which were then prioritized for CUT&RUN (as diagramed in the new Fig S2).

“Conclusions beyond a resource of genes and TFs important in skin differentiation are lacking[...].”

We apologize for our prior lack of clarity in pointing out biological Insights from this work beyond the identification of genes and TFs involved in skin differentiation. We now provide clarity on these, including the following, each of which is now better noted in the revised manuscript:

1. Promoter regions are “buffered” against changes in binding at SNVs in what may represent a conserved mechanism to prevent promoter failure by resisting the effect of potentially pathogenic mutations. This work identifies that motif clustering and binding from additional TFs may help resist altered binding in response to SNVs. We also find that highly-occupied “HOT” regions, regions of dense TF binding, are also buffered against ASB. To the best of our knowledge, this is a novel insight into human disease biology.
2. Studied TFs display distinct patterns of dynamic genomic localization across differentiation, including at promoters, enhancers, and gene bodies in a fashion that sheds light on diverse target gene repertoires in epidermal homeostasis.
3. Essential homeostatic TFs are enriched for binding proximal to DNA variants linked risk for polygenic skin diseases with epidermal involvement (over SNVs linked to non-skin diseases).
4. Binding of essential homeostatic TFs is enriched near differentiation genes as well as near genes whose protein coding mutation causes monogenic skin disease. This finding agrees with the premise that multiple genetic mechanisms (coding mutation, regulatory DNA variants) converge on a common set of essential differentiation genes to drive pathogenesis of both polygenic and Mendelian disorders.
5. Non-coding disease-linked variants recurrently perturb physiologic DNA binding by essential homeostatic TFs.
6. Integrating ASB data with CRISPR screening, motif analysis, and population sampling CUT&RUN data above identify outsized potential contributions for SP/KLF and AP-1/AP-2 TFs in converting non-coding variants into altered disease risk.
7. Specific insights into prevalent polygenic human skin diseases, including:
 - a. *FLG*: Gene editing data of the rs72696969 daSNV (linked by GWAS to risk for inflammatory skin disease, notably atopic dermatitis [U.S. prevalence >35 million persons]) showed diminished *FLG* expression by the risk nucleotide in epidermal keratinocytes compared to the protective nucleotide. *FLG* encodes the filaggrin protein (previously referred to as natural moisturizer factor) essential for barrier function, as exemplified by its causative protein coding mutation in the human Mendelian skin disorder, ichthyosis vulgaris. The latter is characterized by defective skin barrier function, indicating that more subtle dysregulation of epidermal differentiation-mediated barrier formation by this regulatory daSNV comprises a contributing pathogenic factor to the development of inflammatory skin disease.
 - b. *NIPAL4*: Gene editing data with the rs4704864 daSNV (linked by GWAS to risk for inflammatory skin diseases atopic dermatitis and psoriasis [combined U.S. prevalence >45 million persons]) diminished expression of the *NIPAL4* gene compared to the protective nucleotide. *NIPAL4* encodes a divalent cation transporter expressed in the differentiating epidermal granular layer. *NIPAL4*, which is also known as ichthyin, is essential for normal epidermal differentiation, as exemplified by its causative protein coding mutation in the human skin disorder, autosomal recessive congenital ichthyosis (ARCI). This disease is also characterized by defective skin barrier function, further underscoring that subtle dysregulation of epidermal differentiation-mediated barrier formation by this regulatory daSNV is a contributing pathogenic factor

in the development of inflammatory skin disease, consistent with the FLG example above.

- c. *CASP8*: Gene editing data of the rs2540334 daSNV (linked by GWAS to risk for cutaneous basal cell carcinoma and cutaneous squamous cell carcinoma [U.S. cancer incidence ~4.5 million persons per year]) diminished expression of the *CASP8* gene compared to the protective nucleotide. *CASP8* encodes a protease essential for physiologic cell death, including death after damage by ultraviolet light, which is the major carcinogenic mutagen in human skin. These findings suggest reduced *CASP8* due to this disease risk daSNV contributes to skin cancer risk by impairing removal of pre-malignant epidermal cells over the human lifespan.
8. In addition, to enhance the accessibility and impact of this Resource to the research community, new web portals have been generated, including a compendium of MPRA data for disease linked DNA variants and their target genes (<https://arvid-data.shinyapps.io/skin/>) along with the compendium of TF binding CUT&RUN data noted above (<https://genome.ucsc.edu/s/Max/cnr>).

“and since the authors previously published a very similar resource using similar approaches it would be helpful to focus in on the salient new findings.

We clarify that the previous DS Kim paper was directed at applying machine learning approaches to decode DNA motif grammars in normal epidermal differentiation and that it analyzed only 11 GWAS SNVs; only a single sequence overlapped with the current work. The DS Kim paper utilized sequences with motif pairs to identify rules of expression, while the current paper uses disease-linked variants to examine factors that may be important in “reading” disease-linked variation into altered phenotypes. These points are clarified in the revised Discussion.

This is compacted by the abstract from reference 4 which reads “Variants linked to human polygenic skin disease were enriched in these time-dependent combinatorial motif rules.” The abstract of the current submission would be greatly improved with a better take home message of new findings beyond the final sentence which really just reinforces what the scientific community already know – dysregulated differentiation of a tissue leads to disease.”

We thank the Reviewer for this very helpful suggestion. The abstract has now accordingly been revised to focus on novel findings and a take-home message more specific than dysregulated differentiation leads to disease.

2. Some of the figures are overly complicated and not well explained by the legends. A simple example being Figure 1d where the legend is “Number of variants by disease” essentially showing the 852 GWAS lead skin disease hits from reference 29, 26,732 from the same dataset that were in LD but not lead hits followed by 3,451 of the 26,732 that were associated with chromatin in cultured keratinocytes in reference 4, all stratified by the disease where the SNVs were identified from. In this example disease enrichment barely changes and has no real impact on the conclusions made. Overall, why data are being presented and what data are should be clearer since it is very difficult to distill the approach from the current presentation and much of the data could

be punted to supplemental or even removed to make it easier to read.

The Reviewer raises excellent points. Rearranging the text of the comment for organization:

1. “Some of the figures are overly complicated and not well explained by the legends. Figures have been rearranged for clarity, with the number of main figures increased from 7 to 9, and supplementary figures increased from 7 to 18. Figure legends were altered throughout for clarity.
2. A simple example being Figure 1d where the legend is “Number of variants by disease” essentially showing the 852 GWAS lead skin disease hits from reference 29, 26,732 from the same dataset that were in LD but not lead hits followed by 3,451 of the 26,732 that were associated with chromatin in cultured keratinocytes in reference 4, all stratified by the disease where the SNVs were identified from. In this example disease enrichment barely changes and has no real impact on the conclusions made. We agree with the Reviewer and apologize for our prior lack of clarity. The conclusion from the prior Fig 1d was that there was no enrichment. We have accordingly moved it to Fig S3.
3. Overall, why data are being presented and what data are should be clearer since it is very difficult to distill the approach from the current presentation and much of the data could be punted to supplemental or even removed to make it easier to read. “ We apologize for the prior lack of clarity and have completely reconfigured the manuscript to enhance the flow and clarity of the work, while augmenting it with substantial amounts of new data and analyses. The text and figure legends and figures have been further revised for clarity. Some figures, such as those on the *TP63* allele in previous Fig 1, have been moved to supplements to improve the flow of the paper.

3. There is a great deal of focus on disease genes since this is the starting point for the SNV screening but I feel this is overstated throughout since all the conclusions are drawn from in vitro experiments with keratinocytes. I appreciate that differentiation is important and that many genes perturbed in skin disease are involved in keratinocyte differentiation, but this is not a new concept and feels overstated throughout.

We appreciate the Reviewer’s thoughtful remark, which we have addressed in the following ways:

1. We have decreased the amount of text spent on emphasizing that differentiation genes are perturbed in skin disease.
2. We have now added LD score regression analysis (the **new Fig S3**), which is based on population statistics and is not in vitro.
3. We have shifted some of the emphasis to novel observations by the addition of two new main figures on the CUT&RUN analysis.
4. We have altered the discussion to focus more on new and specific findings, and less on the disease-differentiation connection.

4. Did the MPRA experiments include replicate sequences from reference 29? I appreciate that the new submission includes new data from CRISPR and CUT&RUN but there is no real acknowledgement of the fact that MPRA has previously been performed in an identical way (albeit using 10 million keratinocytes as opposed to 12 million used in 2021) and no comparisons are commented upon. How the libraries differed would be helpful, maybe they are completely different sequences but they are comparing wild-type and mutated pairs which could be represented by allelic differences and would likely include similar sequences since much of the data are based on the same starting point.

We thank the Reviewer for the suggestion. Reference 29 was <https://pubmed.ncbi.nlm.nih.gov/30445434/>, which is the GWAS catalog; we believe the Reviewer is referring to Reference 4, from our lab, which contains MPRA of keratinocytes using 12 million per replicate and was published in 2021: <https://www.nature.com/articles/s41588-021-00947-3>. The libraries differed in that the 2021 libraries were constructed from identifying fragments with motifs of interest, with only 11 sequences from GWAS. Only a single sequence from the 2021 libraries (chr3:189897953-189898439) overlapped with the MPRA sequences in this work. This point is now clarified in the revised text.

REVIEWER COMMENTS

Reviewer #1 (Remarks to the Author):

In the revised manuscript, the authors have included additional Perturb-seq split-pool results, which further support the major findings and validate the quality of the accompanying data. The strong “buffering effect” of ASBs in promoters has been analyzed and well-explained. Most concerns have been properly addressed.

We appreciate the positive assessment and Reviewers’ thoughtful remarks.

In addition to the responses below, the following minor modifications were made, none of which alter the main conclusions drawn from them:

1. A stray “o” character was removed as a typo from Fig S10c.
2. Table S3 sheet “Perturb-seq & K10 CRISPR-flow” was updated to include the split-pool perturb-seq results.
3. Figure S10e was modified such that the x-axis was recalculated using all of the perturb-seq data for the pseudotime effect. This increases the correlation from 0.11 to 0.18 because it no longer uses only the cells from which the pseudotime effects on IL1 expression were regressed out.
4. Figure 9f was slightly modified when figure-making scripts were rerun to produce source data files.
5. Figure S15c (ANOVA of ASBs) was modified by moving the read depth control factor to a log scale. The title was also changed from “Promoter buffering controlling for other factors” to “ASB buffering controlling for other factors”.
6. The SP3 bar was removed from Figure 9n.

Below are some minor concerns:

1. Comparisons between daSNVs and other MPRA daSNVs (i.e., non daSNVs tested in MPRA) or all 26,278 SNVs identified as skin-disease-associated variants. How much are they associated with biological pathways and enriched in the proximity of monogenic genes linked to skin diseases? It will be a GREAT-like analysis (<http://great.stanford.edu/public/html/index.php>).

These analyses will help determine whether daSNVs are involved in core pathways underlying skin cell differentiation and/or skin disease development.

Furthermore, including ASB SNVs in these analyses could compare functional preferences between daSNVs and ASB SNVs.

Using GREAT-like analysis, both the set of all MPRA SNVs and daSNVs are enriched for monogenic disease genes, both ~4-fold above the genome ($P=0$ and $P=5.107026e-15$ as reported by the R package rGREAT to perform GREAT-style analysis using a custom gene set). Using the default parameters associates each SNV with all TSS within a 1 Mbp distance, which is a very loose gene association. Consistent with our simulation test, when running with the default parameters, daSNVs were not significantly enriched relative to non-daSNVs ($P=0.3805507$, Binomial, from rGREAT). It was then reasoned that the subsetting to MPRA SNVs has already targeted the relevant general regions of the genome, but a tighter assignment of locations to genes might reveal an association.

Based on this logic, the assignment to genes was restricted to a maximum distance of 10 kbp instead of the default 1 Mbp, which produces a 1.76-fold enrichment for daSNVs vs non-daSNVs tested in MPRA at a significant $P=0.03$. Further restricting the assignment to just the closest TSS using the "oneClosest" parameter, increases the enrichment to 1.9 and lowers the P value to $P=0.02$.

Because these P values are not astronomical, the enrichment might still be chance, but it is some evidence that daSNVs have enriched for monogenic skin disease genes. Using the more stringent assignment to genes leads to a 22-fold enrichment of daSNVs relative to the genome ($P=1E-14$).

Querying GO term sets for daSNV vs non-daSNV MPRA SNV enrichments does not produce any terms that survive a multiple hypothesis adjustment (adjusted $P>0.2$). Using all regions as the control, rather than non-daSNVs, does produce numerous significant terms consistent with the gestalt of the gene regulatory network in Figure 2.

Top hit for BP: "positive regulation of CD8-positive, alpha-beta T cell activation", (P adjusted=0, 337-fold enriched):

Top CC: "cornified envelope" (P adjusted=0, 145-fold)

Top MF: structural constituent of skin epidermis (P adjusted=0, 82-fold enriched)

Performing the same analysis ("oneClosest", 10 kbp distance) on putative ASBs produces a 4.3-fold enrichment over the genome ($P=4E-11$) for monogenic disease genes, consistent with our previous analysis. As expected for healthy donors, putative ASBs were not enriched for monogenic disease gene proximity relative to the combined dataset of putative ASBs and MPRA-tested SNVs ($P>0.2$). The overlap set of putative ASBs and MPRA SNVs and variants within 10 kbp of a monogenic skin disease gene TSS was too small for robust analysis.

These results are now included in the text as the sentence:

“GREAT-style analysis revealed an enrichment of genes containing causative mutations for monogenic skin disease in the proximity of daSNVs, 22-fold vs the genome ($P=10^{-14}$) and 1.9-fold vs other MPRA-tested SNVs ($P=0.02$).”

2. Disparity between CRISPR and Perturb-seq for FOSL1. The response letter discusses this issue in detail, but the manuscript only briefly mentions it with the statement: “The idiosyncratic result of FOSL1 might reflect it being a highly selective gene” (page 15). It would be helpful to incorporate a more detailed summary of this discussion into the main text.

We thank the reviewer for the suggestion. We have added the following to the text: “FOSL1 was unusual in perturb-seq as well as CRISPR-flow: progenitor RNA expression was decreased, while differentiation RNAs were inconsistently upregulated (Fig. 4e, bottom block). As a selective gene, FOSL1 loss may lead to cells that are neither progenitors nor correctly differentiated.”

3. The statement “Perturbations were successful overall (Extended Data Fig. 8c)” requires further clarification. Is this conclusion based on the general observation that KO cells exhibit low gene expression levels (as shown in Fig. 8c)? Does TF knockout always lead to gene downregulation? A clearer explanation would be helpful.

We apologize for the lack of clarity. Extended Data Fig 8c depicts the expression of the target gene RNA in KO cells vs all cells. That is, we separate cells with an IRF6 guide detected in the single cell sequencing and plot the histogram of IRF6 expression in those cells vs all cells. Because single cell sequencing has limited reads-per-cell, and TFs are often not highly expressed, we only applied this to the TFs in the top half of expression. It is also true that CRISPR KOs do not always lead to decreased RNA expression of the target RNA; if the edit causes an in-frame deletion or point mutation, NMD would not be triggered, and the target RNA level might be unreduced or even increased by feed-back mechanisms responding to a loss of its protein’s function. However, it appears that NMD is triggered frequently enough to observe a clear down-regulation of target RNAs. This is now reflected in the revised sentence “Perturbations were successful overall, as cells with a guide targeting a TF had reduced average RNA expression for the target TF, consistent with activated nonsense-mediated decay (Extended Data Fig. 8c).”

4. The statement “ATF4 upregulates IL1A/IL1B/IL18 before late differentiation (Extended Data Fig. 10g)”(page 18) would be more compelling if supported by p-values or statistical significance measures.

We agree this section needed statistical support. The sentence has been updated to state “ATF4 upregulates IL1A/IL1B/IL18 before late differentiation (Extended Data Fig.

10g, t-test FDR 0.15).” This was calculated by the following method (now included in the Methods section): 1. The average gene expression is calculated across all cells and LOWESS smoothing is used to generate an “expected” gene expression value for a given pseudotime value. 2. The “expected” gene expression value is subtracted from the actual observed value for each cell to generate the residual. 3. Whether a gene has a significant effect on gene expression is calculated by performing a t-test between the residuals for KO cells vs non-KO cells. 4. This number is corrected to a BH FDR number to correct for testing multiple TFs.

5. Based on Fig. 6d, does IRX2 function for maintaining undifferentiated cells? Page 19, the statement is “In contrast, IRX2, FOXJ2, NRF1 ... depletions impaired normal differentiation gene induction (Fig. 6d-f)”.

We agree this is what the figure suggests: based on Fig 6d, IRX2 would appear to maintain undifferentiated cells in progenitor conditions while being required for proper differentiation under differentiation conditions. We focused on the latter observation because it was better supported. The screening data was only for differentiating conditions and the “IRX2 promotes differentiation” effect was observed in both FACS and Perturb-seq, so the observation was reproduced by both siRNA and CRISPR KOs using multiple assays. The “maintenance of the progenitor state under proliferating conditions” phenotype could only be evaluated in the siRNA experiment, and as a result we have less confidence in it. In addition, a general impairment of cell health will tend to bias towards differentiation in progenitor conditions, making it less interesting than the pro-differentiation effect. We also note that RNAi against IRX2 shows a greater general impact on cell proliferation than CRISPR in DepMap’s cancer lines, raising questions about whether an effect seen with siRNA might reproduce.

We have added the line “A progenitor maintenance role for IRX2 is also tentatively suggested by **Fig 6d**.” to the text.

6. The references to fig. 5h and fig. 5i might be typos (page 20).

These references were indeed typos and have been corrected to fig 6h and 6i. We thank the Reviewer for the catch.

7. “Both methods suggest that non-coding risk alleles may be less biased toward loss of binding than coding variants are biased toward loss of expression or folding”. Here, it lacks sufficient supporting the statement regarding coding variants.

Upon further review, we have removed this suggestion as hasty. This comparison requires more nuance than the current MS has space for, and the issue is complicated by observation biases in the ASB approach here vs a random sampling of protein-coding mutations such as in <https://www.nature.com/articles/s41467-019-11959-3>. For

example, we are more likely to detect ASBs at sites that are highly bound (and therefore potentially functional sites) and at variants that are closest to 50% MAF in the population (and therefore more likely benign variants) because such cases would be the most likely to generate many reads at a heterozygous site in a random individual. We have changed this sentence clause to “both methods therefore suggest non-coding risk alleles are not dramatically biased to loss-of-binding.”

Reviewer #2 (Remarks to the Author):

We thank the Reviewer for their service in reviewing the manuscript.

Reviewer #3 (Remarks to the Author):

The authors sufficiently addressed all the points raised by this referee.

We thank the Reviewer for their helpful suggestions and positive assessment.